# Exact risk curves of signSGD in High-Dimensions: quantifying preconditioning and noise-compression effects

**Ke Liang Xiao** [1] **Noah Marshall** [1] **Atish Agarwala** [2] **Elliot Paquette** [1]

## Abstract

In recent years, SIGNSGD has garnered interest as both a practical optimizer as well as a simple model to understand adaptive optimizers like ADAM. Though there is a general consensus that SIGNSGD acts to precondition optimization and reshapes noise, quantitatively understanding these effects in theoretically solvable settings remains difficult. We present an analysis of SIGNSGD in a high-dimensional limit, and derive a limiting SDE and ODE to describe the risk. Using this framework we quantify four effects of SIGNSGD: effective learning rate, noise compression, diagonal preconditioning, and gradient noise reshaping. Our analysis is consistent with experimental observations but moves beyond that by quantifying the dependence of these effects on the data and noise distributions. We conclude with a conjecture on how these results might be extended to ADAM.

## 1. Introduction

The success of deep learning has been driven by the effectiveness of relatively simple stochastic optimization algorithms. Stochastic gradient descent (SGD) with momentum can be used to train models like ResNet50 with minimal hyperparameter tuning. The workhorse of modern machine learning is ADAM, which was designed to give an approximation of preconditioning with a diagonal, online approximation of the Fisher information matrix (Kingma, 2014). Additional hypotheses for the success of ADAM include its ability to maintain balanced updates to parameters across layers and its potential noise-mitigating effects (Zhang et al., 2020b; 2024). Getting a quantitative, theoretical understand-

ing of Adam and its variants is hindered by their complexity. While the multiple exponential moving averages are easy to implement, they complicate analysis.

The practical desire for simpler, more efficient learning algorithms as well as the theoretical desire for simpler models to analyze have led to a resurgence in the study of SIGNSGD. SIGNSGD is a variant of SGD where the stochastic gradient is passed through the sign function $\sigma$, leading to an update vector of $\pm 1$s. On average, SIGNSGD's updates at every step have positive dot product with the average SGD step, but it can have dramatically different convergence properties (Bernstein et al., 2018a; Karimireddy et al., 2019). Multiple studies point towards sign-based methods as an effective proxy given that the sign component of the gradient has been shown to play an important role in ADAM (Kunstner et al., 2023; Balles & Hennig, 2018; Bernstein et al., 2018b). SIGNSGD is also the basis for new practical methods; the LION algorithm (Chen et al., 2023) combines SIGNSGD with multiple exponential moving averages, and SIGNSGD + momentum was used to train LLMs with performance comparable to ADAM (Zhao et al., 2024).

Despite the promise of SIGNSGD, a detailed quantitative understanding of its dynamics in realistic settings remain elusive—in particular the nature of the preconditioning and the effect of the $\sigma$ function on the noise are not well understood. A crucial first step is to understand these effects on quadratic optimization problems.

Motivated by these questions, we provide the first analysis of the learning dynamics of SIGNSGD in a high-dimensional stochastic setting (Section 2). We make the following contributions:

- We derive a limiting stochastic differential equation (SDE) for SIGNSGD and combine it with a concentration result to derive a deterministic ordinary differential equation (ODE) that describes the dynamics of the risk in our setting (Section 3).

- We compare the dynamics of SIGNSGD and vanilla SGD, isolating 4 effects: effective learning rate, noise-compression, diagonal preconditioning, and gradient noise reshaping (Section 4).

[1]Department of Mathematics and Statistics, McGill University, Montreal, Canada [2]Google Deepmind, Mountain View, United States of America. Correspondence to: Ke Liang Xiao <keliang.xiao@mail.mcgill.ca>.

*Proceedings of the 42$^{nd}$ International Conference on Machine Learning*, Vancouver, Canada. PMLR 267, 2025. Copyright 2025 by the author(s).

- We quantitatively analyze these four effects and their contributions to learning, including exact results in specific settings (remainder of Section 4).

Our work addresses significant technical challenges in analyzing both the preconditioning and noise transformation effects of SIGNSGD. Our analysis is consistent with more general experimental observations about adaptive methods, but provides a more quantitative understanding in our setting. We conclude with a discussion of the implications of our results for future study of adaptive algorithms, including a conjecture on the limiting form of ADAM in an equivalent setting.

In concurrent work, Compagnoni et al. (2025) also derives SDEs for SIGNSGD in the weak-approximation setting of (Li et al., 2019). We discuss this in more detail in Appendix G.

## 2. Problem Setup

Our work considers linear regression using the mean-squared loss $\mathcal{L}$ in the one-pass scenario, where data is not reused. SIGNSGD, with mini-batching of size $b$, is first initialized by some $\boldsymbol{\theta}_0 \in \mathbb{R}^d$ and then follows the update rule:

$$\boldsymbol{\theta}_{k+1} = \boldsymbol{\theta}_k - \eta_k' \sigma \left( \frac{1}{b} \sum_{i=1}^{b} \nabla_{\boldsymbol{\theta}} \mathcal{L}(\boldsymbol{\theta}_k, \mathbf{x}_{k+1}^{(i)}, y_{k+1}^{(i)}) \right), \quad (1)$$

$$\mathcal{L}(\boldsymbol{\theta}, \mathbf{x}, y) = \| \langle \mathbf{x}, \boldsymbol{\theta} \rangle - y \|^2 / 2, \quad (2)$$

where $\sigma$ denotes the sign function applied element-wise and $\nabla_{\boldsymbol{\theta}} \mathcal{L}(\boldsymbol{\theta}_k, \mathbf{x}_{k+1}, y_{k+1}) = (\langle \boldsymbol{\theta}_k, \mathbf{x}_{k+1} \rangle - y_{k+1}) \mathbf{x}_{k+1}$. It is typically assumed that the mini-batch $(\mathbf{x}_{k+1}^{(i)}, y_{k+1}^{(i)})$ for $1 \le i \le b$ are drawn i.i.d.

We will assume that the samples $\{(\mathbf{x}_k, y_k)\}_{k \ge 0}$, consisting of data $\mathbf{x}_k$ and targets $y_k$, satisfy the following:

**Assumption 1.** *The data* $\mathbf{x}$ *are mean* $0$ *and Gaussian with positive definite covariance matrix* $\mathbf{K} \in \mathbb{R}^{d \times d}$. *The targets* $y$ *are generated by* $y = \langle \mathbf{x}, \boldsymbol{\theta}_* \rangle + \epsilon$, *where* $\boldsymbol{\theta}_*$ *is the ground-truth and* $\epsilon$ *the label noise.*

**Definition 1.** *Define the population risk* $\mathcal{P}$ *and the noiseless risk* $\mathcal{R}$:

$$\mathcal{P}(\boldsymbol{\theta}) = \mathbb{E}_{(\mathbf{x}, y)} \left[ (\langle \mathbf{x}, \boldsymbol{\theta} \rangle - y)^2 \right] / 2$$
$$\mathcal{R}(\boldsymbol{\theta}) = \mathbb{E}_{\mathbf{x}} \left[ \langle \mathbf{x}, \boldsymbol{\theta} - \boldsymbol{\theta}_* \rangle^2 \right] / 2. \quad (3)$$

Although our theory is framed in the setting of Gaussian data, as we will see, the results are still a good description for real-world, *a priori* non-Gaussian settings (Figure 1). This is an instance of *universality*, wherein the details of the data distribution do not affect the precise high-dimensional limit

law (see discussion in (Tao, 2023) section 2.2). Formalizing this is left to future work.

In contrast, the distribution of the label noise has a nontrivial impact on the behavior of the process. We shall require that the noise is well-behaved in a neighborhood around 0.

**Assumption 2.** *There exists* $a_0 > 0$ *such that the law of the noise* $\epsilon$ *has an* $C^2$ *density on* $(-a_0, a_0)$.

Assumption 2 ensures our SDE (8) is Lipschitz (c.f. Lemma 12) and applies to many distributions; it encompasses heavy-tailed distributions such as $\alpha$-stable laws, and we make no assumptions on any tail properties of the noise. Due to the non-smoothness of the $\sigma$ function at 0, extraordinary behavior of the noise near 0 will lead to degraded performance of SIGNSGD as the risk vanishes. At the cost of a less-informative theorem, it is possible to drop Assumption 2; see Theorem 6 in the Appendix.

An important characterizing feature of SIGNSGD is its effect on the covariance of the signed stochastic gradients. We introduce the following transformations on $\mathbf{K}$:

$$\overline{\mathbf{K}} \equiv \mathbf{D}^{-1} \mathbf{K} \quad \text{and}$$

$$\mathbf{K}_{\sigma} \equiv \left[ \frac{\pi}{2} \mathbb{E}_{\mathbf{x}} [\sigma(\mathbf{x}_i) \sigma(\mathbf{x}_j)] \right]_{i,j} = \left[ \arcsin \left( \frac{\mathbf{K}_{ij}}{\sqrt{\mathbf{K}_{ii} \mathbf{K}_{jj}}} \right) \right]_{i,j}, \quad (4)$$

where $\mathbf{D} = \sqrt{\text{diag}(\mathbf{K})}$. We remark that $\overline{\mathbf{K}}$ is similar in the matrix-sense to $\mathbf{D}^{-\frac{1}{2}} \mathbf{K} \mathbf{D}^{-\frac{1}{2}}$, thus $\overline{\mathbf{K}}$ has all real, positive eigenvalues. $\mathbf{K}_{\sigma}$ is proportional to the covariance of $\sigma(\mathbf{x})$. We assume some properties of the matrices $\mathbf{K}$, $\overline{\mathbf{K}}$, and $\mathbf{K}_{\sigma}$.

**Assumption 3.** *Suppose:*
  *i). The spectrum of* $\mathbf{K}$ *is bounded from above and away from* $0$ *independently of* $d$.
  *ii). The sign-data matrix* $\mathbf{K}_{\sigma}$ *also has operator norm bounded independent of* $d$.
  *iii). The resolvent of* $\overline{\mathbf{K}}$ *defined by* $\mathbf{R}(z; \overline{\mathbf{K}}) = (\overline{\mathbf{K}} - z\mathbf{I})^{-1}$ *satisfies*

$$\max_{i \le d} \max_{i \ne j} \| \mathbf{R}(z; \overline{\mathbf{K}})_{ij} \| = O \left( \frac{d^{\delta_0}}{\sqrt{d}} \right), \quad (5)$$

  *for all* $z \in \partial B_{2\|\overline{\mathbf{K}}\|}$ *and for some* $\delta_0 < 1/12$. *(Equivalently, one may instead assume the same bounds with* $\overline{\mathbf{K}}$ *replaced by* $\mathbf{K}$*).*

The upper bound on $\mathbf{K}$ in Assumption 3 *(i)* is standard and can always be achieved by rescaling the risk. But the lower bound is a nontrivial assumption that is necessary for analyzing how $\sigma$ affects the stochastic gradient. Assumption 3 *(ii)* is convenient for the proof. A full understanding of when it holds is highly nontrivial; there exists some theory establishing it for some random $\mathbf{K}$ (Fan & Montanari, 2019). A simple case where (ii) is satisfied is when $\|\mathbf{D}^{-1} \mathbf{K} \mathbf{D}^{-1}\| < 1$.

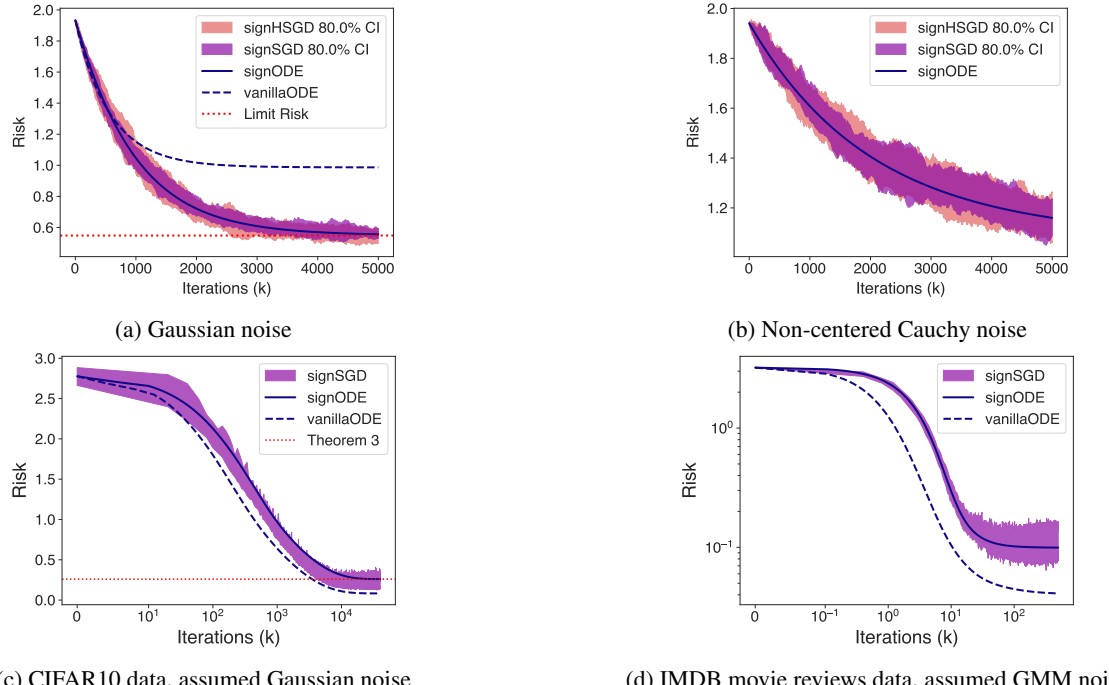

Figure 1: Dynamics of the risk under SIGNSGD and SIGNHSGD on synthetic and real datasets. SIGNHSGD and its deterministic equivalent ODE are good models for the risk dynamics even for $d = 500$ (a, b) or on real datasets (c, d). The convergence of SIGNSGD for Cauchy noise (b) is remarkable given that SGD fails to converge there. The usefulness of the ODE on CIFAR10 and IMDB movie reviews is remarkable due to the non-Gaussian nature of the data, and the significant estimation of key quantities like $\boldsymbol{\theta}_*$ or $\epsilon$. For the CIFAR10 dataset, we validate the results of Theorem 3 which gives the limit risk of SIGNODE under Gaussian data. We include the deterministic equivalent for SGD (VANILLAODE) for reference. Details of these experiments may be found in Appendix I.

Assumption 3 *(iii)* can be interpreted as a condition that the eigenvectors of $\mathbf{K}$ contain no low-dimensional structure: for example, it is satisfied with high probability if the eigenvectors of $\mathbf{K}$ are taken to be uniformly random. Additionally, it is trivially satisfied for any diagonal $\mathbf{K}$. For a further discussion, including applicability to real data, see Paquette & Paquette (2022), Figure 2.

We assume the learning rates have a high-dimensional limiting profile:

**Assumption 4.** *The learning rates follow*

$$\eta'_t = \eta(t/d)/d, \tag{6}$$

*where $\eta : \mathbb{R}^+ \to \mathbb{R}^+$ is a continuous bounded function. We will write $\eta_t$ for $\eta(t)$.*

This scaling is critical: it ensures that as the problem size grows, both the bias and variance terms in the risk evolution are balanced (see e.g. Equation (25)).

Finally, we assume the initialization remains (stochastically) bounded across $d$:

**Assumption 5.** *The difference between $\boldsymbol{\theta}_*$ and initialization*

$\boldsymbol{\theta}_0$ *satisfies*

$$\mathbb{P}\left(\left|\mathbf{R}(z;\overline{\mathbf{K}})_i^{\mathrm{T}}(\boldsymbol{\theta}_0 - \boldsymbol{\theta}_*)\right| \geq t\right) \leq \\ C \exp\left(-ct^2 d/\|\mathbf{R}(z;\overline{\mathbf{K}})_i\|^2\right), \tag{7}$$

*for all $1 \leq i \leq d$ with absolute, positive constants $c, C$.*

For example, this assumption holds for any initialization and target generated by normalized Gaussian. In the case that the initialization and target are deterministic, if $\boldsymbol{\theta}_0 - \boldsymbol{\theta}_*$ have entries of order $O(\frac{1}{\sqrt{d}})$ then rotating the data $\mathbf{x}$ by a Haar-distributed orthogonal matrix $\mathbf{U}$, we obtain a new data covariance matrix $\mathbf{A} = \mathbf{U}\mathbf{K}\mathbf{U}^{\mathrm{T}}$. Moreover, $\overline{\mathbf{A}} \approx \sqrt{d/\operatorname{Tr}(\mathbf{K})}\mathbf{A}$ with high probability. See (33). It follows with high probability that $\mathbf{R}(z;\overline{\mathbf{A}}) \approx \frac{1}{\xi}\mathbf{U}\mathbf{R}(z/\xi;\mathbf{K})\mathbf{U}^{\mathrm{T}}$ where $\xi = \sqrt{\operatorname{Tr}(\mathbf{K})/d}$. One may then derive a similar probabilistic bound as in (7) on non-symmetric rays $(-\infty, -t + O(\frac{1}{\sqrt{d}})]$ and $[t + O(\frac{1}{\sqrt{d}}), \infty)$, where the probability is now taken with respect to $\mathbf{U}$.

## 3. SIGNHSGD

The analysis of SIGNSGD in high-dimensional settings presents a unique set of technical challenges and requires

careful mathematical treatment. A core difficulty lies in the transformative effect of the sign operator on the gradient. Unlike traditional SGD, where the gradient direction remains consistent with the magnitude of the update, SIGNSGD changes the gradient's direction, via a *non-Lipschitz* compression operation. This compression alters the optimization landscape observed by the optimizer, in ways we will explore in Section 4.

Nonetheless, we show that under the assumptions above, there is a *continuous* stochastic process Sign-Homogenized SGD (SIGNHSGD) which captures the high-dimensional behaviour of SIGNSGD; see (Thygesen, 2023) or (Karatzas & Shreve, 1991) for background on SDEs.

**Definition 2** (SIGNHSGD). *We define* $\boldsymbol{\Theta}_t$ *as the solution of the stochastic differential equation:*

$$\mathrm{d}\boldsymbol{\Theta}_t = -\eta_t N_b \frac{\varphi^{(b)}(\mathcal{R}(\boldsymbol{\Theta}_t))}{\sqrt{2\mathcal{R}(\boldsymbol{\Theta}_t)}} \overline{\mathbf{K}}(\boldsymbol{\Theta}_t - \boldsymbol{\theta}_*)\mathrm{d}t + \eta_t \sqrt{\frac{2\mathbf{K}_\sigma}{\pi d}}\mathrm{d}\mathbf{B}_t,$$
(8)

*with initial condition* $\boldsymbol{\Theta}_0 = \boldsymbol{\theta}_0$. *Here, the constant* $N_b = \mathbb{E}[\|(Z_1, Z_2, \ldots, Z_b)\|]/\mathbb{E}[|Z_1|]$ *for i.i.d. standard normals* $Z_j$, *and*

$$\varphi^{(b)}(\mathcal{R}(\boldsymbol{\Theta}_t)) = \frac{2}{\pi}\mathbb{E}_{\epsilon_b}\left[\exp\left(\frac{-\epsilon_b^2}{4\mathcal{R}(\boldsymbol{\Theta}_t)}\right)\right],$$
(9)

*where* $\epsilon_b = \sum_{i=1}^b \epsilon^{(i)}\omega_i$ *for* $\{\epsilon^{(i)}\}_{i=1}^b$ *and* $\{\omega_i\}_{i=1}^b$ *i.i.d samples of* $\epsilon$ *and entries of* $b$-*dimensional uniform vectors on the sphere, respectively.*

**Remark 1.** $N_b$ *may be computed explicitly to be*

$$N_b = \frac{\Gamma((b+1)/2)}{\Gamma(b/2)\Gamma(1/2)} \sim \sqrt{b} \quad as\ b \to \infty.$$
(10)

*This is consistent with the square-root scaling hypothesis, as the bias increases relative to the noise by a factor of square-root of the batch (which leads to the popular square-root learning rate scaling rule).*

**Remark 2.** *In the case* $\epsilon \equiv 0$, *we take* $\varphi^{(b)}(x) \equiv 2/\pi$. *While this* $\epsilon$ *does not satisfy Assumption 2, we formulate Theorem 6 in the Appendix which covers this case.*

It is worth noting that, in practice, $\varphi^{(b)}$ is often easy and inexpensive to compute numerically; we compute it analytically for some common distributions (Figure 2). In general, it is simple to fit a Gaussian mixture model to the noise and use that to compute $\varphi^{(b)}$ (Appendix I).

For the remainder of this paper, we focus on the single batch setting, i.e., $b = 1$. For notational convenience, we will write $\varphi$ in place of $\varphi^{(1)}$ in the single batch setting. Numerical validation of our main result, Theorem 1, for larger batch sizes ($b > 1$) may be seen in Figure 5.

We can now state the first part of our main theorem:

**Theorem 1** (Main Theorem, part 1). *Given* $b = 1$, *Assumptions 1–5 and choosing any fixed even moment* $2p \in (0, d)$, *there exists a constant* $C(\overline{\mathbf{K}}, \epsilon) > 0$ *such that for any* $\delta \in (1/3, 1/2)$ *and all* $T > 3$,

$$\sup_{0 \le t \le T}|\mathcal{R}(\boldsymbol{\theta}_{\lfloor td \rfloor}) - \mathcal{R}(\boldsymbol{\Theta}_t)| \le$$
$$\frac{Td^\delta \|\mathbf{K}\|}{\sqrt{d}}\exp\left(C(\overline{\mathbf{K}}, \epsilon)\|\eta\|_\infty T\right),$$
(11)

*with probability at least* $1 - c(2p, \overline{\mathbf{K}})d^{p(1/3-\delta)}$ *for a constant* $c(2p, \overline{\mathbf{K}})$ *independent to* $d$.

In other words, the risk curves of SIGNSGD are well approximated by the risk curves of SIGNHSGD and this approximation improves as dimension grows. Numerical simulations suggest that in practice this correspondence is strong even by $d = 500$ (Figure 1 (a), (b)). Other statistics like iterate norms or distance to optimality can also be computed using a general result across quadratics (Theorem 5 in the Appendix).

The risk curves of both SIGNSGD and SIGNHSGD concentrate around the same deterministic path. Let $R_t$ be this *deterministic equivalent* of SIGNSGD; we will refer to the solution as SIGNODE for reasons that will soon be made clear. In order to find the deterministic equivalent we introduce a family of scalars $\{r_i\}_{i=1}^d$ which loosely correspond to the magnitudes of the residual $\boldsymbol{\Theta}_t - \boldsymbol{\theta}_*$ projected onto an eigenbasis (see Appendix B). The sum of these scalars then gives the deterministic equivalent for the risk:

$$R_t \overset{def}{=} \sum_{i=1}^d r_i(t).$$
(12)

The scalars $r_i$ follow a coupled system of ODEs:

$$\frac{\mathrm{d}r_i}{\mathrm{d}t} = -2\eta_t\frac{\varphi(R_t)}{\sqrt{2R_t}}\lambda_i(\overline{\mathbf{K}})r_i + \eta_t^2\frac{\mathbf{w}_i^\mathrm{T}\mathbf{K}_\sigma\mathbf{K}\mathbf{u}_i}{\pi d}$$
(13a)

$$r_i(0) = \frac{1}{2}\langle\boldsymbol{\theta}_0 - \boldsymbol{\theta}_*, \mathbf{K}\mathbf{u}_i\rangle\langle\mathbf{w}_i, \boldsymbol{\theta}_0 - \boldsymbol{\theta}_*\rangle$$
(13b)

where $\lambda_i(\overline{\mathbf{K}})$, $\mathbf{u}_i$ and $\mathbf{w}_i$ are the eigenvalues and left/right eigenvectors of $\overline{\mathbf{K}}$ respectively. A similar argument, can be used to derive a coupled system of ODEs that describe the risk of vanilla SGD (Collins-Woodfin & Paquette, 2023), which we will call VANILLAODE (Appendix B).

We can now present a deterministic version of Theorem 1:

**Theorem 2** (Main Theorem, part 2). *Let* $R_t$ *be given by* (12) *and* (13). *Then given Assumptions 1–5 and choosing any fixed even moment* $2p \in (0, d)$ *there exists a constant* $C(\overline{\mathbf{K}}, \epsilon) > 0$ *such that for any* $\delta \in (1/3, 1/2)$ *and all* $T > 3$,

$$\sup_{0 \le t \le T}|\mathcal{R}(\boldsymbol{\theta}_{\lfloor td \rfloor}) - R_t| \le$$
$$Td^{\delta-1/2}\|\mathbf{K}\|\exp\left(C(\overline{\mathbf{K}}, \epsilon)\|\eta\|_\infty T\right),$$
(14)

| Distribution | $\varphi(x)$ |
|---|---|
| 0 | $\frac{2}{\pi}$ |
| $N\left(0, v^2\right)$ | $\frac{2\sqrt{2x}}{\pi\sqrt{2x+v^2}}$ |
| Rademacher | $\frac{2}{\pi}\exp\left(-\frac{1}{4x}\right)$ |
| Unif $(-1,1)$ | $2\sqrt{\frac{x}{\pi}}\,\mathrm{erf}\left(\frac{1}{2\sqrt{x}}\right)$ |
| $\sqrt{\mathrm{Levy}(\lambda)}$ | $\frac{2}{\pi}\exp\left(-\sqrt{\frac{\lambda}{2x}}\right)$ |

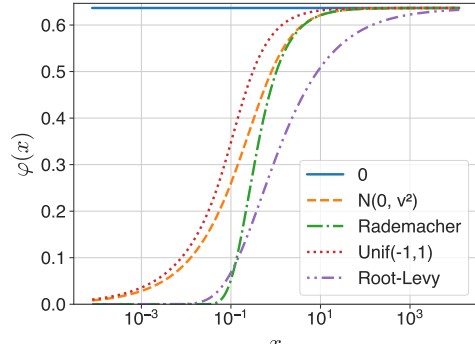

Figure 2: Examples of $\varphi$ for simple noise distributions. $\sqrt{\mathrm{Levy}}$ has Cauchy type-tails and vanishing density near 0. We note that $\varphi(x)$ is trivially bounded above by $\frac{2}{\pi}$ and converges to $\frac{2}{\pi}$ as $x \to \infty$; the rate of convergence at $\infty$ is related to the tail decay rate. At 0, $\varphi(x)/\sqrt{x}$ converges to the density of the noise at 0 scaled by $2/\pi$.

*with probability at least $1 - c(2p, \overline{\mathbf{K}})d^{p(1/3-\delta)}$ for a constant $c(2p, \overline{\mathbf{K}})$ independent to $d$.*

This ODE captures the behavior of the risk even at finite $d = 500$ (Figure 1 (a), (b)). Moreover, it seems to capture the behavior of high-dimensional linear regression on real, non-Gaussian datasets as well (Figure 1 (c), (d)).

## 4. Comparing SIGNSGD to vanilla SGD

To produce an apples-to-apples comparison, we compare the SIGNHSGD to the analogous SDE for vanilla SGD from (Collins-Woodfin & Paquette, 2023):

$$d\mathbf{\Theta}_t^{\mathrm{SGD}} = -\eta_t^{\mathrm{SGD}}\mathbf{K}(\mathbf{\Theta}_t^{\mathrm{SGD}} - \boldsymbol{\theta}_*)dt$$
$$+ \eta_t^{\mathrm{SGD}}\sqrt{\frac{2\mathbf{K}\mathcal{P}(\mathbf{\Theta}_t^{\mathrm{SGD}})}{d}}d\mathbf{B}_t \quad (15)$$

To control for the adaptive-scheduling inherent in SIGNSGD, we run vanilla SGD with a risk dependent learning rate schedule $\eta_t^{\mathrm{SGD}}$ given by

$$\eta_t^{\mathrm{SGD}} = \frac{2}{\pi}\frac{\eta_t}{\sqrt{2\mathcal{P}(\mathbf{\Theta}_t^{\mathrm{SGD}})}} = \frac{2}{\pi}\frac{\eta_t}{\sqrt{\mathbb{E}\|\nabla_{\boldsymbol{\theta}}\mathcal{L}(\boldsymbol{\theta}, \mathbf{x}, y)\|^2}}, \quad (16)$$

which is to say that we scale the steps in SGD inversely proportional to the norm of the gradients. We note that the $\mathcal{P}$-risk requires the noise $\epsilon$ to have finite variance $v$; indeed if the variance is infinite, then SIGNSGD is overwhelmingly favored, see Remark 1 in Zhang et al. (2020a). Training SIGNSGD with learning rate $\eta_t$ and SGD with learning rate $\eta_t^{\mathrm{SGD}}$, we can use (8) to write, with $\psi$ as in (18)),

$$d\mathbf{\Theta}_t^{\mathrm{SGD}} = -\eta_t^{\mathrm{SGD}}\mathbf{K}(\mathbf{\Theta}_t^{\mathrm{SGD}} - \boldsymbol{\theta}_*)dt + \eta_t\sqrt{\frac{4\mathbf{K}}{\pi^2 d}}d\mathbf{B}_t \quad (17a)$$

$$d\mathbf{\Theta}_t = -\eta_t^{\mathrm{SGD}}\underbrace{\psi(\mathcal{R}(\mathbf{\Theta}_t))}_{\epsilon\text{-compress.}}\underbrace{\mathbf{D}^{-1}}_{\text{D.Precond.}}\mathbf{K}(\mathbf{\Theta}_t - \boldsymbol{\theta}_*)dt$$
$$+ \eta_t\underbrace{\sqrt{\frac{2\mathbf{K}_\sigma}{\pi d}}}_{\text{Reshape}}d\mathbf{B}_t. \quad (17b)$$

Comparison of the minibatched version can be found in Appendix D.

We summarize the precise effects below:

**Effective learning rate.** The effective learning rate of SIGNSGD can be considered as risk dependent, effectively matching the expected $\ell^2$–norm of a gradient.

**$\epsilon$-compression.** The distribution of the label noise (be it from model-misspecification or otherwise) rescales the bias term. Formally, letting $v^2 = \mathbb{E}[\epsilon^2]$,

$$\psi(x) = \frac{\pi\varphi(x)\sqrt{2x + v^2}}{2\sqrt{2x}}. \quad (18)$$

**Diagonal preconditioner.** The matrix $\mathbf{D}^{-1}$ gives the diagonal preconditioner ($\mathbf{D}_{ii} = \sqrt{\mathbf{K}_{ii}}$).

**Gradient noise reshaping.** Finally, passing the gradient through the $\sigma$ function results in a different covariance structure to the gradients, which is accounted for in the differing diffusion term.

Although all the effects appear in concert in SIGNSGD, we will now attempt to isolate and address each one separately in the following sections.

## 4.1. Effective learning rate and convergence

We recall that to match the learning rate of SGD to SIGNSGD, we had to use the identification (16),

$$\eta_t^{\text{SGD}} = \frac{2}{\pi} \frac{\eta_t}{\sqrt{2\mathcal{P}(\Theta_t^{\text{SGD}})}}.$$

In particular, the effective learning rate gets smaller when the optimizer's position is far from optimality and gets larger as it gets closer. In the convex setting this is generally undesirable at both extremes. When far from optimality, the algorithm slows far beyond what would tend to be favorable, while at small risks this behavior can impede convergence. On the other hand, it can easily be addressed by using a learning rate schedule.

In a nonconvex setting, identifying $2\mathcal{P}(\Theta_t^{\text{SGD}})$ with the expected square-norm of the gradients (c.f. (16)), one possible benefit of this schedule is that it may be helpful in dynamically adjusting to saddle manifolds in the loss landscape.

### 4.1.1. STATIONARY POINT OF SIGNSGD

If the learning rate is *any* constant $\eta_t \equiv \eta$, we have a unique stationary point of the ODE system (13a) which is locally attractive. The $\eta$ dependence of this stationary point demonstrates the effect of an aggressive learning rate, which is accentuated in the presence of small noise variance $v$.

**Theorem 3.** *With fixed learning rate $\eta_t \equiv \eta \in (0, \infty)$ and $\epsilon \sim N(0, v^2)$, the ODEs have a unique stationary point $[s_i : 1 \le i \le d]$ given by Equation (244). Then, the limiting risk, $R_\infty = \sum s_i$, is given by*

$$R_\infty = \frac{\pi\eta}{32d} \text{Tr}(\mathbf{D}) \left( \frac{\pi\eta \, \text{Tr}(\mathbf{D})}{2d} + \sqrt{\frac{\pi^2\eta^2 \, \text{Tr}(\mathbf{D})^2}{4d^2} + 16v^2} \right).$$

(19)

Notice that the limiting risk's dependence on $\eta$ changes depending on the relationship between $\eta$ and $v$, for small $\eta$ it will be proportional to $\eta$. See Figure 7 for numerical validation, and see (255) for analogous SGD limit risk.

## 4.2. $\epsilon$-compression

The influence of the distribution of the noise $\epsilon$ on the optimization, in the case of finite variance, can be summarized by (c.f. (18) and (9))

$$\psi(\mathcal{R}) = \mathbb{E}\left[ \exp\left( \frac{-\epsilon^2}{4\mathcal{R}} \right) \right] \times \sqrt{1 + \frac{\mathbb{E}[\epsilon^2]}{2\mathcal{R}}}. \quad (20)$$

When $\psi < 1$, the descent term of (17b) is decreased, and hence SIGNSGD is slowed with respect to SGD with learning rate $\eta_t^{\text{SGD}}$. Conversely, when $\psi > 1$ the descent term is increased, and SIGNSGD is favored. When $\mathbb{E}[\epsilon^2] = \infty$,

$\psi$ can be interpreted as $\infty$, corresponding to overwhelming SIGNSGD favor, although the quantitative meaning in (17b) breaks down.

In the Gaussian case, $\psi = 1$. Hence, we can interpret $\psi$ as the effect that *deviation from Gaussianity* has on the drift term of the SDE. We note that all the influence of the label noise $\epsilon$ on SIGNSGD is entirely through (20) which in turn only depends on $\epsilon^2$. Hence SIGNSGD *symmetrizes* the noise distribution.

A full comparison of SGD and SIGNSGD requires optimizing the learning rates of both algorithms independently. We will show that in the case of isotropic data, this procedure is tractable and produces a different threshold $\psi = \frac{\pi}{2}$ above which SIGNSGD is favored (see Equation (28)).

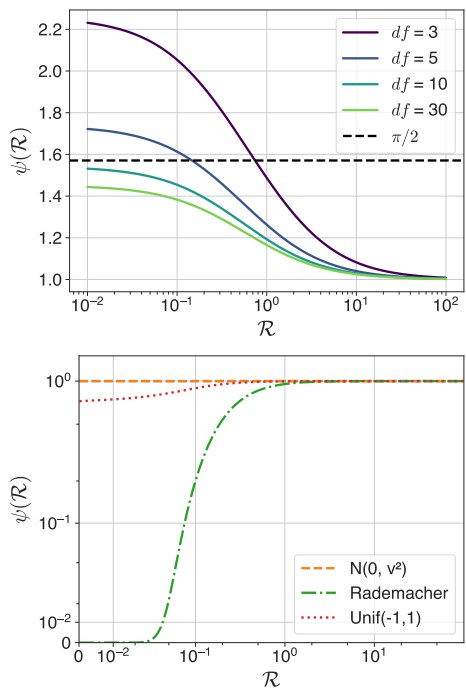

Figure 3: Top: $\psi$ for Student's-$t$. Here $\psi$ is always greater than 1 and $\epsilon$-compression accelerates SIGNSGD. For sufficiently small $df$, $\psi > \pi/2$ over some range of $\mathcal{R}$ and SIGNSGD also converges faster than SGD in the isotropic setting. Bottom: $\psi$ for $N(0, v^2)$, Rademacher, Unif$(-1, 1)$.

**Setups favoring SIGNSGD.** In the presence of heavy tails, $\psi(\mathcal{R})$ can be large and hence SIGNSGD is very favored. Indeed, among some parametric classes, such as the Student's-t family, this is observed numerically to always be larger than 1 (Figure 3, left) and increase to $\infty$ as the kurtosis increases. More generally, as $\mathcal{R}$ tends to 0, letting $f_\epsilon(0)$ be the density of the noise at 0, one has

$$\lim_{\mathcal{R} \to 0} \psi(\mathcal{R}) = \sqrt{2\pi\mathbb{E}[\epsilon^2]} f_\epsilon(0), \quad (21)$$

which can be arbitrarily large, in particular when the noise distribution puts more mass in its tails than at 0.

Conversely, for *all* distributions, we also observe that when the risk is relatively large, SIGNSGD is always modestly favored over SGD under the $\eta^{\text{SGD}}$ learning rate as we have

$$1 \leq \mathbb{E}\left[1 - \frac{\epsilon^2}{4\mathcal{R}}\right] \times \sqrt{1 + \frac{\mathbb{E}[\epsilon^2]}{\mathcal{R}}} \leq \psi(\mathcal{R}) \leq \sqrt{1 + \frac{\mathbb{E}[\epsilon^2]}{\mathcal{R}}} \tag{22}$$

for all $\mathbb{E}[\epsilon^2]/\mathcal{R} \leq \frac{3}{2}$.

**Setups where SIGNSGD does not improve.** For light-tailed noises, the factor $\psi$ can only mildly favor SGD. A density $f$ on $\mathbb{R}$ is called *log-concave* if it can be written as $e^g$ for concave $g$ (see (Saumard & Wellner, 2014) for discussion). The exponential, uniform and many other canonical noise distributions are log-concave. Note these decay no slower than exponentially at infinity. Then as $\varphi(\mathcal{R})/\sqrt{4\pi\mathcal{R}}$ is the density at 0 of a log-concave density, Saumard & Wellner (2014), Proposition 5.2 gives us

$$\psi(\mathcal{R}) \leq \sqrt{2\pi}. \tag{23}$$

Hence, for these distributions, while there may be limited gains from using SIGNSGD, they are bounded by an absolute constant factor.

**Setups where SIGNSGD is catastrophic.** In the situation that the noise is bounded away from 0 by some $\delta$, it follows that we have the upper bound:

$$\psi(\mathcal{R}) \leq e^{-\frac{\delta^2}{4\mathcal{R}}} \times \sqrt{1 + \frac{\mathbb{E}[\epsilon^2]}{2\mathcal{R}}}. \tag{24}$$

This tends to 0 *exponentially* in $1/\mathcal{R}$ (e.g. see the Rademacher case of Figure 3). For such noise distributions, SIGNSGD will effectively experience a floor on the risk, which is completely induced by distributional properties of the noise (and unrelated to the underlying optimization problem geometry). In this situation, SGD is heavily favored for small risks, which are seen late in training.

**Scheduling SIGNSGD.** We have discussed adjusting the SGD learning rate to match the behaviour of SIGNSGD. However, when using SIGNSGD there is the question of how to optimize its learning rate. This is analytically tractable for isotropic data $\mathbf{K} = \mathbf{I}_d$, for which $\overline{\mathbf{K}} = \mathbf{K}$ and $\mathbf{K}_\sigma = \frac{\pi}{2}\mathbf{I}_d$ which allows us to isolate the effects of the label noise. It is easy to check that the $d$-system of ODEs for SIGNSGD in (12) may be reduced to the following single ODE:

$$\frac{\mathrm{d}R_t}{\mathrm{d}t} = -\frac{2\eta_t \varphi(R_t)}{\sqrt{2R_t}} R_t + \frac{\eta_t^2}{2}, \qquad R_0 = \mathcal{R}(\boldsymbol{\theta}_0). \tag{25}$$

If we greedily optimize in $\eta_t$ we arrive at

$$\frac{\mathrm{d}R_t}{\mathrm{d}t} = -\varphi(R_t)^2 R_t, \qquad \text{where} \quad \eta_t^* = \varphi(R_t)\sqrt{2R_t}. \tag{26}$$

So generally for large risks, the optimal stepsize compensates for the effective gradient rescaling in (16). This compensation is seen for all risks in the Gaussian $\epsilon$ setting.

As a point of comparison, we may repeat the same procedure for the SGD risk ODE $R^{\text{S}}$ with learning rate $\eta^{\text{S}}$, which can be derived from (15):

$$\frac{\mathrm{d}R_t^{\text{S}}}{\mathrm{d}t} = -2\eta_t^{\text{S}} R_t^{\text{S}} + \frac{(\eta_t^{\text{S}})^2}{2}(2R_t^{\text{S}} + v^2)$$
$$\frac{\mathrm{d}R_t^{\text{S}}}{\mathrm{d}t} = -\frac{2R_t^{\text{S}}}{2R_t^{\text{S}} + v^2} R_t^{\text{S}} \quad (\text{optimized in } \eta^{\text{S}}). \tag{27}$$

Hence (26) can also be expressed as

$$\frac{\mathrm{d}R_t}{\mathrm{d}t} = -\left(\frac{4}{\pi^2}\psi^2(R_t)\right) \times \frac{2R_t}{2R_t + v^2} R_t. \tag{28}$$

Thus the performance benefits of SIGNSGD using the optimal learning rate can again be reduced to a question of the magnitude of $\psi$, albeit with a crossover at $\psi = \pi/2$.

In the non-isotropic setting, locally greedy stepsizes can be very far from optimal, even with two eigenvalues (Collins-Woodfin et al., 2024). But we expect the conclusion of (28) remains mostly true in well-conditioned settings.

### 4.3. Diagonal preconditioner

Next, and strikingly, we see that SIGNSGD performs a diagonal preconditioning step on the gradients, with the preconditioner given by $\mathbf{D}_{ii} = \sqrt{\mathbf{K}_{ii}} = \sqrt{\mathbb{E}[\mathbf{x}_i^2]}$, where $\mathbf{x}$ is a sample. To produce this bias term in SGD, we would need to run the algorithm

$$\boldsymbol{\theta}_{k+1} = \boldsymbol{\theta}_k - \eta_k \mathbf{D}^{-1}\left(\nabla_{\boldsymbol{\theta}}\mathcal{L}(\boldsymbol{\theta}, \mathbf{x}, y)\right). \tag{29}$$

We expect the dynamical preconditioner in ADAM also simplifies to $\mathbf{D}$ in high-dimensions; for details, see Appendix F.

As $\overline{\mathbf{K}}$ appears naturally in (8), its spectrum regulates the rate of convergence of the optimization to stationarity. By utilizing our $d$-systems of ODEs we can establish the following convergence rate:

**Theorem 4.** *Assume* $\epsilon \sim N(0, v^2)$ *and let* $s_i$ *be the stationary points to* (13a). *Then there is an absolute constant* $c > 0$ *so that if*

$$\eta \frac{\mathrm{Tr}(\overline{\mathbf{K}})}{2d} \leq \min\left\{c, \frac{4v}{\pi}\right\}, \quad \text{and} \quad R_0 \leq cv, \tag{30}$$

*then we have, setting* $R_\infty = \sum_{i=1}^d s_i$ *to be the limit risk,*

$$|R_t - R_\infty| \leq 2(R_0 + R_\infty)e^{-t\eta\lambda_{\min}(\overline{\mathbf{K}})/(\pi v)}. \tag{31}$$

The proof is given in Appendix C. In contrast to vanilla SGD, where the risk converges (in a high-dimensional setting) with rate $\frac{1}{\bar{\kappa}}$, where $\bar{\kappa}(\mathbf{K}) = \frac{\mathrm{Tr}(\mathbf{K})}{d\lambda_{\min}(\mathbf{K})}$ is the average condition number (Paquette et al., 2022b). Theorem 4 states that the risk of SIGNSGD converges at a rate $\frac{\mathrm{Tr}(\overline{\mathbf{K}})}{d\lambda_{\min}(\overline{\mathbf{K}})}$, after selecting the largest allowed $\eta$. SIGNSGD is therefore favored over SGD when $\bar{\kappa}(\overline{\mathbf{K}}) < \bar{\kappa}(\mathbf{K})$. See Figure 9 for experimental validation.

**Settings in which the preconditioned $\overline{\mathbf{K}}$ is preferable.** Theorem 4 shows that the rate of convergence is governed entirely by $\overline{\mathbf{K}}$. The clearest setting when this is favourable is if $\mathbf{K}$ is diagonal, so that $\overline{\mathbf{K}} = \sqrt{\mathbf{K}}$. In this case, the convergence rate is, up to constants

$$\frac{1}{d}\frac{\mathrm{Tr}(\overline{\mathbf{K}})}{\lambda_{\min}(\overline{\mathbf{K}})} = \frac{1}{d}\frac{\mathrm{Tr}(\overline{\mathbf{K}})}{\sqrt{\lambda_{\min}(\mathbf{K})}} \leq \frac{\sqrt{\frac{1}{d}\mathrm{Tr}(\mathbf{K})}}{\sqrt{\lambda_{\min}(\mathbf{K})}}. \qquad (32)$$

Hence on diagonal problems, SIGNSGD attains a speedup over SGD commensurate to the speedup of optimal deterministic convex optimization algorithms such as Conjugate gradient over gradient descent (Nocedal & Wright, 2006).

Diagonally dominant matrices should see similar benefits — consistent with prior work showing that SIGNSGD is effective when the the Hessian of the risk ($\mathbf{K}$ in our setting) is sufficiently diagonally concentrated (Balles et al., 2020).

A second situation in which one may have substantial speedups are for block-diagonal $\mathbf{K}$, where the blocks are scaled by greatly differing constants; diagonal preconditioning by $\mathbf{D}$ partially corrects for this effect. It has been argued that one of the principal advantages of ADAM is that it correctly adapts learning rates across different layers of Transformers and MLPs (Zhang et al., 2024), which can have similar structures in their Jacobians.

**Settings in which $\overline{\mathbf{K}}$ does not help.** Like preconditioning generally, $\overline{\mathbf{K}}$ does not always have a smaller condition number than $\mathbf{K}$. See Appendix H for a counter example.

In addition, if the eigenvectors of $\mathbf{K}$ are randomized to make a new covariance matrix $\mathbf{A}$, say by performing a uniformly random orthogonal change of basis, the entries of the diagonal of $\mathbf{A}$ will concentrate to be

$$\max_i \left| \mathbf{A}_{ii} - \frac{\mathrm{Tr}(\mathbf{K})}{d} \right| = O((\log d)d^{-1/2}), \qquad (33)$$

and so the preconditioner $\mathrm{diag}(\mathbf{A})^{-1/2}$ does not affect the condition number of $\mathbf{A}$. Thus, the benefit of diagonal preconditioning is strongly tied to special properties of the basis in which the optimization is performed (Nocedal & Wright, 2006).

### 4.4. Gradient noise reshaping

Finally, there is gradient noise reshaping, wherein the SGD gradient noise matrix $\mathbf{K}$ is replaced by the matrix $\mathbf{K}_\sigma$ up to constants. This is a complicated mapping; there is no short answer about the impact of this effect. In some cases passing from $\mathbf{K} \to \mathbf{K}_\sigma$ might affect the magnitudes of the eigenvalues but not their structure; see Figure 4 for an example on CIFAR10.

In the case that $\mathbf{K}$ itself is a sample covariance matrix, the matrix $\mathbf{K}_\sigma$ is strongly related to a *kernel inner product matrix*, for which there is a large literature. This includes properties of bulk spectra (Karoui, 2010; Cheng & Singer, 2013), norms (Fan & Montanari, 2019) and more.

When $\mathbf{K}$ is a diagonal matrix then $\mathbf{K}_\sigma = \frac{\pi}{2}\mathbf{I}$ and so this can be considered a type of preconditioning of the gradient noise, albeit with a *more* aggressive preconditioner than $\mathbf{D}$.

We *expect* that for $\mathbf{K}$ with power-law spectra, often seen in practice in vision (e.g. in Figure 4) and language embeddings, $\mathbf{K}_\sigma$ again has powerlaw spectra of the same exponent. Beyond the spectral distribution, replacing $\mathbf{K}$ by $\mathbf{K}_\sigma$ may also serve to slightly break the alignment of large directions of gradient variance from large gradient biases (they are perfectly aligned in SGD), which should be beneficial both to stability of the algorithm and performance.

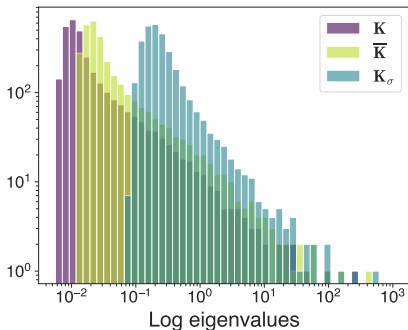

Figure 4: Log eigenvalues of $\mathbf{K}, \overline{\mathbf{K}}, \mathbf{K}_\sigma$ computed for the CIFAR10 dataset.

## 5. Minibatching

Although our main theorem is established for a batch size of 1, we can numerically validate SIGNHSGD at larger batch sizes $b > 1$ (Figure 5). SIGNHSGD with $b > 1$ exhibits all of the same core properties as the single-pass case, i.e. preconditioning, $\epsilon$-compression and gradient reshaping. The two main differences between $b = 1$ and $b > 1$ can be seen in scaling factor $N_b$ (see Remark 1) and $\epsilon$-compression $\varphi^{(b)}$ found in the drift term of SIGNHSGD. When $b > 1$, the noise compression factor $\varphi^{(b)}$ becomes the expectation of a function of the risk across a distribution that differs from

the original noise. This can have significant effects on the noise compared to single-pass. Indeed, when $b > 1$, $\omega_i$ as defined in (9) is not identically 1, allowing $\epsilon_b$ to gain some regularity over $\epsilon^{(i)}$. Which is to say that $\varphi^{(b)}$ may have improve decay rate of no-worse than $O(\sqrt{\mathcal{R}})$ when $\mathcal{R}$ approaches 0. This due to the addition of a density $\omega_i$. A straightforward application of the Lebesgue differentiation Theorem reveals that $\varphi^{(b)}(\mathcal{R})/\sqrt{\mathcal{R}}$ can be bounded from below by $Cf(0)$ where $C$ is an absolute constant and $f$ denotes the respective density, in the small risk regime. For example, in the Radamacher case, in which $\varphi^{(1)}$ can be very badly behaved at small risk, $\epsilon_b$ simply becomes the law of an entry of a spherical vector, which is a good unimodular, bounded density. In addition, as batch grows, the $\varphi$ will be pushed into the Gaussian regime due to the central limit theorem, provided that $\epsilon^{(i)}$ has a second moment.

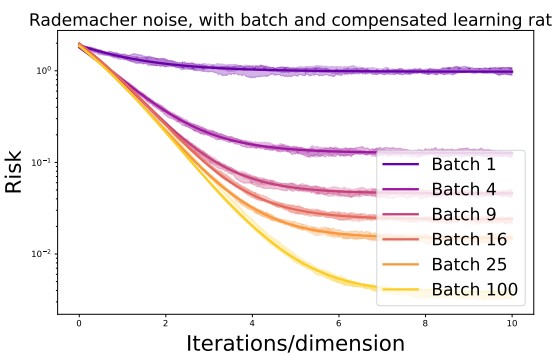

Figure 5: Minibatch-SIGNSGD versus theory. SIGNHSGD and SIGNODE adapt to this case, as in (8). LR rescaled by inverse of square-root-batch (more specifically by $1/N_r$) to illustrate square-root scaling.

## 6. Discussion

Our high-dimensional limit sheds a quantitative light on the precise ways in which SIGNSGD can be compared to SGD, via change of effective learning rate, noise compression, preconditioning, and reshaping of the gradient noise.

Theorem 2, the main technical contribution of this work, required substantial technical efforts. Although similar in formulation to existing work like Collins-Woodfin et al. (2024), there are technical complexities in working with the nonsmooth $\sigma$ function: both in terms of deriving the relevant concentration of measure estimates (the textbook versions of which require smoothness) and in terms of the additional pathology of the resulting SIGNHSGD (especially the $\varphi$). We believe that a version of Theorem 2 is true in much greater generality than we have proven it, even for the linear setting: two desirable mathematical generalizations are quantifying dimension *intrinsically* (instead of through the ambient dimension) and generalizing the theory to settings

of non-Gaussian data. We note that our analysis is in a qualitatively different regime than the concurrent work of Compagnoni et al. (2025); see Appendix G for more details.

More detailed analysis of the effects of non-diagonal covariance is a key future direction. There is an active area of research on the benefits of *non-diagonal* preconditioning methods as an alternative to Adam variants, and studying the non-diagonal case in SIGNSGD may provide more insight into cases where the diagonal preconditioner is insufficient to improve optimization. Our SDEs and ODEs provide the tools to analyze such settings.

Though our work focuses on the case of MSE loss and linear regression, there is a path towards extending results to more general settings using recent results in high-dimensional optimization (Collins-Woodfin et al., 2023). In practical settings, models undergo dramatic changes in local geometry during training; nonetheless, stability analysis of the linearized problem is still useful for understanding aspects of the non-linear dynamics of these systems (Cohen et al., 2022; Agarwala & Pennington, 2024).

Finally, our analysis of SIGNSGD gives hints towards understanding ADAM in a similar setting. A heuristic analysis shows that ADAM has a homogenized process similar to SIGNHSGD: it appears to share the preconditioner $\mathbf{D}$ while differing from SIGNHSGD in its noise compression and gradient noise effects $\mathbf{K}_\sigma$ (Appendix F). For well-behaved noises $\epsilon$, SIGNSGD should be nearly path-identical to ADAM; we note that LION has been recently observed to do just that (Zhao et al., 2024). We leave investigation of ADAM for future work.

## Impact Statement

This paper presents work whose goal is to advance the field of Machine Learning. There are many potential societal consequences of our work, none which we feel must be specifically highlighted here.

## Acknowledgments

We would like to thank Courtney Paquette and Lechao Xiao for taking the time to proofread and provide feedback throughout the writing of this paper. We would also like to thank anonymous referees for suggesting the natural extension to minibatching.

Research by E. Paquette was supported by a Google Grant and a Discovery Grant from the Natural Science and Engineering Research Council (NSERC) of Canada.

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

## Overview of supplementary material

The supplementary material is primarily dedicated to the proofs of the main theorems, Theorem 1 and 2. Here we give the organization of the appendices.

In Appendix A, we give the proof of these main theorems, including their extensions in Theorem 5 and 6. The key approximations to the update rules of SIGNSGD are given in Appendix A.1, including the key technical Lemma 3. In Appendix A.2, we show how these tools are used to give the main proof (but we defer the estimates on the stochastic errors to Appendix A.4), culminating in Lemma 7, which in fact proves the main theorem statement (that of Theorem 5). In Appendix A.3, we discuss the extension Theorem 6 – as this is a modification of the proof of Theorem 5, we do not go into details.

In Appendix B, we give the derivation of the ODEs SIGNODE and VANILLAODE from their homogenized counterparts, and discuss the proof of Theorem 2, which follows the same strategy as Theorem 5 (for full details of this type of ODE comparison, see (Collins-Woodfin et al., 2023)). Here also, we discuss the derivation of the VANILLAODE, which is a special case of (Collins-Woodfin et al., 2023).

In Appendix C, we proof the analysis of the SIGNODE and VANILLAODE that gives its limit level (Theorem 3) and a local convergence rate (Theorem 4).

In Appendix E, we provide additional supporting simulations, corroborating aspects of the main theorems.

In Appendix F, we give a heuristic derivation of the high-dimensional limit of ADAM.

In Appendix G we show the "Weak Approximation" theory of ADAM produces a different SDE prediction (see the discussion there as well).

In Appendix H, we give an example of a matrix where diagonal preconditioning hurts.

Finally in Appendix I, we give some additional information on how the experiments were performed.

# A. Proof of Main Theorem

## A.1. Approximation of the conditional updates

For simplicity of our proofs, we will assume $\eta$ is constant as well as batch-size is $b = 1$. The proof remains unchanged if $\eta$ is defined as in Assumption 4 and if $b > 1$. For the convenience of the reader and to avoid confusion, we provide the typical notions of convergence in high-dimensions.

**Definition 3.** *An event $A \subset \mathbb{R}^d$ holds with high-probability, if there exists some $\delta > 0$ independent to $d$ such that $\mathbb{P}(A) \geq 1 - Cd^{-\delta}$ for some $C$ independent to $d$.*

**Definition 4.** *An event $A \subset \mathbb{R}^d$ holds with overwhelming-probability, if for all $\delta > 0$ there exists $C_\delta$ such that $\mathbb{P}(A) \geq 1 - C_\delta d^{-\delta}$.*

Denote $\mathcal{F}_k$ to be the natural filtration generated by the data $\{\mathbf{x}_j\}_{j=1}^k$ and the label noise $\{\epsilon_j\}_{j=1}^k$. For notational convenience, we define the "centered" SIGNSGD iterate $\boldsymbol{\nu}_k = \boldsymbol{\theta}_k - \boldsymbol{\theta}_*$. We also denote $\mathcal{R}_k = \mathcal{R}(\boldsymbol{\theta}_k)$ when it is clear.

Notice now, that the $k + 1$th update of SIGNSGD is given by

$$\boldsymbol{\nu}_{k+1} - \boldsymbol{\nu}_k = -\frac{\eta}{d}\sigma(\mathbf{x}_{k+1})\sigma(\langle \mathbf{x}_{k+1}, \boldsymbol{\nu}_k \rangle - \epsilon_{k+1}). \tag{34}$$

A key component of our proof aims to compute the mean of this update (conditioned on $\mathcal{F}_k$) and then simplify this mean by introducing errors which vanish in high-dimensions.

**Lemma 1.** *Conditional on $\mathcal{F}_k$, the mean of the $i$-th element of (34) is given by*

$$\mathbb{E}\left[\boldsymbol{\nu}_{k+1}^i - \boldsymbol{\nu}_k^i | \mathcal{F}_k\right] = -\frac{\eta}{d}\sqrt{\frac{2}{\pi}}2h_k^i(0)\left\langle \overline{\mathbf{K}}_i, \boldsymbol{\nu}_k \right\rangle - \frac{\eta}{d}\mathbb{E}[\sigma(\mathbf{x}_{k+1}^i)R_{k+1}^i], \tag{35}$$

*where, denoting $\mathcal{L}_\epsilon$ as the law of the noise,*

$$h_k^i(x) = \frac{1}{\sqrt{2\pi\left(2\mathcal{R}_k - \left\langle \overline{\mathbf{K}}_i, \boldsymbol{\nu}_k \right\rangle^2\right)}}\int_{\mathbb{R}}\exp\left(\frac{-(x+y)^2}{2\left(2\mathcal{R}_k - \left\langle \overline{\mathbf{K}}_i, \boldsymbol{\nu}_k \right\rangle^2\right)}\right)\,d\mathcal{L}_\epsilon(y), \tag{36}$$

*and*

$$R_{k+1}^i = O\left(\left(\frac{\langle \mathbf{K}_i, \boldsymbol{\nu}_k \rangle}{\mathbf{K}_{ii}}\mathbf{x}_{k+1}^i\right)^3\right). \tag{37}$$

*Proof.* Following the update rule (1), we start by computing the conditional update of the $i$-th entry of the iterates

$$\mathbb{E}\left[\boldsymbol{\nu}_{k+1}^i - \boldsymbol{\nu}_k^i | \mathcal{F}_k\right] = -\frac{\eta}{d}\mathbb{E}\left[\sigma(\mathbf{x}_{k+1}^i)\sigma(\langle \mathbf{x}_{k+1}, \boldsymbol{\nu}_k \rangle - \epsilon_{k+1})|\mathcal{F}_k\right] \tag{38}$$

$$= -\frac{\eta}{d}\mathbb{E}\left[\sigma(\mathbf{x}_{k+1}^i)\mathbb{E}\left[\sigma\left(\boldsymbol{\nu}_k^i\mathbf{x}_{k+1}^i + \sum_{j\neq i}\boldsymbol{\nu}_k^j\mathbf{x}_{k+1}^j - \epsilon_{k+1}\right)\bigg|\mathcal{F}_k, \mathbf{x}_{k+1}^i\right]\bigg|\mathcal{F}_k\right]. \tag{39}$$

Given that the data is Gaussian distributed, upon conditioning on $\mathcal{F}_k$ we see that

$$\sum_{j\neq i}\boldsymbol{\nu}_k^j\mathbf{x}_{k+1}^j \sim N(0, 2\mathcal{R}_k - 2\boldsymbol{\nu}_k^i\langle \mathbf{K}_i, \boldsymbol{\nu}_k \rangle + \mathbf{K}_{ii}(\boldsymbol{\nu}_k^i)^2). \tag{40}$$

Additionally, for $c_i$ is any constant, we can write

$$\sum_{j\neq i}\boldsymbol{\nu}_k^j\mathbf{x}_{k+1}^j = \left(\sum_{j\neq i}\boldsymbol{\nu}_k^j\mathbf{x}_{k+1}^j - c_i\boldsymbol{\nu}_k^i\mathbf{x}_{k+1}^i\right) + c_i\boldsymbol{\nu}_k^i\mathbf{x}_{k+1}^i. \tag{41}$$

Let $y^i = \sum_{j \neq i} \boldsymbol{\nu}_k^j \mathbf{x}_{k+1}^j - c_i \boldsymbol{\nu}_k^i \mathbf{x}_{k+1}^i$. Choosing $c_i = \frac{\langle \mathbf{K}_i, \boldsymbol{\nu}_k \rangle - \mathbf{K}_{ii} \boldsymbol{\nu}_k^i}{\mathbf{K}_{ii} \boldsymbol{\nu}_k^i}$, makes $y^i$ uncorrelated to $\mathbf{x}_{k+1}^i$ and hence independent.
Additionally, notice $y^i \sim N(0, 2\mathcal{R}_k - \langle \overline{\mathbf{K}}_i, \boldsymbol{\nu}_k \rangle^2)$. Moreover, since $y^i$ is independent to $\epsilon_{k+1}$ their difference $y_i - \epsilon_{k+1}$ has density given by

$$h_k^i(x) = \frac{1}{\sqrt{2\pi \left( 2\mathcal{R}_k - \langle \overline{\mathbf{K}}_i, \boldsymbol{\nu}_k \rangle^2 \right)}} \int_{\mathbb{R}} \exp \left( \frac{-(x+y)^2}{2 \left( 2\mathcal{R}_k - \langle \overline{\mathbf{K}}_i, \boldsymbol{\nu}_k \rangle^2 \right)} \right) \, \mathrm{d}\mathcal{L}_\epsilon(y). \tag{42}$$

Using (42), we compute

$$\mathbb{E} \left[ \sigma \left( \boldsymbol{\nu}_k^i \mathbf{x}_{k+1}^i + \sum_{j \neq i} \boldsymbol{\nu}_k^j \mathbf{x}_{k+1}^j - \epsilon_{k+1} \right) \bigg| \mathcal{F}_k, \mathbf{x}_{k+1}^i \right]$$
$$= \mathbb{E} \left[ \sigma \left( y^i - \epsilon_{k+1} + (1 + c_i) \boldsymbol{\nu}_k^i \mathbf{x}_{k+1}^i \right) \bigg| \mathcal{F}_k, \mathbf{x}_{k+1}^i \right]$$
$$= \mathbb{P}_{\mathcal{F}_k, \mathbf{x}_{k+1}^i} \left( y_i - \epsilon_{k+1} > -(1 + c_i) \boldsymbol{\nu}_k^i \mathbf{x}_{k+1}^i \right) - \mathbb{P}_{\mathcal{F}_k, \mathbf{x}_{k+1}^i} \left( y_i - \epsilon_{k+1} \leq -(1 + c_i) \boldsymbol{\nu}_k^i \mathbf{x}_{k+1}^i \right)$$
$$= \int_{-(1+c_i) \boldsymbol{\nu}_k^i \mathbf{x}_{k+1}^i}^{\infty} h_k^i(x) \, \mathrm{d}x - \int_{-\infty}^{-(1+c_i) \boldsymbol{\nu}_k^i \mathbf{x}_{k+1}^i} h_k^i(x) \, \mathrm{d}x. \tag{43}$$

Define

$$H(s) = \int_{-s}^{\infty} h_k^i(x) \, \mathrm{d}x - \int_{-\infty}^{-s} h_k^i(x) \, \mathrm{d}x, \tag{44}$$

where upon differentiating it is easy to see that

$$H'(s) = 2h_k^i(-s). \tag{45}$$

Taylor expanding around 0 we obtain,

$$\mathbb{E} \left[ \sigma \left( \boldsymbol{\nu}_k^i \mathbf{x}_{k+1}^i + \sum_{j \neq i} \boldsymbol{\nu}_k^j \mathbf{x}_{k+1}^j - \epsilon_{k+1} \right) \bigg| \mathcal{F}_k, \mathbf{x}_{k+1}^i \right] = H((1 + c_i) \boldsymbol{\nu}_k^i \mathbf{x}_{k+1}^i)$$
$$= H(0) + 2h_k^i(0) \frac{\langle \mathbf{K}_i, \boldsymbol{\nu}_k \rangle}{\mathbf{K}_{ii}} \mathbf{x}_{k+1}^i$$
$$+ \frac{\mathrm{d}}{\mathrm{d}s} h_k^i(0) \left( \frac{\langle \mathbf{K}_i, \boldsymbol{\nu}_k \rangle}{\mathbf{K}_{ii}} \mathbf{x}_{k+1}^i \right)^2 + R_{k+1}^i, \tag{46}$$

where

$$R_{k+1}^i = O \left( \left( \frac{\langle \mathbf{K}_i, \boldsymbol{\nu}_k \rangle}{\mathbf{K}_{ii}} \mathbf{x}_{k+1}^i \right)^3 \right). \tag{47}$$

Plugging this back into (39) yields

$$\mathbb{E} \left[ \boldsymbol{\nu}_{k+1}^i - \boldsymbol{\nu}_k^i | \mathcal{F}_k \right] = -\frac{\eta}{d} \mathbb{E} \left[ \sigma(\mathbf{x}_{k+1}^i) \left( H(0) + 2h_k^i(0) \frac{\langle \mathbf{K}_i, \boldsymbol{\nu}_k \rangle}{\mathbf{K}_{ii}} \mathbf{x}_{k+1}^i \right) \bigg| \mathcal{F}_k \right]$$
$$- \frac{\eta}{d} \mathbb{E} \left[ \sigma(\mathbf{x}_{k+1}^i) \left( \frac{\mathrm{d}}{\mathrm{d}s} h_k^i(0) \left( \frac{\langle \mathbf{K}_i, \boldsymbol{\nu}_k \rangle}{\mathbf{K}_{ii}} \mathbf{x}_{k+1}^i \right)^2 + R_{k+1}^i \right) \bigg| \mathcal{F}_k \right]$$
$$= -\frac{\eta}{d} 2h_k^i(0) \frac{\langle \mathbf{K}_i, \boldsymbol{\nu}_k \rangle}{\mathbf{K}_{ii}} \mathbb{E}|\mathbf{x}_{k+1}^i| - \frac{\eta}{d} \mathbb{E}[\sigma(\mathbf{x}_{k+1}^i) R_{k+1}^i]$$
$$= -\frac{\eta}{d} 2h_k^i(0) \langle \overline{\mathbf{K}}_i, \boldsymbol{\nu}_k \rangle \sqrt{\frac{2}{\pi}} - \frac{\eta}{d} \mathbb{E}[\sigma(\mathbf{x}_{k+1}^i) R_{k+1}^i]. \tag{48}$$

$\square$

In the next two lemmas we will show that for all $1 \leq i \leq d$, the risk dependent factors $2\sqrt{\frac{2}{\pi}}h_k^i(0)$ can be well-approximated by $\frac{1}{\sqrt{2\mathcal{R}_k}}\varphi(\mathcal{R}_k)$, where

$$\varphi(\mathcal{R}_k) = \frac{2}{\pi}\int_{\mathbb{R}}\exp\left(\frac{-y^2}{4\mathcal{R}_k}\right)\,\mathrm{d}\mathcal{L}_\epsilon(y). \tag{49}$$

To do this, we will show that: $\langle \overline{\mathbf{K}}_i, \boldsymbol{\nu}_k \rangle = O(d^{-s})$ for some $s > 0$. Additionally, this would also imply the error $R_{k+1}^i$ vanishes as $d \to \infty$. This simplifies (35) by removing the latter error term and reducing each $h_k^i$ into a single constant factor, i.e.

$$\mathbb{E}\left[\boldsymbol{\nu}_{k+1}^i - \boldsymbol{\nu}_k^i \Big| \mathcal{F}_k\right] \approx -\frac{\eta}{d}\frac{\varphi(\mathcal{R}_k)}{\sqrt{2\mathcal{R}_k}}\langle \overline{\mathbf{K}}_i, \boldsymbol{\nu}_k \rangle. \tag{50}$$

Our proof makes use of the *resolvent* $\mathbf{R}(z; \mathbf{K}) = (\mathbf{K} - z\mathbf{I})^{-1}$, a matrix valued function essentially encoding powers of $\mathbf{K}$. The following lemma will allows us to to control $\langle \overline{\mathbf{K}}_i, \boldsymbol{\nu}_k \rangle$ by a finite net of resolvents.

**Lemma 2.** *There exists a net $\Gamma_0 \subset \Gamma$ of order $O(d)$ and $C(\overline{\mathbf{K}}) > 0$ such that for all $k$ and $1 \leq i \leq d$,*

$$|\langle \overline{\mathbf{K}}_i, \boldsymbol{\nu}_k \rangle| \leq C_{\overline{\mathbf{K}}} \max_{z \in \Gamma_0} \max_{1 \leq i \leq d}\left|\mathbf{R}(z; \overline{\mathbf{K}})_i^{\mathrm{T}}\boldsymbol{\nu}_k\right|. \tag{51}$$

*Proof.* It is easy to check by the Cauchy's integral formula that

$$\langle \overline{\mathbf{K}}_i, \boldsymbol{\nu}_k \rangle = -\frac{1}{2\pi i}\oint_\Gamma z\mathbf{R}(z; \overline{\mathbf{K}})_i^{\mathrm{T}}\boldsymbol{\nu}_k\,\mathrm{d}z. \tag{52}$$

By Assumption 3, we may bound $\left|\mathbf{R}(z; \overline{\mathbf{K}})_i\right|$ for all $z \in \Gamma$ by a finite collection of $z_0 \in \Gamma$. Indeed, if $\Gamma_0$ is a $1/\sqrt{d}$-net on $\Gamma$ then $|\Gamma_0| = O(d)$. It follows that for all $z \in \Gamma$, there exists some $z_0 \in \Gamma_0$ such that $|z - z_0| \leq 1/\sqrt{d}$. Then, by resolvent identities we see that for all $1 \leq i \leq d$ and $\mathbf{a} \in \mathbb{R}^d$,

$$\begin{aligned}
\left|\mathbf{R}(z; \overline{\mathbf{K}})_i^{\mathrm{T}}\mathbf{a}\right| &= \left|\mathbf{R}(z_0; \overline{\mathbf{K}})_i^{\mathrm{T}}\mathbf{a} + (z - z_0)[\mathbf{R}(z; \overline{\mathbf{K}})\mathbf{R}(z_0; \overline{\mathbf{K}})]_i^{\mathrm{T}}\mathbf{a}\right| \\
&\leq \left|\mathbf{R}(z_0; \overline{\mathbf{K}})_i^{\mathrm{T}}\mathbf{a}\right| + \frac{1}{\sqrt{d}}\left\|\mathbf{R}(z; \overline{\mathbf{K}})_i\right\|\left\|\mathbf{R}(z_0; \overline{\mathbf{K}})\mathbf{a}\right\| \\
&\leq (1 + M_R)\max_{1 \leq i \leq d}\left|\mathbf{R}(z_0; \overline{\mathbf{K}})_i^{\mathrm{T}}\mathbf{a}\right|.
\end{aligned} \tag{53}$$

In particular,

$$\max_{z \in \Gamma}\max_{1 \leq i \leq d}\left|\mathbf{R}(z; \overline{\mathbf{K}})_i^{\mathrm{T}}\mathbf{a}\right| \leq (1 + M_R)\max_{z_0 \in \Gamma_0}\max_{1 \leq i \leq d}\left|\mathbf{R}(z_0; \overline{\mathbf{K}})_i^{\mathrm{T}}\mathbf{a}\right|. \tag{54}$$

Plugging this into (52),

$$\begin{aligned}
\left|\langle \overline{\mathbf{K}}_i, \boldsymbol{\nu}_k \rangle\right| &\leq \frac{1}{2\pi}\oint_\Gamma |z|(1 + M_R)\max_{\mathbf{z}_0 \in \Gamma_0}\max_{1 \leq i \leq d}\left|\mathbf{R}(z_0; \overline{\mathbf{K}})_i^{\mathrm{T}}\boldsymbol{\nu}_k\right|\,\mathrm{d}z \\
&= 4(1 + M_R)\left\|\overline{\mathbf{K}}\right\|^2\max_{z \in \Gamma_0}\max_{1 \leq i \leq d}\left|\mathbf{R}(z; \overline{\mathbf{K}})_i^{\mathrm{T}}\boldsymbol{\nu}_k\right| \\
&= C_{\overline{\mathbf{K}}}\max_{z \in \Gamma_0}\max_{1 \leq i \leq d}\left|\mathbf{R}(z; \overline{\mathbf{K}})_i^{\mathrm{T}}\boldsymbol{\nu}_k\right|.
\end{aligned} \tag{55}$$

$\square$

Note that terms such as $\eta$ and $\left\|\overline{\mathbf{K}}\right\|$ are bounded by assumption, thus we make the convention moving forward any constants independent to $d$ such as $C_{\overline{\mathbf{K}}}$ may change from line to line. Therefore, to show that $\langle \overline{\mathbf{K}}_i, \mathbf{v}_k \rangle$ shrinks as $d \to \infty$, it suffices to show that $\max_{z \in \Gamma_0}\max_{1 \leq i \leq d}\left|\mathbf{R}(z; \overline{\mathbf{K}})_i^{\mathrm{T}}\mathbf{v}_k\right|$ shrinks as $d \to \infty$.

Before we do so, it will be convenient to work under the setting that the risk is bounded. As such, let $L > 0$ and define the following stopping time,

$$\tau_0 = \min\{k; \|\boldsymbol{\nu}_k\| > L\}, \tag{56}$$

as well as the stopped process $\mathbf{v}_k = \boldsymbol{\nu}_{k \wedge \tau_0}$. We show in Lemma 8 that $L$ may be chosen so that $\mathbf{v}_k = \boldsymbol{\nu}_k$ with overwhelming probability, effectively removing the bounded constraint.

**Lemma 3.** *Given Assumptions 1 - 5, there exists a net $\Gamma_0 \subset \Gamma$ of order $O(d)$, such that for all $t > 0$ and $1/6 + \delta_0 < \delta < 1/4$,*

$$\max_{z \in \Gamma_0} \max_{1 \leq i \leq d} \max_{0 \leq k \leq \lfloor td \rfloor} \left| \mathbf{R}(z; \overline{\mathbf{K}})_i^{\mathrm{T}} \mathbf{v}_k \right| < \frac{d^\delta}{\sqrt{d}} \tag{57}$$

*with high-probability.*

*Proof.* For clarity of notation, let $\widetilde{h}_k^i = 2\sqrt{\frac{2}{\pi}} h_k^i(0)$ and $\widetilde{h}_k = \frac{1}{\sqrt{2\mathcal{R}_k}} \varphi(\mathcal{R}_k)$. In addition, define $\mathbf{A}_k$ be a diagonal matrix with entries given by $\widetilde{h}_k^i$, as well as the vector $\mathbf{E}_{k+1} = (\mathbb{E}[\sigma(\mathbf{x}_{k+1}^i) R_{k+1}^i])_{i=1}^d$. By (35), for a fixed $z \in \Gamma_0$ and $1 \leq i \leq d$,

$$\begin{aligned}
\mathbb{E}\left[\mathbf{R}(z; \overline{\mathbf{K}})_i^{\mathrm{T}}(\mathbf{v}_{k+1} - \mathbf{v}_k)|\mathcal{F}_k\right] &= \mathbf{R}(z; \overline{\mathbf{K}})_i^{\mathrm{T}} \mathbb{E}\left[\mathbf{v}_{k+1} - \mathbf{v}_k|\mathcal{F}_k\right] \\
&= -\frac{\eta}{d} \mathbf{R}(z; \overline{\mathbf{K}})_i^{\mathrm{T}} \left(\mathbf{A}_k \overline{\mathbf{K}} \mathbf{v}_k + \mathbf{E}_{k+1}\right) \\
&= -\frac{\eta}{d} \widetilde{h}_k \mathbf{R}(z; \overline{\mathbf{K}})_i^{\mathrm{T}} \overline{\mathbf{K}} \mathbf{v}_k \\
&\quad - \frac{\eta}{d} \mathbf{R}(z; \overline{\mathbf{K}})_i^{\mathrm{T}} \left(\mathbf{A}_k \overline{\mathbf{K}} \mathbf{v}_k - \widetilde{h}_k \overline{\mathbf{K}} \mathbf{v}_k + \mathbf{E}_{k+1}\right) \\
&= -\frac{\eta}{d} \widetilde{h}_k \left(z \mathbf{R}(z; \overline{\mathbf{K}})_i^{\mathrm{T}} \mathbf{v}_k + \mathbf{v}_k^i\right) \\
&\quad - \frac{\eta}{d} \mathbf{R}(z; \overline{\mathbf{K}})_i^{\mathrm{T}} \left(\mathbf{A}_k \overline{\mathbf{K}} \mathbf{v}_k - \widetilde{h}_k \overline{\mathbf{K}} \mathbf{v}_k + \mathbf{E}_{k+1}\right) \\
&= -\frac{\eta}{d} \widetilde{h}_k z \mathbf{R}(z; \overline{\mathbf{K}})_i^{\mathrm{T}} \mathbf{v}_k \\
&\quad + \underbrace{\frac{\eta}{d} \left(-\widetilde{h}_k \mathbf{v}_k^i + \mathbf{R}(z; \overline{\mathbf{K}})_i^{\mathrm{T}} \left(\widetilde{h}_k \overline{\mathbf{K}} \mathbf{v}_k - \mathbf{A}_k \overline{\mathbf{K}} \mathbf{v}_k - \mathbf{E}_{k+1}\right)\right)}_{:= \mathcal{E}_k^i(z)}. 
\end{aligned} \tag{58}$$

By the Doob decomposition we see that,

$$\begin{aligned}
\mathbf{R}(z; \overline{\mathbf{K}})_i^{\mathrm{T}} \mathbf{v}_{k+1} &= \mathbf{R}(z; \overline{\mathbf{K}})_i^{\mathrm{T}} \mathbf{v}_k + \mathbb{E}\left[\mathbf{R}(z; \overline{\mathbf{K}})_i^{\mathrm{T}}(\mathbf{v}_{k+1} - \mathbf{v}_k)|\mathcal{F}_k\right] + \Delta M_{k+1}^i(z) \\
&= \left(1 - \frac{\eta}{d} \widetilde{h}_k z\right) \mathbf{R}(z; \overline{\mathbf{K}})_i^{\mathrm{T}} \mathbf{v}_k + \mathcal{E}_k^i(z) + \Delta M_{k+1}^i(z),
\end{aligned} \tag{59}$$

where $\Delta M_{k+1}^i(z)$ are the martingale increments of $\mathbf{R}(z; \overline{\mathbf{K}})_i^{\mathrm{T}}(\mathbf{v}_{k+1} - \mathbf{v}_k)$. Let

$$L_k = \prod_{j=0}^k \left(1 - \frac{\eta}{d} \widetilde{h}_j z\right),$$

then upon iterating (59) we obtain

$$\mathbf{R}(z; \overline{\mathbf{K}})_i^{\mathrm{T}} \mathbf{v}_{k+1} = L_k \mathbf{R}(z; \overline{\mathbf{K}})_i^{\mathrm{T}} \mathbf{v}_0 + L_k \sum_{j=0}^k \frac{1}{L_j} \left(\mathcal{E}_j^i(z) + \Delta M_{j+1}^i(z)\right). \tag{60}$$

It is easy to check that $\sum_{j=0}^k \frac{1}{L_j} \Delta M_{j+1}^i(z)$ is a martingale so we shall denote it by $\overline{\mathcal{M}}_{k+1}^i(z)$. Let

$$\tau_1 = \min\left\{k; \quad \left|\mathbf{R}(z, \overline{\mathbf{K}})_i^{\mathrm{T}} \mathbf{v}_k\right| \geq \frac{d^\delta}{\sqrt{d}} \text{ for some } 1 \leq i \leq d \text{ and } z \in \Gamma_0\right\}. \tag{61}$$

It suffices to show (57) holds for the stopped process $\mathbf{v}_{k \wedge \tau_1}$ given that

$$\begin{aligned}
\mathbb{P}&\left(\max_{1 \leq k \leq \lfloor td \rfloor} \max_{z \in \Gamma_0} \max_{1 \leq i \leq d} \left|\mathbf{R}(z; \overline{\mathbf{K}})_i^{\mathrm{T}} \mathbf{v}_k\right| \geq \frac{d^\delta}{\sqrt{d}}\right) \\
&= \mathbb{P}\left(\max_{z \in \Gamma_0} \max_{1 \leq i \leq d} \left|\mathbf{R}(z; \overline{\mathbf{K}})_i^{\mathrm{T}} \mathbf{v}_{\lfloor td \rfloor \wedge \tau_1}\right| \geq \frac{d^\delta}{\sqrt{d}}\right).
\end{aligned} \tag{62}$$

For notational clarity, we will write $\widetilde{\mathbf{v}}_k = \mathbf{v}_{k \wedge \tau_1}$. Note that (60) holds all $k$, so it must also hold for the stopped process $\widetilde{\mathbf{v}}_k$. Given that the entries of $\widetilde{\mathbf{v}}_k$ move at increments of $\frac{\eta}{d}$, we observe the following bound on the stopped process,

$$
\begin{aligned}
\left| \mathbf{R}(z; \overline{\mathbf{K}})_i^{\mathrm{T}} \widetilde{\mathbf{v}}_k \right| &\leq \frac{d^\delta}{\sqrt{d}} + \frac{\eta M_R}{\sqrt{d}} \\
&\leq \frac{\eta M_R' d^\delta}{\sqrt{d}}.
\end{aligned} \tag{63}
$$

Moreover, by Lemma 12 we know that $\widetilde{h}_k \leq M_\epsilon$. This in turn implies $L_k$ is bounded from above and below. Indeed,

$$
\begin{aligned}
|L_k| &= \prod_{j=0}^k \left| 1 - \frac{\eta}{d} \widetilde{h}_k z \right| \\
&\leq \prod_{j=0}^k 1 + \frac{\eta M_\epsilon |z|}{d} \\
&\leq \left( 1 + \frac{2 \eta M_\epsilon \|\overline{\mathbf{K}}\|}{d} \right)^{\lfloor td \rfloor} \\
&\leq \exp\left( \eta C_{t, \epsilon, \overline{\mathbf{K}}} \right).
\end{aligned} \tag{64}
$$

Similarly for the lower bound,

$$
\begin{aligned}
|L_k| &\geq \prod_{j=0}^k 1 - \frac{\eta}{d} \widetilde{h}_k |z| \\
&\geq \left( 1 - \frac{2 \eta M_\epsilon \|\overline{\mathbf{K}}\|}{d} \right)^{\lfloor td \rfloor} \\
&\geq \exp\left( \frac{-\frac{2 \eta M_\epsilon \|\overline{\mathbf{K}}\|}{d} \lfloor td \rfloor}{1 - \frac{2 \eta M_\epsilon \|\overline{\mathbf{K}}\|}{d}} \right) \\
&\geq \exp\left( -\frac{2 \eta M_\epsilon \|\overline{\mathbf{K}}\|}{d} \lfloor td \rfloor \right) \\
&= \exp\left( -\eta C_{t, \epsilon, \overline{\mathbf{K}}} \right),
\end{aligned} \tag{65}
$$

provided that $\frac{\eta M_\epsilon \|\overline{\mathbf{K}}\|}{d} < \frac{1}{2}$. Therefore, up to a constant factor

$$
\left| \mathbf{R}(z; \overline{\mathbf{K}})_i^{\mathrm{T}} \widetilde{\mathbf{v}}_k \right| \leq C_{\eta, t, \epsilon, \overline{\mathbf{K}}} \left( \left| \mathbf{R}(z; \overline{\mathbf{K}})_i^{\mathrm{T}} \widetilde{\mathbf{v}}_0 \right| + \left| \overline{\mathcal{M}}_{k+1}^i(z) \right| + \sum_{j=0}^{k-1} \left| \mathcal{E}_j^i(z) \right| \right). \tag{66}
$$

We will now bound the error $\mathcal{E}_j^i(z)$. By (51), we already know that

$$
\left| \langle \overline{\mathbf{K}}_i, \widetilde{\mathbf{v}}_j \rangle \right| \leq C_{\overline{\mathbf{K}}} \max_{z \in \Gamma_0} \max_{1 \leq i \leq d} \left| \mathbf{R}(z; \overline{\mathbf{K}})_i^{\mathrm{T}} \widetilde{\mathbf{v}}_j \right|. \tag{67}
$$

Similarly,

$$
\left| \widetilde{\mathbf{v}}_j^i \right| \leq C_{\overline{\mathbf{K}}} \max_{z \in \Gamma_0} \max_{1 \leq i \leq d} \left| \mathbf{R}(z; \overline{\mathbf{K}})_i^{\mathrm{T}} \widetilde{\mathbf{v}}_j \right|. \tag{68}
$$

We also observe for all $1 \leq i \leq d$,

$$
\begin{aligned}
\mathbf{E}_{j+1}^i &= \mathbb{E}\left[\sigma(\mathbf{x}_{j+1}^i)R_{j+1}^i\right] \\
&\leq O\left(\mathbb{E}\left[\left|\frac{\langle \mathbf{K}_i, \widetilde{\mathbf{v}}_j\rangle \mathbf{x}_{j+1}^i}{\mathbf{K}_{ii}}\right|^3\right]\right) \\
&= O\left(|\langle \overline{\mathbf{K}}_i, \widetilde{\mathbf{v}}_j\rangle|^3\right) \\
&= O\left(\left(\max_{z\in\Gamma_0}\max_{1\leq i\leq d}\left|\mathbf{R}(z;\overline{\mathbf{K}})_i^{\mathrm{T}}\widetilde{\mathbf{v}}_j\right|\right)^3\right).
\end{aligned}
\tag{69}
$$

In particular, for some constant $C_{\overline{\mathbf{K}}} > 0$,

$$
\begin{aligned}
|\mathbf{E}_{j+1}| &= \sqrt{\sum_{i=1}^d \left(E_{j+1}^i\right)^2} \\
&\leq \sqrt{d}C_{\overline{\mathbf{K}}}\left(\max_{z\in\Gamma_0}\max_{1\leq i\leq d}\left|\mathbf{R}(z;\overline{\mathbf{K}})_i^{\mathrm{T}}\widetilde{\mathbf{v}}_j\right|\right)^3.
\end{aligned}
\tag{70}
$$

For our last error term we apply the Lipschitz bound obtained by Lemma 12. That is the map

$$
s \mapsto \psi(s) = \frac{2}{\pi\sqrt{s}}\int_{-\infty}^{\infty}\exp\left(\frac{-y^2}{2s}\right)\,\mathrm{d}\mu_\epsilon(y),
\tag{71}
$$

is Lipschitz with constant $L_\epsilon$. Moreover, $\psi(2\mathcal{R}_j - \langle \overline{\mathbf{K}}_i, \widetilde{\mathbf{v}}_j\rangle^2) = \widetilde{h}_j^i$ and $\psi(2\mathcal{R}_j) = \widetilde{h}_j$. By (67), for all $1 \leq i \leq d$,

$$
\begin{aligned}
|\widetilde{h}_j^i - \widetilde{h}_j| &\leq L_\epsilon\langle \overline{\mathbf{K}}_i, \widetilde{\mathbf{v}}_j\rangle^2 \\
&\leq L_\epsilon\left(C_{\overline{\mathbf{K}}}\max_{z\in\Gamma_0}\max_{1\leq i\leq d}\left|\mathbf{R}(z;\overline{\mathbf{K}})_i^{\mathrm{T}}\widetilde{\mathbf{v}}_j\right|\right)^2.
\end{aligned}
\tag{72}
$$

It follows that

$$
\begin{aligned}
\left|\widetilde{h}_j\overline{\mathbf{K}}\widetilde{\mathbf{v}}_j - \mathbf{A}_j\overline{\mathbf{K}}\widetilde{\mathbf{v}}_j\right| &\leq \left\|\widetilde{h}_j\mathbf{I}_d - \mathbf{A}_j\right\|\left\|\overline{\mathbf{K}}\widetilde{\mathbf{v}}_j\right\| \\
&\leq L_\epsilon\left(C_{\overline{\mathbf{K}}}\max_{z\in\Gamma_0}\max_{1\leq i\leq d}\left|\mathbf{R}(z;\overline{\mathbf{K}})_i^{\mathrm{T}}\widetilde{\mathbf{v}}_j\right|\right)^2\left\|\overline{\mathbf{K}}\widetilde{\mathbf{v}}_j\right\| \\
&\leq C_{\epsilon,\overline{\mathbf{K}}}\sqrt{d}\left(\max_{z\in\Gamma_0}\max_{1\leq i\leq d}\left|\mathbf{R}(z;\overline{\mathbf{K}})_i^{\mathrm{T}}\widetilde{\mathbf{v}}_j\right|\right)^3.
\end{aligned}
\tag{73}
$$

For notational clarity, let us write $\omega_k = \max_{z\in\Gamma_0}\max_{1\leq i\leq d}\left|\mathbf{R}(z;\overline{\mathbf{K}})_i^{\mathrm{T}}\widetilde{\mathbf{v}}_k\right|$. Putting all this together we have up to constant factor,

$$
\begin{aligned}
|\mathcal{E}_j^i(z)| &\leq \frac{\eta}{d}\left(\left|\widetilde{h}_j\widetilde{\mathbf{v}}_j^i\right| + \|\mathbf{R}(z;\overline{\mathbf{K}})\|\left(\left\|\widetilde{h}_j\mathbf{I}_d - \mathbf{A}_j\right\|\left\|\overline{\mathbf{K}}\widetilde{\mathbf{v}}_j\right\| + \|\mathbf{E}_{j+1}\|\right)\right) \\
&\leq \frac{\eta C_{\epsilon,\overline{\mathbf{K}}}}{d}\left(\omega_j + 2\sqrt{d}\omega_j^3\right).
\end{aligned}
\tag{74}
$$

Returning to (66), upon taking the max across $z \in \Gamma_0$ and $1 \leq i \leq d$ and up to a constant $C_{\eta,t,\epsilon,\overline{\mathbf{K}}} > 0$, we obtain for all $k \leq \lfloor td\rfloor$

$$
\omega_k \leq C_{\eta,t,\epsilon,\overline{\mathbf{K}}}\left(\omega_0 + \max_{z\in\Gamma_0}\max_{1\leq i\leq d}\max_{1\leq k\leq\lfloor td\rfloor}\overline{\mathcal{M}}_k^i(z) + \sum_{j=0}^{k-1}\frac{\eta}{d}\left(\omega_j + 2\sqrt{d}\omega_j^3\right)\right).
\tag{75}
$$

Define

$$\beta_t = C_{\eta,t,\epsilon,\overline{\mathbf{K}}} \left( \omega_0 + \max_{z \in \Gamma_0} \max_{1 \leq i \leq d} \max_{1 \leq k \leq \lfloor td \rfloor} \overline{\mathcal{M}}_k^i(z) \right), \tag{76}$$

as well as the stopping time

$$\tau_2 = \min \left\{ k \, ; \omega_k \geq 3\beta_t \exp \left( C_{\eta,t,\epsilon,\overline{\mathbf{K}}} \right) \right\}. \tag{77}$$

As before, we note that $\omega_k$ can only move at increments of at-most $\frac{\eta M_r}{\sqrt{d}}$. Thus,

$$\omega_{k \wedge \tau_2} \leq 3\beta_t \exp(C_{\eta,t,\epsilon,\overline{\mathbf{K}}}) + \frac{\eta M_r}{\sqrt{d}} =: \beta_t', \tag{78}$$

for all $k \in \mathbb{N}$. Plugging this into (75),

$$\omega_{\lfloor td \rfloor \wedge \tau_2} \leq \beta_t + C_{\eta,t,\epsilon,\overline{\mathbf{K}}} \left( \sum_{j=0}^{(\lfloor td \rfloor - 1) \wedge \tau_2} \frac{\eta}{d} \omega_j + \sum_{j=0}^{\lfloor td \rfloor} \frac{\eta}{d} \left( 2\sqrt{d}\omega_j^3 \right) \right)$$

$$\leq \beta_t + C_{\eta,t,\epsilon,\overline{\mathbf{K}}} \left[ 2\sqrt{d}(\beta_t')^3 \right] + \sum_{j=0}^{(\lfloor td \rfloor - 1) \wedge \tau_2} C_{\eta,t,\epsilon,\overline{\mathbf{K}}} \frac{\eta}{d} \omega_j. \tag{79}$$

By Gronwall's inequality,

$$\omega_{\lfloor td \rfloor \wedge \tau_2} \leq \left( \beta_t + C_{\eta,t,\epsilon,\overline{\mathbf{K}}} \sqrt{d} \left[ 2(\beta_t')^3 \right] \right) \exp \left( C_{\eta,t,\epsilon,\overline{\mathbf{K}}} \right). \tag{80}$$

If we can show that $\beta_t$ can be made sufficiently small so that

$$C_{\eta,t,\epsilon,\overline{\mathbf{K}}} \left[ \sqrt{d}(\beta_t')^3 \right] \leq \beta_t, \tag{81}$$

then $\omega_{\lfloor td \rfloor \wedge \tau_2} = \omega_{\lfloor td \rfloor}$. To see this, recall by Assumption 5 we know that for any constant $\xi > 0$, the former term of $\beta_t$ has the following tail bound,

$$\mathbb{P} \left( \max_{z \in \Gamma_0} \max_{1 \leq i \leq d} \left| \mathbf{R}(z; \overline{\mathbf{K}})_i^{\mathrm{T}} \mathbf{v}_0 \right| \geq \frac{\xi d^\delta}{\sqrt{d}} \right) \leq C d^2 \exp \left( -c' \xi^2 d^{2\delta} \right). \tag{82}$$

To bound the martingale term, we first fix $z \in \Gamma_0$ and $1 \leq i \leq d$, then let

$$\tau_3 = \min \left\{ k \, ; |\overline{\mathcal{M}}_k^i(z)| \geq \frac{\xi d^\delta}{\sqrt{d}} \right\}. \tag{83}$$

Let $X_k = \overline{\mathcal{M}}_{k \wedge \tau_3}^i(z)$. Notice that $\mathbb{E} \left[ \overline{\mathcal{M}}_k^i(z) \right] = 0$, so $\mathbb{E} \left[ X_k \right] = 0$. It follows that

$$\mathbb{P} \left( \max_{1 \leq k \leq \lfloor td \rfloor} |\overline{\mathcal{M}}_k^i(z)| \geq \frac{\xi d^\delta}{\sqrt{d}} \right) = \mathbb{P} \left( \left| X_{\lfloor td \rfloor} \right| \geq \frac{\xi d^\delta}{\sqrt{d}} \right). \tag{84}$$

Notice that

$$\left| \overline{\mathcal{M}}_{k+1}^i(z) - \overline{\mathcal{M}}_k^i(z) \right| = \frac{1}{|L_k|} \left| \mathbf{R}(z; \overline{\mathbf{K}})_i^{\mathrm{T}} (\mathbf{v}_{k+1} - \mathbf{v}_k) - \mathbf{R}(z; \overline{\mathbf{K}})_i^{\mathrm{T}} \mathbb{E} \left[ \mathbf{v}_{k+1} - \mathbf{v}_k \middle| \mathcal{F}_k \right] \right|$$

$$\leq \frac{C_{t,\overline{\mathbf{K}}} \eta M_R}{\sqrt{d}}. \tag{85}$$

Hence, $|X_k - X_{k-1}| \leq \frac{C_{t,\overline{\mathbf{K}}} \eta M_R}{\sqrt{d}}$ almost surely for all $k$. However, we may improve this increment bound by $\frac{d^s}{d}$ for $\frac{1}{6} + \delta_0 < s < \delta$. Indeed, by Corollary 2 for all even moments $2p < d$, there exists a constant $C(2p, \eta, \mathbf{K})$ such that

$$\mathbb{P} \left( |X_{k+1} - X_k| \geq \frac{d^s}{d} \right) \leq C(2p, \eta, \mathbf{K}) d^{p\left(\frac{1}{3} - 2s + 2\delta_0\right)}. \tag{86}$$

It follows by Lemma 15,

$$\mathbb{P}\left(\left|X_{\lfloor td\rfloor}\right| \geq \frac{\xi d^{\delta}}{\sqrt{d}}\right) \leq 2\exp\left(-\frac{\xi^2 d^{2(\delta-s)}}{C_{\eta,t,\overline{\mathbf{K}}}}\right) + \lfloor td\rfloor C(2p,\eta,\mathbf{K})d^{p\left(\frac{1}{3}-2s+2\delta_0\right)} \tag{87}$$

Thus, taking union bounds across $z \in \Gamma_0$ and $1 \leq i \leq d$ gives,

$$\mathbb{P}\left(\max_{z\in\Gamma_0}\max_{1\leq i\leq d}\max_{1\leq k\leq\lfloor td\rfloor+1}\left|\overline{\mathcal{M}}_k^i(z)\right| \geq \frac{\xi d^{\delta}}{\sqrt{d}}\right) \leq Cd^2\exp\left(-\frac{\xi^2 d^{2(\delta-s)}}{C_{\eta,t,\overline{\mathbf{K}}}}\right)$$
$$+ 2Cd^2\lfloor td\rfloor C(2p,\eta,\mathbf{K})d^{p\left(\frac{1}{3}-2s+2\delta_0\right)} \tag{88}$$

It is easy to see that for $d$ sufficiently large, we may choose $p$ large so that $s > \frac{1}{6} + \delta_0 + \frac{3}{2p}$, implying the latter term converges to 0 as $d \to \infty$. Therefore, $\beta_t \leq \frac{\xi d^{\delta}}{\sqrt{d}}$ with high-probability. Returning to (81), up to a constant factor that is independent to $d$,

$$\mathbb{P}\left(\sqrt{d}\beta_t^3 > \beta_t\right) = \mathbb{P}\left(\beta_t^2 > \frac{1}{\sqrt{d}}\right)$$
$$= \mathbb{P}\left(\frac{d^{2\delta}}{d} > \beta_t^2 > \frac{1}{\sqrt{d}}\right) + \mathbb{P}\left(\beta_t^2 > \frac{1}{\sqrt{d}}, \beta_t^2 \geq \frac{d^{2\delta}}{d}\right)$$
$$= \mathbb{P}\left(\beta_t \geq \frac{d^{\delta}}{\sqrt{d}}\right), \tag{89}$$

provided that $\delta < 1/4$. Thus, (81) is satisfied and $\omega_{\lfloor td\rfloor\wedge\tau_2} = \omega_{\lfloor td\rfloor}$ with high-probability. By choosing $\xi$ appropriately in accordance to (77), we conclude $\omega_{\lfloor td\rfloor} \leq \frac{d^{\delta}}{\sqrt{d}}$ with high-probability. $\qquad\square$

We can now formalize our prior statement of

$$\mathbb{E}\left[\mathbf{v}_{k+1}^i - \mathbf{v}_k^i\middle|\mathcal{F}_k\right] \approx -\frac{\eta}{d}\frac{\varphi(\mathcal{R}_k)}{\sqrt{2\mathcal{R}_k}}\left\langle\overline{\mathbf{K}}_i, \mathbf{v}_k\right\rangle, \tag{90}$$

for all $1 \leq i \leq d$.

**Lemma 4.** *Conditional on $\mathcal{F}_k$, the mean of* (34) *is*

$$\mathbb{E}\left[\mathbf{v}_{k+1} - \mathbf{v}_k\middle|\mathcal{F}_k\right] = -\frac{\eta\varphi(\mathcal{R}_k)}{d\sqrt{2\mathcal{R}_k}}\overline{\mathbf{K}}\mathbf{v}_k + \mathbf{E}_{k+1}, \tag{91}$$

*where $\mathbf{E}_{k+1}$ is an error term such that for all $\rho \in (1/6 + \delta_0, 1/4)$, with high-probability $|\mathbf{E}_{k+1}^i| = O\left(\frac{d^{3\rho}}{d^{5/2}}\right)$ for all $1 \leq i \leq d$.*

*Proof.* By (35), the coordinate-wise error can be defined as

$$\mathbf{E}_{k+1}^i = \frac{\eta}{d}\left(\mathbb{E}\left[\sigma(\mathbf{x}_{k+1}^i)R_{k+1}^i\right] + \frac{\varphi(\mathcal{R}_k)}{\sqrt{2\mathcal{R}_k}}\left\langle\overline{\mathbf{K}}_i, \mathbf{v}_k\right\rangle - \sqrt{\frac{2}{\pi}}2h_k^i(0)\left\langle\overline{\mathbf{K}}_i, \mathbf{v}_k\right\rangle\right). \tag{92}$$

Applying Lemma 3 onto (69) and (72) yields the result. $\qquad\square$

### A.2. Convergence of SIGNSGD to SIGNHSGD

In this section we will show convergence of the dynamics of SIGNSGD to that of SIGNHSGD. Recall SIGNHSGD is defined as in (8). Similarly to SIGNSGD we will impose a stopping time onto SIGNHSGD,

$$\tau_0' = \min_{t>0}\left\{t; \|\mathbf{\Theta}_t - \boldsymbol{\theta}_*\| > L\right\}. \tag{93}$$

We will also define the stopped process by $\mathbf{V}_t = \mathbf{\Theta}_{t\wedge\tau_0'} - \boldsymbol{\theta}_*$. We will prove the main result for the stopped SIGNSGD process $\mathbf{v}_k$ and stopped SIGNHSGD process $\mathbf{V}_t$, then in Lemma 8 we will generalize to the non-stopped process $\boldsymbol{\theta}_k$ and $\mathbf{\Theta}_t$.

We shall use the following:

**Definition 5** (Quadratic). *A function $q : \mathbb{R}^d \to \mathbb{R}$ is quadratic if it may be written in the form*

$$q(\mathbf{x}) = \mathbf{x}^{\mathrm{T}}\mathbf{A}\mathbf{x} + \mathbf{b}^{\mathrm{T}}\mathbf{x} + c$$

*for some $\mathbf{A} \in \mathbb{R}^{d \times d}$, $\mathbf{b} \in \mathbb{R}^d$, and $c \in \mathbb{R}$.*

Once again, for notational convenience we will denote $\sigma_{k+1} = \sigma(\mathbf{x}_{k+1})\sigma(\langle \mathbf{x}_{k+1}, \boldsymbol{\nu}_k \rangle - \epsilon_{k+1})$ when it is clear. Now if $q : \mathbb{R}^d \to \mathbb{R}$ is quadratic, it is easy to see that

$$q(\mathbf{v}_{k+1}) - q(\mathbf{v}_k) = -\frac{\eta}{d}\nabla q(\mathbf{v}_k)^{\mathrm{T}}(\sigma_{k+1}) + \frac{\eta^2}{2d^2}(\sigma_{k+1})^{\mathrm{T}}\nabla^2 q(\mathbf{v}_k)(\sigma_{k+1}). \tag{94}$$

Thus, taking its conditional expectation we obtain

$$\mathbb{E}[q(\mathbf{v}_{k+1}) - q(\mathbf{v}_k)|\mathcal{F}_k] = -\frac{\eta\varphi(\mathcal{R}_k)}{d\sqrt{2\mathcal{R}_k}}\nabla q(\mathbf{v}_k)^{\mathrm{T}}\overline{\mathbf{K}}\mathbf{v}_k + \frac{2\eta^2}{d^2\pi}\langle \nabla^2 q(\mathbf{v}_k), \mathbf{K}_\sigma \rangle + O\left(\frac{d^{3\rho}}{d^2}\right). \tag{95}$$

By the Doob-decomposition, we have

$$q(\mathbf{v}_{k+1}) - q(\mathbf{v}_k) = -\frac{\eta\varphi(\mathcal{R}_k)}{d\sqrt{2\mathcal{R}_k}}\nabla q(\mathbf{v}_k)^{\mathrm{T}}\overline{\mathbf{K}}\mathbf{v}_k + \frac{2\eta^2}{d^2\pi}\langle \nabla^2 q(\mathbf{v}_k), \mathbf{K}_\sigma \rangle \tag{96}$$

$$+ O\left(\frac{d^{3\rho}}{d^2}\right) + \Delta\mathcal{M}_{k+1}^{lin} + \Delta\mathcal{M}_{k+1}^{quad}, \tag{97}$$

where

$$\Delta\mathcal{M}_{k+1}^{lin} = -\frac{\eta}{d}\nabla q(\mathbf{v}_k)^{\mathrm{T}}(\sigma_{k+1} - \mathbb{E}[\sigma_{k+1}|\mathcal{F}_k]), \tag{98}$$

and

$$\Delta\mathcal{M}_{k+1}^{quad} = \frac{\eta^2}{2d^2}(\sigma_{k+1}^{\mathrm{T}}\nabla^2 q(\mathbf{v}_k)\sigma_{k+1} - \mathbb{E}[\sigma_{k+1}^{\mathrm{T}}\nabla^2 q(\mathbf{v}_k)\sigma_{k+1}|\mathcal{F}_k]). \tag{99}$$

Similarly, by Ito's lemma on $\mathbf{V}_t$, we see that

$$dq(\mathbf{V}_t) = \left(-\frac{\eta\varphi(\mathcal{R}(\mathbf{V}_t))}{\sqrt{2\mathcal{R}(\mathbf{V}_t)}}\nabla q(\mathbf{V}_t)^{\mathrm{T}}\overline{\mathbf{K}}\mathbf{V}_t + \frac{2\eta^2}{\pi d}\langle \nabla^2 q(V_t), \mathbf{K}_\sigma \rangle\right)dt + d\mathcal{M}_t^\sigma, \tag{100}$$

where

$$d\mathcal{M}_t^\sigma = \eta\nabla q(\mathbf{V}_t)^{\mathrm{T}}\left(\sqrt{\frac{2\mathbf{K}_\sigma}{d\pi}}d\mathbf{B}_t\right). \tag{101}$$

Comparing (97) and (100), we see that predictable part of signSGD and the total variation part of HSGD depend only on $\nabla q(\mathbf{x})^{\mathrm{T}}\overline{\mathbf{K}}\mathbf{x}$ and $\mathcal{R}(\mathbf{x})$. We capture these statistics in a "closed" manifold defined by

$$Q_q = \left\{\mathbf{x}^{\mathrm{T}}\mathbf{x}, q(\mathbf{x}), \nabla q(\mathbf{x})^{\mathrm{T}}\mathbf{R}(z; \overline{\mathbf{K}})\mathbf{x}, \mathbf{x}^{\mathrm{T}}\mathbf{R}(z; \overline{\mathbf{K}})^{\mathrm{T}}\nabla^2 q(\mathbf{x})\mathbf{R}(y; \overline{\mathbf{K}})\mathbf{x}; \ z, y \in \Gamma\right\}. \tag{102}$$

To be precise in our notion of closure, given any $g \in Q_q$, the predictable part of (97) (not accounting the error) and the drift part of (100) may be expressed via contour integral around $\Gamma$ by a linear combination of functions from $Q_q$. Let us look at an example. Suppose $g(\mathbf{x}) = \nabla q(\mathbf{x})^{\mathrm{T}}\mathbf{R}(z; \overline{\mathbf{K}})\mathbf{x}$. It is easy to see that

$$\mathbb{E}\left[\nabla g(\mathbf{v}_k)^{\mathrm{T}}(\mathbf{v}_{k+1} - \mathbf{v}_k)|\mathcal{F}_k\right] = -\frac{\eta\varphi(\mathcal{R}_k)}{d\sqrt{2\mathcal{R}_k}}\left(\mathbf{v}_k^{\mathrm{T}}\nabla^2 q\mathbf{R}(z; \overline{\mathbf{K}})\overline{\mathbf{K}}\mathbf{v}_k + \mathbf{v}_k^{\mathrm{T}}\mathbf{R}(z; \overline{\mathbf{K}})^{\mathrm{T}}\nabla^2 q\overline{\mathbf{K}}\mathbf{v}_k\right)$$

$$+ O\left(\frac{d^{3\rho}}{d^2}\right) \tag{103}$$

$$= -\frac{\eta\varphi(\mathcal{R}_k)}{d\sqrt{2\mathcal{R}_k}}\left(\mathbf{v}_k^{\mathrm{T}}\nabla^2 q\mathbf{v}_k\right) + O\left(\frac{d^{3\rho}}{d^2}\right)$$

$$- \frac{\eta\varphi(\mathcal{R}_k)}{d\sqrt{2\mathcal{R}_k}}\underbrace{\left(z\mathbf{v}_k^{\mathrm{T}}\nabla^2 q\mathbf{R}(z; \overline{\mathbf{K}})\mathbf{v}_k + \mathbf{v}_k^{\mathrm{T}}\mathbf{R}(z; \overline{\mathbf{K}})^{\mathrm{T}}\nabla^2 q\overline{\mathbf{K}}\mathbf{v}_k\right)}_{p(\mathbf{v}_k)}. \tag{104}$$

Notice that our error $O\left(\frac{d^{3\rho}}{d^2}\right)$ is independent to choice of $g$. This is because the resolvent $\mathbf{R}(z;\overline{\mathbf{K}})$ has uniformly bounded operator norm for all $z \in \Gamma$, thus the $\|\cdot\|_{C^2}$ is also uniformly bounded for all $g \in Q_q$. It then follows that

$$\nabla g(\mathbf{v}_k)^{\mathrm{T}}\mathbf{E}_{k+1} \leq \|\nabla g(\mathbf{v}_k)\|\,\|\mathbf{E}_{k+1}\| \leq \|g\|_{C^2}\,(1+\|\mathbf{v}_k\|)\,\|\mathbf{E}_{k+1}\| = O\left(\frac{d^{3\rho}}{d^2}\right). \tag{105}$$

In addition, by the Cauchy's integral theorem we may express $p(\mathbf{v}_k)$ by

$$p(\mathbf{v}_k) = -\frac{1}{2\pi i}\oint_\Gamma z\mathbf{v}_k^{\mathrm{T}}\mathbf{R}(y;\overline{\mathbf{K}})^{\mathrm{T}}\nabla^2 q\mathbf{R}(z;\overline{\mathbf{K}})\mathbf{v}_k + y\mathbf{v}_k^{\mathrm{T}}\mathbf{R}(z;\overline{\mathbf{K}})^{\mathrm{T}}\nabla^2 q\mathbf{R}(y;\overline{\mathbf{K}})\mathbf{v}_k\,\mathrm{d}y, \tag{106}$$

as well as

$$\mathbf{v}_k^{\mathrm{T}}\nabla^2 q\mathbf{v}_k = \frac{1}{4\pi^2}\oint_\Gamma\oint_\Gamma \mathbf{v}_k^{\mathrm{T}}\mathbf{R}(z;\overline{\mathbf{K}})\nabla^2 q\mathbf{R}(y;\overline{\mathbf{K}})\mathbf{v}_k\,\mathrm{d}z\mathrm{d}y. \tag{107}$$

Consequently, we see that

$$\left|\mathbb{E}\left[\nabla g(\mathbf{v}_k)^{\mathrm{T}}(\mathbf{v}_{k+1}-\mathbf{v}_k)|\mathcal{F}_k\right]\right| \leq \frac{\eta\varphi(\mathcal{R}_k)}{d\sqrt{2\mathcal{R}_k}}12\left\|\overline{\mathbf{K}}\right\|^2\max_{g\in Q_q}|g(\mathbf{v}_k)| + O\left(\frac{d^{3\rho}}{d^2}\right) \tag{108}$$

$$\leq \frac{12\eta M_\epsilon}{d}\left\|\overline{\mathbf{K}}\right\|^2\max_{g\in Q_q}|g(\mathbf{v}_k)| + O\left(\frac{d^{3\rho}}{d^2}\right), \tag{109}$$

where we applied Lemma 12 in the second inequality. Note the constant factor of $12\left\|\overline{\mathbf{K}}\right\|^2$ depended on $g$. We may work around this quadratic dependent constant to obtain a uniform bound on (109) for all $g \in Q_q$ with the following lemma:

**Lemma 5.** *Let $Q_q$ be defined as above then for all $n > 0$ there exists $\overline{Q}_q \subset Q_q$ such that $|\overline{Q}_q| \leq C(\overline{\mathbf{K}})d^{4n}$ and for all $g \in Q_q$, there exists $g_0 \in \overline{Q}_q$ satisfying $\|g - g_0\|_{C^2} \leq d^{-2n}$.*

The proof of Lemma 5 may be found in Collins-Woodfin et al. (2024).

**Lemma 6.** *There exists constants $C(\overline{\mathbf{K}}), M_\epsilon > 0$ such that for all $g \in Q_q$, $k \in \mathbb{N}$ and $\rho \in (1/6+\delta_0, 1/4)$,*

$$\left|\mathbb{E}\left[\nabla g(\mathbf{v}_k)^{\mathrm{T}}(\mathbf{v}_{k+1}-\mathbf{v}_k)|\mathcal{F}_k\right]\right| \leq \frac{\eta M_\epsilon}{d}C(\overline{\mathbf{K}})\max_{g\in Q_q}|g(\mathbf{v}_k)| + O\left(\frac{d^{3\rho}}{d^2}\right). \tag{110}$$

*Proof.* Let $g \in Q_q$ and $n > 0$, then by Lemma 5 there exists $g_0 \in \overline{Q}_q$ such that $\|g - g_0\|_{C^2} \leq d^{-2n}$. It follows that

$$\left|\mathbb{E}\left[\nabla g(\mathbf{v}_k)^{\mathrm{T}}(\mathbf{v}_{k+1}-\mathbf{v}_k)|\mathcal{F}_k\right]\right| = \left|\frac{\eta\varphi(\mathcal{R}_k)}{d\sqrt{2\mathcal{R}_k}}\nabla g(\mathbf{v}_k)^{\mathrm{T}}\overline{\mathbf{K}}\mathbf{v}_k\right| + O\left(\frac{d^{3\rho}}{d^2}\right) \tag{111}$$

$$\leq \frac{\eta\varphi(\mathcal{R}_k)}{d\sqrt{2\mathcal{R}_k}}\left(|\nabla(g-g_0)(\mathbf{v}_k)^{\mathrm{T}}\overline{\mathbf{K}}\mathbf{v}_k| + |\nabla g_0(\mathbf{v}_k)^{\mathrm{T}}\overline{\mathbf{K}}\mathbf{v}_k|\right) \tag{112}$$

$$+ O\left(\frac{d^{3\rho}}{d^2}\right) \tag{113}$$

$$\leq \frac{\eta M_\epsilon}{d}\left(\|g-g_0\|_{C^2}\left\|\overline{\mathbf{K}}\right\|\|\mathbf{v}_k\|^2 + C_{g_0}(\overline{\mathbf{K}})\max_{g\in Q_q}|g(\mathbf{v}_k)|\right) \tag{114}$$

$$+ O\left(\frac{d^{3\rho}}{d^2}\right) \tag{115}$$

$$\leq \frac{\eta M_\epsilon}{d}\left(d^{-2n}\left\|\overline{\mathbf{K}}\right\| + C_{g_0}(\overline{\mathbf{K}})\right)\max_{g\in Q_q}|g(\mathbf{v}_k)| + O\left(\frac{d^{3\rho}}{d^2}\right), \tag{116}$$

where $C_{g_0}(\overline{\mathbf{K}})$ is the choice dependent constant as in (109). By taking the max across our finite net $\overline{Q}_q$, there exists $C(\overline{\mathbf{K}}) > 0$ such that for all $g \in Q_q$,

$$\left|\mathbb{E}\left[\nabla g(\mathbf{v}_k)^{\mathrm{T}}(\mathbf{v}_{k+1}-\mathbf{v}_k)|\mathcal{F}_k\right]\right| \leq \frac{\eta M_\epsilon}{d}C(\overline{\mathbf{K}})\max_{g\in Q_q}|g(\mathbf{v}_k)| + O\left(\frac{d^{3\rho}}{d^2}\right). \tag{117}$$

$\square$

We are now ready to prove our main result. It would be convenient to extend the indexing of $\mathbf{v}_k$ from $\mathbb{N}$ to $\mathbb{R}$ by defining the sequence $t_k = k/d$. With some slight abuse of notation, let $\mathbf{v}_{t_k} = \mathbf{v}_k$. If $t_{k-1} \leq t < t_k$, then define $\mathbf{v}_t = \mathbf{v}_{t_{k-1}}$.

**Lemma 7.** *Given $0 < 2p < d$ and a quadratic $q$ such that $\|q\|_{C^2} \leq 1$, define $Q = Q_q \cup Q_{\mathcal{R}}$, where $\mathcal{R}$ is the risk. For all $T > 3$ and $1/3 < \delta < 1/2$, there exists $C(\overline{\mathbf{K}}, \epsilon) > 0$ such that*

$$\sup_{0 \leq t \leq T} |q(\mathbf{v}_t) - q(\mathbf{V}_t)| \leq \frac{T d^\delta}{\sqrt{d}} \exp\left(C(\overline{\mathbf{K}}, \epsilon) \|\eta\|_\infty T\right), \tag{118}$$

*with probability at least $1 - c(2p, \overline{\mathbf{K}}) d^{p(1/3-\delta)}$.*

*Proof.* Let $g \in Q$, by (97), we see that

$$g(\mathbf{v}_t) = g(\mathbf{v}_0) - \frac{\eta}{d} \sum_{i=0}^{\lfloor td \rfloor} \frac{\varphi(\mathcal{R}(\mathbf{v}_i))}{\sqrt{2\mathcal{R}(\mathbf{v}_i)}} \nabla g(\mathbf{v}_i)^{\mathrm{T}} \overline{\mathbf{K}} \mathbf{v}_i \tag{119}$$

$$+ \lfloor td \rfloor O\left(\frac{d^{3\rho}}{d^2}\right) + \frac{\eta^2}{d^2 \pi} \sum_{i=0}^{\lfloor td \rfloor} \left\langle \nabla^2 g(\mathbf{v}_k), \mathbf{K}_\sigma \right\rangle + \mathcal{M}_t^{lin} + \mathcal{M}_t^{quad}$$

$$= g(\mathbf{v}_0) - \eta \int_0^t \frac{\varphi(\mathcal{R}(\mathbf{v}_s))}{\sqrt{2\mathcal{R}(\mathbf{v}_s)}} \nabla g(\mathbf{v}_s)^{\mathrm{T}} \overline{\mathbf{K}} \mathbf{v}_s \, ds \tag{120}$$

$$+ \lfloor td \rfloor O\left(\frac{d^{3\rho}}{d^2}\right) + \frac{2\eta^2}{d\pi} \int_0^t \left\langle \nabla^2 g(\mathbf{v}_s), \mathbf{K}_\sigma \right\rangle ds$$

$$+ \mathcal{M}_t^{lin} + \mathcal{M}_t^{quad}. \tag{121}$$

Taking the difference with SIGNHSGD, we see that

$$|g(\mathbf{v}_t) - g(\mathbf{V}_t)| \leq \eta \int_0^t \left| \frac{\varphi(\mathcal{R}(\mathbf{v}_s))}{\sqrt{2\mathcal{R}(\mathbf{v}_s)}} \nabla g(\mathbf{v}_s)^{\mathrm{T}} \overline{\mathbf{K}} \mathbf{v}_s - \frac{\varphi(\mathcal{R}(\mathbf{V}_s))}{\sqrt{2\mathcal{R}(\mathbf{V}_s)}} \nabla g(\mathbf{V}_s)^{\mathrm{T}} \overline{\mathbf{K}} \mathbf{V}_s \right| ds$$

$$+ \sup_{0 \leq s \leq t} \left(|\mathcal{M}_s^{lin}| + |\mathcal{M}_s^{quad}| + |\mathcal{M}_s^\sigma|\right) + O\left(\frac{d^{3\rho}}{d}\right) t. \tag{122}$$

However, Lemma 12 tells us the map

$$(a, b) \mapsto \frac{\varphi(a)}{\sqrt{2a}} b, \tag{123}$$

on a bounded domain is Lipschitz continuous with constant $L_\epsilon > 0$. Thus, using the same argument as in (110) we may bound the integrand by

$$\left| \frac{\varphi(\mathcal{R}(\mathbf{v}_s))}{\sqrt{2\mathcal{R}(\mathbf{v}_s)}} \nabla g(\mathbf{v}_s)^{\mathrm{T}} \overline{\mathbf{K}} \mathbf{v}_s - \frac{\varphi(\mathcal{R}(\mathbf{V}_s))}{\sqrt{2\mathcal{R}(\mathbf{V}_s)}} \nabla g(\mathbf{V}_s)^{\mathrm{T}} \overline{\mathbf{K}} \mathbf{V}_s \right|$$

$$\leq L_\epsilon \sqrt{\left(\nabla g(\mathbf{v}_s)^{\mathrm{T}} \overline{\mathbf{K}} \mathbf{v}_s - \nabla g(\mathbf{V}_s)^{\mathrm{T}} \overline{\mathbf{K}} \mathbf{V}_s\right)^2 + \left(\mathcal{R}(\mathbf{v}_s) - \mathcal{R}(\mathbf{V}_s)\right)^2}$$

$$\leq L_\epsilon C(\overline{\mathbf{K}}) \max_{g \in Q} |g(\mathbf{v}_s) - g(\mathbf{V}_s)|. \tag{124}$$

Plugging into (122) we get

$$\sup_{g \in Q} |g(\mathbf{v}_t) - g(V_t)| \leq \sup_{0 \leq s \leq t} \left(|\mathcal{M}_s^{lin}| + |\mathcal{M}_s^{quad}| + |\mathcal{M}_s^\sigma|\right) + O\left(\frac{d^{3\rho}}{d}\right) t$$

$$+ \eta L_\epsilon C(\overline{\mathbf{K}}) \int_0^t \max_{g \in Q} |g(\mathbf{v}_s) - g(\mathbf{V}_s)| \, ds. \tag{125}$$

By Gronwall's inequality,

$$\sup_{g \in Q} |g(\mathbf{v}_t) - g(\mathbf{V}_t)| \leq \left( \sup_{0 \leq s \leq t} \left(|\mathcal{M}_s^{lin}| + |\mathcal{M}_s^{quad}| + |\mathcal{M}_s^\sigma|\right) + O\left(\frac{d^{3\rho}}{d}\right) t \right) \exp\left(\eta L_\epsilon C(\overline{\mathbf{K}})t\right). \tag{126}$$

Lemmas 9, 10 and 11 bounds the martingales by $\frac{d^\delta}{\sqrt{d}}$ for $1/3 < \delta < 1/2$. Subsequently, $\frac{d^\delta}{\sqrt{d}}$ bounds $O\left(\frac{d^{3\rho}}{d}\right)$, concluding the proof. □

We have now shown that the stopped processes satisfy the conclusion of Theorem 1. We will conclude the proof of Theorem 1 by showing that, with high-probability, the process is not stopped.

**Lemma 8.** *For all $T > 0$, there exists $C(\overline{\mathbf{K}}, \mathbf{K}_\sigma) > 0$ such that*

$$\max_{0 \leq t \leq T} \|\mathbf{V}_t\| \leq \exp\left(TC(\overline{\mathbf{K}}, \mathbf{K}_\sigma)\right), \tag{127}$$

*with overwhelming probability.*

*Proof.* For $\mathbf{z} \in \mathbb{R}^d$ let $\psi(\mathbf{z}) = \log(1 + \|\mathbf{z}\|^2)$. By Itô's lemma,

$$d\psi(\mathbf{V}_t) = \left[\frac{-2\eta\varphi(\mathcal{R}_t)}{\sqrt{2\mathcal{R}_t}(1 + \|\mathbf{V}_t\|^2)}\mathbf{V}_t^{\mathrm{T}}\overline{\mathbf{K}}\mathbf{V}_t - \frac{\eta^2}{d\pi(1 + \|\mathbf{V}_t\|^2)}\mathbf{V}_t^{\mathrm{T}}\mathbf{K}_\sigma\mathbf{V}_t\right]dt$$
$$+ \left[\frac{2\eta^2}{d\pi(1 + \|\mathbf{V}_t\|^2)}\operatorname{Tr}(\mathbf{K}_\sigma)\right]dt + \frac{2\eta}{(1 + \|\mathbf{V}_t\|^2)}\mathbf{V}_t^{\mathrm{T}}\sqrt{\frac{\mathbf{K}_\sigma}{d\pi}}d\mathbf{B}_t. \tag{128}$$

It is easy to check by the Cauchy-Schwarz inequality that the deterministic terms of may be uniformly bounded by some constant $C(\overline{\mathbf{K}}, \mathbf{K}_\sigma) > 0$. Denote the martingale term by $\mathcal{M}_t^{\sigma-HSGD}$ then the quadratic variation is given by,

$$\langle\mathcal{M}^{\sigma-HSGD}\rangle_t = \frac{4\eta^2}{d\pi(1 + \|\mathbf{V}_t\|^2)^2}\int_0^t \mathbf{V}_s^{\mathrm{T}}\mathbf{K}_\sigma\mathbf{V}_s\,ds \tag{129}$$

$$\leq \frac{\eta^2\|\mathbf{K}_\sigma\|t}{d\pi}. \tag{130}$$

By subgaussian concentration,

$$\mathbb{P}\left(\max_{0 \leq t \leq T}\psi(\mathbf{V}_t) \geq 2TC(\overline{\mathbf{K}}, \mathbf{K}_\sigma)\right) \leq \mathbb{P}\left(\max_{0 \leq t \leq T}\mathcal{M}_t^{\sigma-HSGD} \geq TC(\overline{\mathbf{K}}, \mathbf{K}_\sigma)\right)$$

$$\leq 2\exp\left(\frac{-C(\overline{\mathbf{K}}, \mathbf{K}_\sigma)^2Td\pi}{2\eta^2\|\mathbf{K}_\sigma\|}\right). \tag{131}$$

That is

$$\max_{0 \leq t \leq T}\|\mathbf{V}_t\| \leq \exp\left(TC(\overline{\mathbf{K}}, \mathbf{K}_\sigma)\right), \tag{132}$$

with overwhelming probability. □

Therefore by choosing the upper bound in our stopping $\tau_0$ and $\tau_0'$ in accordance to Lemma 8, we obtain $\mathbf{v}_t = \boldsymbol{\theta}_t - \boldsymbol{\theta}_*$ and $\mathbf{V}_t = \boldsymbol{\Theta}_t - \boldsymbol{\theta}_*$ for all $0 \leq t \leq T$ with overwhelming probability. Combining this with Lemma 7 proves Theorem 1 as well as the following generalization:

**Theorem 5.** *Given Assumptions 1–5 and a quadratic $q : \mathbb{R}^d \to \mathbb{R}$, if $g(\mathbf{x}) = q(\mathbf{x} - \boldsymbol{\theta}_*)$ then choosing any fixed even moment $2p \in (0, d)$, there exists a constant $C(\overline{\mathbf{K}}, \epsilon) > 0$ such that for any $\delta \in (1/3, 1/2)$ and all $T > 3$,*

$$\sup_{0 \leq t \leq T}|g(\boldsymbol{\theta}_{\lfloor td \rfloor}) - g(\boldsymbol{\Theta}_t)| \leq \frac{Td^\delta\|g\|_{C^2}}{\sqrt{d}}\exp\left(C(\overline{\mathbf{K}}, \epsilon)\|\eta\|_\infty T\right), \tag{133}$$

*with probability at least $1 - c(2p, \overline{\mathbf{K}})d^{p(1/3-\delta)}$ for a constant $c(2p, \overline{\mathbf{K}})$ independent to $d$.*

## A.3. Main theorem with badly behaved noise

In this section we formulate a version of Theorem 5 without Assumption 2. The key is that we must work on subsets of the state space where the risk remains away from 0. So suppose that we let

$$\vartheta := \min_{t>0} \{t; \|\mathbf{\Theta}_t - \boldsymbol{\theta}_*\| < \varrho\},$$

for a fixed positive $\varrho > 0$.

We note that the map $x \mapsto \varphi(x)$ is Lipschitz on $[\varrho, \infty)$, even without Assumption 2, since

$$\varphi'(s) = \frac{1}{s} \int_{\mathbb{R}} \frac{y^2}{2s} \exp\left(-\frac{y^2}{2s}\right) \mu(\mathrm{d}y).$$

The function $xe^{-x}$ is uniformly bounded on $x \geq 0$ by $e^{-1}$, and hence $|\varphi'(s)| \leq 1/\varrho$ on the interval $[\varrho, \infty)$.

Thus, we can now proceed with the same proof as Theorem 5, although we do not remove the stopping time $\vartheta$. The end result is the following:

**Theorem 6.** *Given Assumptions 1, 3, 4, 5 and a quadratic $q : \mathbb{R}^d \to \mathbb{R}$, if $g(\mathbf{x}) = q(\mathbf{x} - \boldsymbol{\theta}_*)$ then choosing any fixed even moment $2p \in (0, d)$ and choosing any $\varrho > 0$, there exists a constant $C(\overline{\mathbf{K}}, \epsilon, \varrho) > 0$ such that for any $\delta \in (1/3, 1/2)$ and all $T > 3$,*

$$\sup_{0 \leq t \leq T \wedge \vartheta} |g(\boldsymbol{\theta}_{\lfloor td \rfloor}) - g(\mathbf{\Theta}_t)| \leq \frac{Td^\delta \|g\|_{C^2}}{\sqrt{d}} \exp\left(C(\overline{\mathbf{K}}, \epsilon, \varrho) \|\eta\|_\infty T\right), \tag{134}$$

*with probability at least $1 - c(2p, \overline{\mathbf{K}})d^{p(1/3-\delta)}$ for a constant $c(2p, \overline{\mathbf{K}})$ independent to $d$.*

We remark that if the risk of SIGNHSGD remains bounded away from 0, which will be the case for constant stepsize and nonzero noise, one could additionally show that $\vartheta$ does not occur with high probability. In that case, one can derive as a corollary of Theorem 6 a statement without $\vartheta$.

## A.4. Bounding martingale terms

**Lemma 9.** *For all $g \in Q_q$ as defined in Equation (102) and $1/3 < \delta < 1/2$,*

$$\sup_{0 \leq k \leq \lfloor Td \rfloor} |\mathcal{M}_k^{lin}| < \frac{d^\delta}{\sqrt{d}}, \tag{135}$$

*with high-probability.*

*Proof.* Recall that under $\tau_0$, $\mathbf{v}_k \leq L$. Moreover, given that $\|q\|_{C^2} \leq 1$ and $\|\mathbf{R}(z; \overline{\mathbf{K}})\| \leq M_R$, we see that $\|g\|_{C^2}$ is uniformly bounded for all $g \in Q$. Therefore,

$$\|\nabla g(\mathbf{v}_{k-1})\| \leq \|g\|_{C^2}(1 + L) \tag{136}$$

for all $k$. Now by Corollary 3, for every even moment $2p < d$, there exists $C(2p, \mathbf{K}) > 0$ such that

$$\mathbb{P}\left(|\Delta\mathcal{M}_k^{lin}| \geq \frac{d^\delta}{d}\right) \leq \frac{C(2p, \mathbf{K})\mathbb{E}\left[\|\nabla g(\mathbf{v}_{k-1})\|^{4p}\right]^{1/2} d^{2p/3}}{d^{2\delta p}}$$

$$\leq C(2p, \mathbf{K})(1 + L)^{2p} d^{2p\left(\frac{1}{3}-\delta\right)}. \tag{137}$$

$\square$

**Lemma 10.** *For all $g \in Q$ and $0 < s < 1/2$,*

$$\sup_{0 \leq k \leq \lfloor Td \rfloor} |\mathcal{M}_k^{quad}| < \frac{1}{d^s} \tag{138}$$

*with overwhelming probability.*

*Proof.* From Cauchy-Schwarz, we see that

$$\left| \Delta \mathcal{M}_k^{quad} \right| \leq \frac{\eta^2}{d} \|g\|_{C^2} \tag{139}$$

Then, Azuma's inequality shows that

$$\mathbb{P}\left( \max_{1 \leq k \leq \lfloor Td \rfloor} |\mathcal{M}_k^{quad}| \geq \frac{1}{d^s} \right) \leq 2 \exp\left( \frac{-d^{-2s+1}}{C_T \eta^4 \|g\|_{C^2}^2} \right), \tag{140}$$

which gives the result. $\qquad\square$

**Lemma 11.** *For all $g \in Q$ and $s < 1$,*

$$\sup_{0 \leq t \leq T} |\mathcal{M}_t^\sigma| \leq \frac{1}{d^s}, \tag{141}$$

*with overwhelming probability.*

*Proof.* From Equation (101), we know that

$$\mathcal{M}_t^\sigma = \eta \int_0^t \nabla q(\mathbf{V}_s)^{\mathrm{T}} \sqrt{\frac{2\mathbf{K}_\sigma}{d\pi}} \, d\mathbf{B}_t. \tag{142}$$

Using the $\|q\|_{C^2}$ norm we can bound

$$\|\nabla g(\mathbf{V}_s)\| \leq \|g\|_{C^2} \left( 1 + \|\mathbf{V}_s\| \right). \tag{143}$$

Then, with Assumption 3 and Equation (143) we can bound the quadratic variation as,

$$\begin{aligned}
\langle \mathcal{M}^\sigma \rangle_t &= \frac{2\eta^2}{d\pi} \int_0^t \nabla g(\mathbf{V}_s^\tau)^{\mathrm{T}} \mathbf{K}_\sigma \nabla g(\mathbf{V}_s^\tau) \, ds \\
&\leq \frac{2\eta^2}{d\pi} \int_0^t \|\mathbf{K}_\sigma\| \, \|\nabla g(\mathbf{V}_s)\|^2 \, ds \\
&\leq \frac{2\eta^2}{d\pi} \|\mathbf{K}_\sigma\| \, \|g\|_{C^2}^2 \left( 1 + M \right)^2 t.
\end{aligned} \tag{144}$$

Then, using the subgaussian tail bound for continuous martingales we see that the stopped martingale satisfies,

$$\mathbb{P}\left( \sup_{0 \leq t \leq T} |\mathcal{M}_t^\sigma| \geq t \right) \leq 2 \exp\left( \frac{-t^2 d\pi}{4\eta^2 \|\mathbf{K}_\sigma\| \, \|g\|_{C^2}^2 (1 + M)^2 T} \right). \tag{145}$$

$\qquad\square$

**Lemma 12.** *If $\mu$ is a probability measure on $\mathbb{R}$ with the property that there exists $a_0 > 0$ such that $\frac{\mathrm{d}\mu}{\mathrm{d}x} = g(x)$ on $[-a_0, a_0]$ for $g \in C^2([-a_0, a_0])$, then the map $\alpha : \mathbb{R}^+ \to \mathbb{R}$ defined by*

$$s \mapsto \frac{1}{\sqrt{s}} \int_{\mathbb{R}} \exp\left(\frac{-y^2}{2s}\right) \mathrm{d}\mu(y), \tag{146}$$

*is bounded as well as Lipschitz.*

*Proof.* Notice that it suffices to show (146) is bounded and Lipschitz for $0 < s < 1$. Let $f_s(y) = \frac{2}{\pi\sqrt{s}} \exp\left(\frac{-y^2}{2s}\right)$, as well as $G(y) = \mu((-\infty, y])$. Decomposing the integral into

$$\int_{\mathbb{R}} f_s(y) \, \mathrm{d}\mu(y) = \int_{[-a_0, a_0]} f_s(y) \, \mathrm{d}\mu(y) + \int_{\mathbb{R} \setminus [-a_0, a_0]} f_s(y) \, \mathrm{d}\mu(y), \tag{147}$$

we see that the latter term may be easily bounded by

$$\int_{\mathbb{R} \setminus [-a_0, a_0]} f_s(y) \, \mathrm{d}\mu(y) \leq \frac{1}{\sqrt{s}} \exp\left(\frac{-a_0^2}{2s}\right), \tag{148}$$

which decays to $0$ as $s \to 0$. The former term we apply the integration by parts formula to get

$$\int_{[-a_0, a_0]} f_s(y) \, \mathrm{d}\mu(y) = f_s(y)G(y) \Big|_{y=-a_0}^{y=a_0} + \int_{[-a_0, a_0]} \frac{y}{s^{3/2}} \exp\left(\frac{-y^2}{2s}\right) G(y) \, \mathrm{d}y. \tag{149}$$

Further decomposing the latter integral into positive and negative regions we get

$$\int_0^{a_0} \frac{y}{s^{3/2}} \exp\left(\frac{-y^2}{2s}\right) G(y) \, \mathrm{d}y = \int_0^{a_0} \frac{y}{s^{3/2}} \exp\left(\frac{-y^2}{2s}\right) [G(-a_0) + \mu((-a_0, y])] \, \mathrm{d}y, \tag{150}$$

and

$$\int_{-a_0}^0 \frac{y}{s^{3/2}} \exp\left(\frac{-y^2}{2s}\right) G(y) \, \mathrm{d}y = \int_{-a_0}^0 \frac{y}{s^{3/2}} \exp\left(\frac{-y^2}{2s}\right) [G(-a_0) + \mu((-a_0, y])] \, \mathrm{d}y \tag{151}$$

$$= -\int_0^{a_0} \frac{y}{s^{3/2}} \exp\left(\frac{-y^2}{2s}\right) [G(-a_0) + \mu((-a_0, -y])] \, \mathrm{d}y. \tag{152}$$

Thus,

$$\int_{[-a_0, a_0]} \frac{y}{s^{3/2}} \exp\left(\frac{-y^2}{2s}\right) G(y) \, \mathrm{d}y = \int_0^{a_0} \frac{y}{s^{3/2}} \exp\left(\frac{-y^2}{2s}\right) \mu((-y, y]) \, \mathrm{d}y \tag{153}$$

$$\leq C \int_0^{a_0} \frac{y^2}{s^{3/2}} \exp\left(\frac{-y^2}{2s}\right) \, \mathrm{d}y \tag{154}$$

$$= C \int_0^{a_0/\sqrt{s}} y^2 \exp\left(\frac{-y^2}{2}\right) \, \mathrm{d}y. \tag{155}$$

Putting this all together, we conclude that $\varphi(s)$ is uniformly bounded for all $s > 0$. To see Lipschitz, we apply a similar argument. We first differentiate $f_s(y)$ with respect to $s$ to we get

$$\frac{\mathrm{d}}{\mathrm{d}s} f_s(y) = \frac{1}{2s^{5/2}} \exp\left(\frac{-y^2}{2s}\right) (y^2 - s). \tag{156}$$

Therefore,

$$\alpha'(s) = \int_{[-a_0, a_0]} \frac{\mathrm{d}}{\mathrm{d}s} f_s(y) \, \mathrm{d}\mu(y) + \int_{\mathbb{R} \setminus [-a_0, a_0]} \frac{\mathrm{d}}{\mathrm{d}s} f_s(y) \, \mathrm{d}\mu(y). \tag{157}$$

There exists $s_0 > 0$ such that if $s < s_0$, then $\sqrt{3s} < a_0$. It is easy to check that if $y > \sqrt{3s}$ then $\frac{d}{ds} f_s(y)$ is decreasing in $y$. Likewise, if $y < -\sqrt{3s}$ then $\frac{d}{ds} f_s(y)$ is increasing in $y$. It follows that

$$\int_{\mathbb{R} \backslash [-a_0, a_0]} \frac{d}{ds} f_s(y) \, d\mu(y) \leq \frac{1}{s^{5/2}} \exp\left(\frac{-a_0^2}{2s}\right) (a_0^2 - s), \tag{158}$$

which decays to $0$ as $s \to 0$. Finally, we apply integration by parts once more to get

$$\int_{[-a_0, a_0]} \frac{d}{ds} f_s(y) = \frac{d}{ds} f_s(y) G(y) \Big|_{y=-a_0}^{y=a_0} \tag{159}$$

$$+ \int_0^{a_0} \frac{1}{2s^{7/2}} \exp\left(\frac{-y^2}{2s}\right) (y^3 - 3sy) \mu((-y, y]) \, dy. \tag{160}$$

Since $g \in C^2([-a_0, a_0])$ we may express $\mu((-y, y])$ as

$$\mu((-y, y]) = \int_{-y}^{y} g(0) + g'(0)x + O(x^2) \, dx = 2g(0)y + O(y^3). \tag{161}$$

Plugging this into (160) it is easy to check that

$$\left| \int_0^{a_0} \frac{g(0)}{s^{7/2}} \exp\left(\frac{-y^2}{2s}\right) (y^3 - 3sy)y \, dy \right| = \frac{g(0)a_0^3 \exp\left(\frac{-a_0^2}{2s}\right)}{s^{5/2}}, \tag{162}$$

and

$$\left| \int_0^{a_0} \frac{1}{2s^{7/2}} \exp\left(\frac{-y^2}{2s}\right) (y^3 - 3sy)O(y^3) \, dy \right| \leq C \int_0^{a_0} \frac{1}{2s^{7/2}} \exp\left(\frac{-y^2}{2s}\right) (y^3 + 3sy)y^3 \, dy \tag{163}$$

$$= C \int_0^{a_0/\sqrt{s}} \exp\left(-\frac{y^2}{2}\right) (y^3 + 3y)y^3 \, dy. \tag{164}$$

Combining this with (158), we conclude that $|\alpha'(s)|$ is uniformly bounded for all $s > 0$. $\qquad \square$

**Lemma 13.** *Let $x \sim N(0, \mathbf{K})$ such that $\mathbf{K}$ is positive-definite. If $a \in \mathbb{R}^d$, then for all even moments $2k \leq d$,*

$$\mathbb{E}\left[\langle a, \sigma(\mathbf{x}) \rangle^{2k}\right] \leq C(2k, \mathbf{K}) \|a\|_\infty^{2k} d^{4k/3}, \tag{165}$$

*where $C(2k, \mathbf{K}) > 0$ depends only on $2k$, $\lambda_{\min}(\mathbf{K})$ and $\lambda_{\max}(\mathbf{K})$.*

*Proof.* We start by fixing a $\delta > 0$ and defining the smooth approximation of $\sigma(\mathbf{x})$ to be $\sigma_\delta(\mathbf{x}) = \rho_\delta * \sigma(\mathbf{x})$, where $\rho_\delta : \mathbb{R} \to \mathbb{R}$ is the standard compactly-supported mollifier convolved entry-wise to $\sigma(\mathbf{x})$, i.e. $(\sigma_\delta(\mathbf{x}))_i = \rho_\delta * \sigma(\mathbf{x}_i)$. It follows that

$$\| \langle a, \sigma(\mathbf{x}) \rangle \|_{L^{2k}} \leq \| \langle a, \sigma(\mathbf{x}) - \sigma_\delta(\mathbf{x}) \rangle \|_{L^{2k}} + \| \langle a, \sigma_\delta(\mathbf{x}) \rangle \|_{L^{2k}}. \tag{166}$$

Note that $\rho_\delta$ has support contained in $[-\delta, \delta]$, thus the entry-wise difference of $\sigma(\mathbf{x}) - \sigma_\delta(\mathbf{x})$ may be bounded by

$$|\sigma(\mathbf{x}_i) - \sigma_\delta(\mathbf{x}_i)| \leq \begin{cases} 0 & |\mathbf{x}_i| > 2\delta \\ 2 & |\mathbf{x}_i| \leq 2\delta. \end{cases}$$

Define $N_\delta(\mathbf{x}) = \sum_{i=1}^{d} \mathbb{1}_{\{|\mathbf{x}_i| \leq 2\delta\}}$ to be the number of coordinates of $\mathbf{x}$ within the interval $(-2\delta, 2\delta)$, we see that

$$\mathbb{E}\left[\langle a, \sigma(\mathbf{x}) - \sigma_\delta(\mathbf{x}) \rangle^{2k}\right] \leq \|a\|_\infty^{2k} 2^{2k} \mathbb{E}\left[N_\delta(\mathbf{x})^{2k}\right] \tag{167}$$

$$= \|a\|_\infty^{2k} 2^{2k} \sum_{s \in \mathcal{I}} \mathbb{P}\left((\mathbf{x}_i)_{i \in s'} \in [-2\delta, 2\delta]^{|s'|}\right), \tag{168}$$

where $\mathcal{I} = \{1, \ldots, d\}^{2k}$ and $s'$ is set of distinct elements of $s$. Let $\mathbf{K}^{(s')} = \mathbb{E}\left[(\mathbf{x}_i)_{i \in s'}^{\otimes 2}\right]$, then $(\mathbf{x}_i)_{i \in s'} \sim N(0, \mathbf{K}^{(s')})$. Recall that there exists a permutation matrix $\mathbf{P}$, such $\mathbf{K}^{(s')}$ forms the top $|s'| \times |s'|$ sub-matrix of $\mathbf{PKP}^{-1}$. Given that $\mathbf{PKP}^{-1}$ and $\mathbf{K}$ are similar, they share the same eigenvalues. Therefore, by the Cauchy interlacing-law,

$$\lambda_{\min}(\mathbf{K}) \leq \lambda_{\min}(\mathbf{K}^{(s')}). \tag{169}$$

In particular this implies

$$\det \mathbf{K}^{(s')} = \prod_{i=1}^{|s'|} \lambda_i(\mathbf{K}^{(s')}) \geq \lambda_{\min}(\mathbf{K})^{|s'|}. \tag{170}$$

Plugging this back into (168) and choosing $\delta = d^{-r}$, we get

$$\mathbb{E}\left[\langle a, \sigma(\mathbf{x}) - \sigma_\delta(\mathbf{x}) \rangle^{2k}\right] \leq \|a\|_\infty^{2k} 2^{2k} \sum_{s \in \mathcal{I}} \frac{(4\delta)^{|s'|}}{(2\pi \lambda_{\min}(\mathbf{K}))^{|s'|/2}} \tag{171}$$

$$= \|a\|_\infty^{2k} 2^{2k} \sum_{l=1}^{2k} \binom{d}{l} l! \left\{\begin{matrix} 2k \\ l \end{matrix}\right\} \frac{(4\delta)^l}{(2\pi \lambda_{\min}(\mathbf{K}))^{l/2}} \tag{172}$$

$$\leq \|a\|_\infty^{2k} 2^{2k} \max_{1 \leq l \leq 2k} \left(l! \left\{\begin{matrix} 2k \\ l \end{matrix}\right\}\right) \sum_{l=1}^{2k} \left(\frac{ed}{l}\right)^l \frac{(4\delta)^l}{(2\pi \lambda_{\min}(\mathbf{K}))^{l/2}} \tag{173}$$

$$\leq \|a\|_\infty^{2k} 2^{2k} C(2k) \left(\frac{4ed^{1-r}}{\sqrt{2\pi \min\{\lambda_{\min}(\mathbf{K}), 1\}}}\right)^{2k} \tag{174}$$

$$= \|a\|_\infty^{2k} C(2k, \mathbf{K}) d^{(1-r)2k}, \tag{175}$$

where $\left\{\begin{smallmatrix} 2k \\ l \end{smallmatrix}\right\}$ is Stirling's number of a second-kind. For clarity of notation moving forward, we note that $C(2k, \mathbf{K})$ may change up to factors of constants or powers of $k$ from line to line, while always being independent to $d$.

To control the second term of Equation (166), we modify the proof of concentration of Lipschitz functions of Gaussian random variables. See Lemma 2.1.5 for more details (Adler & Taylor, 2007). Let $G(\mathbf{x}) = \langle a, \sigma_\delta(\mathbf{x}) \rangle$. and $\mathbf{z} \sim N(0, \mathbf{K})$ be independent to $\mathbf{x}$. Define the Gaussian interpolation $\mathbf{z}^\alpha$ to be

$$\mathbf{z}^{(\alpha)} = \alpha \mathbf{x} + \sqrt{1 - \alpha^2} \mathbf{z},$$

and note that $\mathbf{x} \overset{\text{law}}{=} \mathbf{z}^{(\alpha)}$ for all $\alpha \in [0, 1]$. Then by Lemma 2.1.4 in (Adler & Taylor, 2007),

$$\mathbb{E}[G(\mathbf{x})^{2k}] = (2k-1) \int_0^1 \mathbb{E}\left[\left\langle \mathbf{K}(a \odot \sigma_\delta'(\mathbf{x})), a \odot \sigma_\delta'(\mathbf{z}^{(\alpha)}) \right\rangle \cdot G^{2k-2}(\mathbf{x})\right] d\alpha, \tag{176}$$

where $\odot$ represents the Hadamard product. Going forward, we will use Hölder's inequality to break up the expectation and form a recursive equation. As such, consider

$$\mathbb{E}[\langle \mathbf{K}(a \odot \sigma_\delta'(\mathbf{x}), a \odot \sigma_\delta'(\mathbf{y}^\alpha) \rangle^{2p}], \tag{177}$$

for some $p$. Standard linear algebra gives us

$$\mathbb{E}[\left\langle \mathbf{K}(a \odot \sigma_\delta'(\mathbf{x}), a \odot \sigma_\delta'(\mathbf{z}^{(\alpha)}) \right\rangle^{2p}] \leq (\|\mathbf{K}\| \|a\|_\infty^2)^{2p} \mathbb{E}[(\|\sigma_\delta'(\mathbf{x})\| \|\sigma_\delta'(\mathbf{z}^{(\alpha)})\|)^{2p}]$$
$$\leq (\|\mathbf{K}\| \|a\|_\infty^2)^{2p} \mathbb{E}[\|\sigma_\delta'(\mathbf{x})\|^{4p}], \tag{178}$$

with the last line following from Cauchy-Schwartz and equality in law of $\mathbf{x}$ and $\mathbf{z}^{(\alpha)}$. Given that $\sigma_\delta'(\mathbf{x}_i) = 0$ for $\mathbf{x}_i \notin [-2\delta, 2\delta]$, as well as $|\sigma_\delta'(\mathbf{x}_i)| \leq \frac{L_\rho}{\delta}$ for $\mathbf{x}_i \in [-2\delta, 2\delta]$ and $L_\rho$ a universal constant depending on our mollifier,

$$\mathbb{E}[\|\sigma'_\delta(\mathbf{x})\|\|^{4p}] \le \frac{L_\rho^{4p}}{\delta^{4p}}\mathbb{E}[N_\delta^{2p}]. \tag{179}$$

As we have seen in Equation (175),

$$\mathbb{E}[N_\delta(\mathbf{x})^{2p}] \le C(p, \mathbf{K})d^{(1-r)2p}. \tag{180}$$

Therefore, up to absolute constants

$$\mathbb{E}[\langle \mathbf{K}(a \odot \sigma'(\mathbf{x}), a \odot \sigma'(\mathbf{y}^\alpha)\rangle^{2p}] \le C(p, \mathbf{K})(\|a\|_\infty^2)^{2p}\frac{L_\rho^{4p}}{\delta^{4p}}d^{(1-r)2p} \tag{181}$$

$$\le C(p, \mathbf{K})(\|a\|_\infty^2)^{2p}d^{(1+r)2p}.$$

Returning to (176) and choosing $2p = 2k - 1$, we see by Hölder's inequality

$$\mathbb{E}[G(\mathbf{x})^{2k}] \le (2k-1)\left\|G(\mathbf{x})^{2k-2}\right\|_{L^{\frac{2k-1}{2k-2}}}\|[\langle \mathbf{K}(a \odot \sigma'_\delta(\mathbf{x}), a \odot \sigma'_\delta(\mathbf{y}^\alpha)\rangle\|_{L^{2k-1}} \tag{182}$$

$$\le (2k-1)\mathbb{E}[G(\mathbf{x})^{2k-1}]^{\frac{2k-2}{2k-1}}C(k, \mathbf{K})\|a\|_\infty^2 d^{1+r}. \tag{183}$$

Iterating the same inequalities as above for $\mathbb{E}[G^{2k-1}]$, we obtain

$$\mathbb{E}[G(\mathbf{x})^{2k}] \le \prod_{i=1}^{2k-1}\left((2k-i)C(2k-i, \mathbf{K})\|a\|_\infty^2 d^{1+r}\right)^{\frac{2k-i}{2k-1}} \tag{184}$$

$$\le C(2k, \mathbf{K})\|a\|_\infty^{2k} d^{(1+r)k}. \tag{185}$$

Equations (175) and (184)combined give control over Equation (166),

$$\|\langle a, \sigma(\mathbf{x})\rangle\|_{L^{2k}} \le C(2k, \mathbf{K})\|a\|_\infty \left(d^{\frac{1+r}{2}} + d^{1-r}\right). \tag{186}$$

Optimizing over $r$ yields $r = 1/3$ leading to

$$\mathbb{E}\left[\langle a, \sigma(\mathbf{x})\rangle\|^{2k}\right] \le C(2k, \mathbf{K})\|a\|_\infty^{2k}d^{4k/3}, \tag{187}$$

as desired. $\qquad\square$

**Lemma 14.** *Let* $\mathbf{x} \sim N(0, \mathbf{K})$ *such* $\mathbf{K}$ *is positive-definite. If* $\mathbf{y} \in \mathbb{R}^d$ *a random vector independent to* $\mathbf{x}$*, then for all even moments* $2k \le d$*,*

$$\mathbb{E}\left[\langle \mathbf{y}, \sigma(\mathbf{x})\rangle^{2k}\right] \le C(2k, \mathbf{K})\mathbb{E}\left[\|\mathbf{y}\|^{4k}\right]^{1/2}d^{2k/3}, \tag{188}$$

*where* $C(2k, \mathbf{K}) > 0$ *depends only on* $2k$*,* $\lambda_{\min}(\mathbf{K})$ *and* $\lambda_{\max}(\mathbf{K})$*.*

*Proof.* The proof is almost identical to that of Lemma 13, but instead of taking the sup-norm of $a$ in (168), we take the $l_2$ norm via the Cauchy Schwarz inequality. Now proceeding proceeding in a similar fashion we get

$$\mathbb{E}\left[\langle \mathbf{y}, \sigma(\mathbf{x}) - \sigma_\delta(\mathbf{x})\rangle^{2k}\right] \le \mathbb{E}\left[\|\mathbf{y}\|^{2k}\|\sigma(\mathbf{x}) - \sigma_\delta(\mathbf{x})\|^{2k}\right] \tag{189}$$

$$\le \mathbb{E}\left[\|\mathbf{y}\|^{4k}\right]^{1/2}\mathbb{E}\left[\|\sigma(\mathbf{x}) - \sigma_\delta(\mathbf{x})\|^{4k}\right]^{1/2} \tag{190}$$

$$\le \mathbb{E}\left[\|\mathbf{y}\|^{4k}\right]^{1/2}2^{2k}\left(\mathbb{E}\left[N_\delta(\mathbf{x})^{2k}\right]\right)^{1/2} \tag{191}$$

$$\le \mathbb{E}\left[\|\mathbf{y}\|^{4k}\right]^{1/2}C(k, \mathbf{K})d^{(1-r)k}. \tag{192}$$

Lastly, by the independence of $\mathbf{y}$ and $\mathbf{x}$, upon conditioning on $\mathbf{y}$ we see by Gaussian-concentration on $\langle \mathbf{y}, \sigma_\delta(\mathbf{x}) \rangle$ that

$$\mathbb{E}\left[ \langle \mathbf{y}, \sigma_\delta(\mathbf{x}) \rangle^{2k} \right] = \mathbb{E}\left[ \mathbb{E}\left[ \langle \mathbf{y}, \sigma_\delta(\mathbf{x}) \rangle^{2k} \Big| \mathbf{y} \right] \right] \tag{193}$$

$$\leq \mathbb{E}\left[ \left( \frac{C\sqrt{2k}\,\|\mathbf{y}\|\,\sqrt{\lambda_{\max}(\mathbf{K})}}{\delta} \right)^{2k} \right] \tag{194}$$

$$= C(2k, \mathbf{K})\mathbb{E}\left[ \|\mathbf{y}\|^{2k} \right] d^{2kr}, \tag{195}$$

where $C > 0$ is an absolute constant. Combining (192) and (195) then optimizing in $r > 0$ yields the result. $\qquad\square$

**Corollary 1.** *Let* $\mathbf{a} \in \mathbb{R}^d$ *and* $\mathbf{x}_{k+1} \sim N(0, \mathbf{K})$, *then for all even moments* $2p \leq d$, *there exists* $C(2p, \mathbf{K}) > 0$ *such that*

$$\mathbb{P}\left( |\langle \mathbf{a}, \sigma(\ell_{k+1}\mathbf{x}_{k+1}) - \mathbb{E}\left[\sigma(\ell_{k+1}\mathbf{x}_{k+1})|\mathcal{F}_k\right] \rangle | \geq t \right) \leq \frac{C(2p, \mathbf{K})\,\|\mathbf{a}\|_\infty^{2p}\,d^{4p/3}}{t^{2p}}. \tag{196}$$

*Proof.* For notational clarity, let us denote $Y = \langle \mathbf{a}, \sigma(\ell_{k+1}\mathbf{x}_{k+1}) \rangle$. By Jensen's inequality and convexity of $x \mapsto x^{2p}$,

$$\mathbb{E}\left[ |Y - \mathbb{E}[Y|\mathcal{F}_k]|^{2p} \right] \leq 2^{2p}\mathbb{E}\left[ \frac{1}{2}Y^{2p} + \frac{1}{2}\mathbb{E}[Y|\mathcal{F}_k]^{2p} \right] \tag{197}$$

$$\leq 2^{2p}\mathbb{E}\left[ Y^{2p} \right]. \tag{198}$$

However, notice that $\mathbb{E}\left[ Y^{2p} \right] = \mathbb{E}\left[ \langle \mathbf{a}, \sigma(\mathbf{x}_{k+1}) \rangle^{2p} \right]$. By Markov's inequality and Lemma 13,

$$\mathbb{P}\left( |Y - \mathbb{E}[Y|\mathcal{F}_k]| \geq t \right) = \mathbb{P}\left( |Y - \mathbb{E}[Y|\mathcal{F}_k]|^{2p} \geq t^{2p} \right) \tag{199}$$

$$\leq \frac{2^{2p}\mathbb{E}\left[ \langle \mathbf{a}, \sigma(\mathbf{x}_{k+1}) \rangle^{2p} \right]}{t^{2p}} \tag{200}$$

$$\leq \frac{C(2p, \mathbf{K})\,\|\mathbf{a}\|_\infty^{2p}\,d^{4p/3}}{t^{2p}}. \tag{201}$$

$\qquad\square$

**Corollary 2.** *If* $\mathbf{a} \in \mathbb{R}^d$ *such that* $\max_{2 \leq i \leq d} |a^i| = O\left( \frac{d^\delta}{\sqrt{d}} \right)$, *then for all even moments* $2p \leq d$ *and* $s > 0$, *there exists* $C(2p, \mathbf{K}) > 0$ *such that,*

$$\mathbb{P}\left( |\langle \mathbf{a}, \sigma(\ell_{k+1}\mathbf{x}_{k+1}) - \mathbb{E}\left[\sigma(\ell_{k+1}\mathbf{x}_{k+1})|\mathcal{F}_k\right] \rangle | \geq d^s \right) \leq C(2p, \mathbf{K})d^{p\left(\frac{1}{3} - 2s + 2\delta\right)}, \tag{202}$$

*provided that* $d^s > 4|a^1|$.

*Proof.* For ease of notation, let $\sigma_{k+1} = \sigma(\ell_{k+1}\mathbf{x}_{k+1})$. Given that $|a^1(\sigma_{k+1} - \mathbb{E}[\sigma_{k+1}|\mathcal{F}_k])| \leq 2|a^1|$, it follows by Corollary 1,

$$\mathbb{P}\left( |\langle \mathbf{a}, \sigma_{k+1} - \mathbb{E}[\sigma_{k+1}|\mathcal{F}_k] \rangle | \geq d^s \right) \leq \mathbb{P}\left( \left| \sum_{i=2}^d a^i \left( \sigma_{k+1}^i - \mathbb{E}\left[ \sigma_{k+1}^i \Big| \mathcal{F}_k \right] \right) \right| \geq \frac{d^s}{2} \right) \tag{203}$$

$$\leq \frac{C(2p, \mathbf{K})\left( \max_{2 \leq i \leq d} |a^i| \right)^{2p} d^{4p/3}}{d^{2ps}} \tag{204}$$

$$\leq C(2p, \mathbf{K})d^{p\left(\frac{1}{3} - 2s + 2\delta\right)}. \tag{205}$$

$\qquad\square$

**Corollary 3.** *If* $\mathbf{x}_{k+1} \sim N(0, \mathbf{K})$ *and* $\mathbf{y} \in \mathbb{R}^d$ *a random vector independent to* $\mathbf{x}$, *then for all even moments* $2p \leq d$, *there exists* $C(2p, \mathbf{K}) > 0$ *and independent to* $d$ *such that*

$$\mathbb{P}\left( |\langle \mathbf{y}, \sigma(\ell_{k+1}\mathbf{x}_{k+1}) - \mathbb{E}\left[\sigma(\ell_{k+1}\mathbf{x}_{k+1})|\mathcal{F}_k\right] \rangle | \geq t \right) \leq \frac{C(2p, \mathbf{K})\mathbb{E}\left[ \|\mathbf{y}\|^{4p} \right]^{1/2} d^{2p/3}}{t^{2p}}. \tag{206}$$

The proof is identical to that of Corollary 1 but using Lemma 14 instead.

**Lemma 15.** *Let $M_k$ be a martingale such that $|M_k - M_{k-1}| \leq C$ almost-surely and let $S_k \leq C$. Then for all $t > 0$ and $N > 0$,*

$$\mathbb{P}\left(|M_N| \geq t\right) \leq 2 \exp\left(-\frac{t^2}{2\left(C^2 + \sum_{k=1}^{N-1} S_k^2\right)}\right) + \mathbb{P}\left(\exists\, k \leq N - 1\,, |M_k - M_{k-1}| > S_k\right). \tag{207}$$

*Proof.* Let $\tau = \min\{k \,; |M_k - M_{k-1}| > S_k\}$ and $Y_k = M_{k \wedge \tau}$, then on the event that $\{k < \tau\}$, $|Y_k - Y_{k-1}| \leq S_k$. On the other hand, if $\{\tau \leq k\}$ then $|Y_k - Y_{k-1}| \leq C$. Breaking the probability space into the event $\{\tau \leq N - 1\}$ and its complement gives,

$$\mathbb{P}\left(|M_N| \geq t\right) \leq \mathbb{P}\left(|Y_N| \geq t\right) + \mathbb{P}\left(\tau \leq N - 1\right). \tag{208}$$

Azuma's inequality completes the proof as

$$\mathbb{P}\left(|Y_N| \geq t\right) \leq 2 \exp\left(-\frac{t^2}{2\left(C^2 + \sum_{k=1}^{N-1} S_k^2\right)}\right). \tag{209}$$

$\square$

# B. Risk Curve Dynamics

### B.1. Proof of Theorem 2

If $\mathbf{V}_t = \mathbf{\Theta}_t - \boldsymbol{\theta}_*$ where $\mathbf{\Theta}_t$ solves the SDE given by (8) then Itô's lemma applied onto

$$q(x) = \frac{1}{2} x^T \mathbf{K} \mathbf{R}(z; \overline{\mathbf{K}}) x, \tag{210}$$

yields

$$dq(\mathbf{V}_t) = \left( -\frac{\eta \varphi(\mathcal{R}_t)}{\sqrt{2\mathcal{R}_t}} \mathbf{V}_t^T \left( \frac{\mathbf{K} \mathbf{R}(z; \overline{\mathbf{K}}) + \mathbf{R}(z; \overline{\mathbf{K}})^T \mathbf{K}}{2} \right) \overline{\mathbf{K}} \mathbf{V}_t \right) dt \tag{211}$$

$$+ \left( \frac{\eta^2}{\pi d} \left\langle \mathbf{K} \mathbf{R}(z; \overline{\mathbf{K}}), \mathbf{K}_\sigma \right\rangle \right) dt + d\mathcal{M}_t^\sigma, \tag{212}$$

where we denote $\mathcal{R}_t = \mathcal{R}(\mathbf{V}_t)$ for ease of notation. By resolvent identities, we know that

$$\mathbf{K} \mathbf{R}(z; \overline{\mathbf{K}}) \overline{\mathbf{K}} = z \mathbf{K} \mathbf{R}(z; \overline{\mathbf{K}}) + \mathbf{K}. \tag{213}$$

Moreover,

$$\mathbf{R}(z; \overline{\mathbf{K}})^T \mathbf{K} \overline{\mathbf{K}} = (\mathbf{K} \overline{\mathbf{K}} \mathbf{R}(z; \overline{\mathbf{K}}))^T = (z \mathbf{K} \mathbf{R}(z; \overline{\mathbf{K}}) + \mathbf{K})^T, \tag{214}$$

so

$$2 \left( z q(\mathbf{V}_t) + \mathcal{R}_t \right) = \mathbf{V}_t^T \left( \frac{\mathbf{K} \mathbf{R}(z; \overline{\mathbf{K}}) + \mathbf{R}(z; \overline{\mathbf{K}})^T \mathbf{K}}{2} \right) \overline{\mathbf{K}} \mathbf{V}_t. \tag{215}$$

Returning to Itô's we see that

$$dq(\mathbf{V}_t) = \left( -\frac{2\eta \varphi(\mathcal{R}_t)}{\sqrt{2\mathcal{R}_t}} \left( z q(\mathbf{V}_t) + \mathcal{R}_t \right) + \frac{\eta^2}{\pi d} \operatorname{Tr} \left( \mathbf{K} \mathbf{R}(z; \overline{\mathbf{K}}) \mathbf{K}_\sigma \right) \right) dt + d\mathcal{M}_t^\sigma. \tag{216}$$

To recover the risk $\mathcal{R}_t$, we once again turn towards the Cauchy-integral law as well as the Spectral Theorem. Indeed,

$$\overline{\mathbf{K}} = \sum_{i=1}^d \lambda_i(\overline{\mathbf{K}}) \mathbf{u}_i \otimes \mathbf{w}_i \qquad\qquad \mathbf{R}(z; \overline{\mathbf{K}}) = \sum_{i=1}^d \frac{1}{\lambda_i(\overline{\mathbf{K}}) - z} \mathbf{u}_i \otimes \mathbf{w}_i, \tag{217}$$

where $\mathbf{u}_i$ and $\mathbf{w}_i$ are left and right eigenvectors respectively of $\overline{\mathbf{K}}$. We may then write

$$q(\mathbf{V}_t) = \frac{1}{2} \sum_{i=1}^d \frac{1}{\lambda_i(\overline{\mathbf{K}}) - z} \mathbf{V}_t^T (\mathbf{K} \mathbf{u}_i \otimes \mathbf{w}_i) \mathbf{V}_t. \tag{218}$$

Denoting $\widetilde{r}_i(t) = \frac{1}{2} \mathbf{V}_t^T (\mathbf{K} \mathbf{u}_i \otimes \mathbf{w}_i) \mathbf{V}_t$, then upon integrating over $\Gamma_i$, a closed curve enclosing only $\lambda_i(\overline{\mathbf{K}})$, we see that

$$d\widetilde{r}_i = \oint_{\Gamma_i} \frac{dq(\mathbf{V}_t)}{-2\pi i} \, dz \tag{219}$$

$$= \left( -\frac{2\eta \varphi(\mathcal{R}_t)}{\sqrt{2\mathcal{R}_t}} \lambda_i(\overline{\mathbf{K}}) \widetilde{r}_i + \frac{\eta^2}{\pi d} \operatorname{Tr} \left( \mathbf{K} (\mathbf{u}_i \otimes \mathbf{w}_i) \mathbf{K}_\sigma \right) \right) dt + d\mathcal{M}_t^{i,\sigma}, \tag{220}$$

where

$$\mathcal{M}_t^{i,\sigma} = \oint_{\Gamma_i} \frac{\mathcal{M}_t^\sigma}{-2\pi i} \, dz. \tag{221}$$

Given that the martingale terms vanish as $d \to \infty$, we should expect the drift term of $\widetilde{r}_i$ to dominate. Thus, let us define the deterministic equivalent of $\widetilde{r}_i$ as

$$dr_i = \left( -\frac{2\eta \varphi(2R_t)}{\sqrt{R_t}} \lambda_i(\overline{\mathbf{K}}) r_i + \frac{\eta^2}{\pi d} \operatorname{Tr} \left( \mathbf{K} (\mathbf{u}_i \otimes \mathbf{w}_i) \mathbf{K}_\sigma \right) \right) dt, \quad r_i(0) = \widetilde{r}_i(0) \tag{222}$$

where

$$R_t = \sum_{i=1}^{d} r_i. \tag{223}$$

Note, if we integrate (218) around $\Gamma$, we obtain

$$\mathcal{R}_t = \sum_{i=1}^{d} \widetilde{r}_i. \tag{224}$$

We will show that $\mathcal{R}_t$ concentrates around $R_t$ using the same idea as in Lemma 7. We start by considering a set of functions that maps $\mathbf{x} \in \mathbb{R}^d$ to $\mathbb{C}$ by

$$\mathcal{W} = \left\{ \mathcal{J}(z)^{\mathrm{T}} \mathbf{x} \, ; \, z \in \Gamma \right\}, \tag{225}$$

where $\mathcal{J}(z) \in \mathbb{C}^d$ defined coordinate-wise by

$$[\mathcal{J}(z)]_i = \frac{1}{\lambda_i(\overline{\mathbf{K}}) - z}. \tag{226}$$

Let $r(t) = [r_i(t)]_{i=1}^{d}$, $\widetilde{r}(t) = [\widetilde{r}_i(t)]_{i=1}^{d}$ and $g(\mathbf{x}) = \mathcal{J}(z)^{\mathrm{T}} \mathbf{x}$ for some $z \in \Gamma$, then as in (122) it is easy to check that

$$|g(r(t)) - g(\widetilde{r}(t))| \leq \int_0^t \left| \frac{2\eta\varphi(R_s)}{\sqrt{R_s}} \sum_{i=1}^{d} \lambda_i(\overline{\mathbf{K}}) \left[\mathcal{J}(z)\right]_i r_i - \frac{2\eta\varphi(\mathcal{R}_s)}{\sqrt{\mathcal{R}_s}} \sum_{i=1}^{d} \lambda_i(\overline{\mathbf{K}}) \left[\mathcal{J}(z)\right]_i \widetilde{r}_i \right| \mathrm{d}s \tag{227}$$

$$+ \sup_{0 \leq s \leq t} |\mathcal{M}_s|, \tag{228}$$

where $\mathcal{M}_t$ are all the martingale terms grouped together. Utilizing the same Lipschtiz map found in the proof of Lemma 7, there exists $L_\epsilon > 0$ such that the integrand may be bounded by

$$\left| \frac{2\eta\varphi(R_s)}{\sqrt{R_s}} \sum_{i=1}^{d} \lambda_i(\overline{\mathbf{K}})[\mathcal{J}(z)]_i r_i - \frac{2\eta\varphi(\mathcal{R}_s)}{\sqrt{\mathcal{R}_s}} \sum_{i=1}^{d} \lambda_i(\overline{\mathbf{K}})[\mathcal{J}(z)]_i \widetilde{r}_i \right| \tag{229}$$

$$\leq L_\epsilon \sqrt{|R_s - \mathcal{R}_s|^2 + \left| \sum_{i=1}^{d} \lambda_i(\overline{\mathbf{K}}) \left[\mathcal{J}(z)\right]_i (r_i - \widetilde{r}_i) \right|^2} \tag{230}$$

The latter term may be further bounded by

$$\left| \sum_{i=1}^{d} \lambda_i(\overline{\mathbf{K}}) \left[\mathcal{J}(z)\right]_i (r_i - \widetilde{r}_i) \right| \leq \left| \sum_{i=1}^{d} (1 + z[\mathcal{J}(z)]_i)(r_i - \widetilde{r}_i) \right| \tag{231}$$

$$\leq |R_s - \mathcal{R}_s| + 2 \left\| \overline{\mathbf{K}} \right\| |g(r(s)) - g(\widetilde{r}(s))|. \tag{232}$$

However, notice that

$$|R_s - \mathcal{R}_s| = \left| \frac{1}{2\pi i} \int_\Gamma \mathcal{J}(y)^{\mathrm{T}} (r(s) - \widetilde{r}(s)) \, \mathrm{d}y \right| \tag{233}$$

$$\leq \left\| \overline{\mathbf{K}} \right\| \sup_{g \in \mathcal{W}} |g(r(s)) - g(\widetilde{r}(s))|. \tag{234}$$

Therefore,

$$\left| \frac{2\eta\varphi(R_s)}{\sqrt{R_s}} \sum_{i=1}^{d} \lambda_i(\overline{\mathbf{K}})[\mathcal{J}(z)]_i r_i - \frac{2\eta\varphi(\mathcal{R}_s)}{\sqrt{\mathcal{R}_s}} \sum_{i=1}^{d} \lambda_i(\overline{\mathbf{K}})[\mathcal{J}(z)]_i \widetilde{r}_i \right| \tag{235}$$

$$\leq L_\epsilon C(\overline{\mathbf{K}}) \sup_{g \in \mathcal{W}} |g(r(s)) - g(\widetilde{r}(s))|. \tag{236}$$

Putting all this together we see that

$$\sup_{g \in \mathcal{W}} |g(r(t)) - g(\widetilde{r}(t))| \leq \sup_{0 \leq s \leq t} |\mathcal{M}_s| + \int_0^t L_\epsilon C(\overline{\mathbf{K}}) \sup_{g \in \mathcal{W}} |g(r(s)) - g(\widetilde{r}(s))| \, \mathrm{d}s. \tag{237}$$

By Gronwall's inequality,

$$\sup_{g \in \mathcal{W}} |g(r(t)) - g(\widetilde{r}(t))| \leq \sup_{0 \leq s \leq t} |\mathcal{M}_s| \exp \left( L_\epsilon C(\overline{\mathbf{K}}) t \right). \tag{238}$$

By (234) and Lemma 9, we see that $\mathcal{R}_t$ and $R_t$ concentrates. Finally, Theorem 1 concludes the proof.

### B.1.1. RISK CURVES FOR SGD

Following a similar approach but taking

$$q(\mathbf{x}) = \frac{1}{2} \mathbf{x}^{\mathrm{T}} \mathbf{R}(z; \mathbf{K}) \mathbf{x}, \tag{239}$$

we may derive a system of $d$-ODEs for SGD. We note this is not novel, and a full derivation in much greater generality is in (Collins-Woodfin et al., 2023); see also (Collins-Woodfin et al., 2024) for a shorter discussion. Using the HSGD formulation of vanilla streaming SGD (Collins-Woodfin et al., 2024), we arrive at the VANILLAODE for subgaussian noise and variance $\varsigma^2$,

$$\frac{\mathrm{d}v_i}{\mathrm{d}t} = -2\eta \lambda_i(\mathbf{K}) v_i + \frac{\eta(t)^2}{d} \lambda_i(\mathbf{K})(R_t^{SGD} + \varsigma^2/2), \quad \forall 1 \leq i \leq d. \tag{240}$$

$$R_t^{SGD} = \sum_{i=1}^{d} \lambda_i(\mathbf{K}) v_i. \tag{241}$$

## C. Convergence and phase-properties of the ODEs

**Lemma 16.** *If $\epsilon \sim N(0, v^2)$, then $\mathcal{R}(\Theta_t)$ is bounded from above and below for all $t > 0$.*

*Proof.* Take $q(\mathbf{x}) = \frac{1}{2}\mathbf{x}^T\mathbf{D}\mathbf{x}$ then plugging this into (100) we obtain

$$\mathrm{d}q(\mathbf{V}_t) = -\frac{4\eta}{\pi\sqrt{2\mathcal{R}(\mathbf{V}_t) + v^2}}\mathcal{R}(\mathbf{V}_t) + \frac{\eta^2}{\pi d}\mathrm{Tr}(\mathbf{K}_\sigma\mathbf{D})\,\mathrm{d}t + \mathcal{M}_t^\sigma. \tag{242}$$

By concentration inequalities we know that $\mathcal{M}_t^\sigma$ vanishes as $d \to \infty$, thus we will omit the martingale term. Solving for the stationary point yields the following roots,

$$\mathcal{R}_\pm = \frac{C_\eta^2 \pm C_\eta\sqrt{C_\eta^2 + 64v^2}}{64}, \tag{243}$$

where $C_\eta = \frac{\eta}{4d}\mathrm{Tr}(\mathbf{K}_\sigma\mathbf{D}) = \frac{\pi\eta}{8d}\mathrm{Tr}(\mathbf{D})$. Phase diagram analysis shows that if $\mathcal{R}(\mathbf{V}_t) < \mathcal{R}_+$, then $q(\mathbf{V}_t)$ is increasing. Conversely, if $\mathcal{R}(\mathbf{V}_t) > \mathcal{R}_+$ then $q(\mathbf{V}_t)$ is decreasing. Since $\mathbf{D}$ is positive-definite, $q(\mathbf{V}_t) > 0$ provided that $\mathbf{V}_t \neq 0$. The growth and decay conditions of $q(\mathbf{V}_t)$ implies that $q(\mathbf{V}_t)$ cannot converge to 0, nor diverge to $\infty$. Therefore, $q(\mathbf{V}_t)$ is bounded from above and below. Consequently, $\|\mathbf{V}_t\|$ is bounded from above and below and so $\mathcal{R}(\mathbf{V}_t)$ is as well. $\square$

**Theorem 7.** *If $\epsilon \sim N(0, v^2)$ and $\eta \in (0, \infty)$ is a fixed learning rate then there exists unique stationary points*

$$s_i = \frac{\eta\,\mathrm{Tr}(\mathbf{K}(\mathbf{u}_i \otimes \mathbf{w}_i)\mathbf{K}_\sigma)}{16\lambda_i(\overline{\mathbf{K}})d}\left(\frac{\eta\,\mathrm{Tr}(\mathbf{D}\mathbf{K}_\sigma)}{d} + \sqrt{\frac{\eta^2\,\mathrm{Tr}(\mathbf{D}\mathbf{K}_\sigma)^2}{d^2} + 16v^2}\right), \tag{244}$$

*and the limit risk is given by*

$$R_\infty = \frac{\eta}{16d}\mathrm{Tr}(\mathbf{D}\mathbf{K}_\sigma)\left(\frac{\eta\,\mathrm{Tr}(\mathbf{D}\mathbf{K}_\sigma)}{d} + \sqrt{\frac{\eta^2\,\mathrm{Tr}(\mathbf{D}\mathbf{K}_\sigma)^2}{d^2} + 16v^2}\right). \tag{245}$$

*We note that in these formulas, $\mathrm{Tr}(\mathbf{D}\mathbf{K}_\sigma) = \frac{\pi}{2}\mathrm{Tr}(\mathbf{D}) = \frac{\pi}{2}\mathrm{Tr}(\overline{\mathbf{K}})$ on account of $\mathbf{K}_\sigma$ having a constant diagonal.*

*Proof.* Let $Y_t = \frac{\pi\sqrt{2R_t + v^2}}{4\eta}$ and $m_i = \frac{\mathrm{Tr}(\mathbf{K}(\mathbf{u}_i \otimes \mathbf{w}_i)\mathbf{K}_\sigma)}{\pi d}$, then our $d$ coupled ODEs are given by

$$\frac{\mathrm{d}r_i}{\mathrm{d}t} = -\frac{\lambda_i(\overline{\mathbf{K}})r_i}{Y_t} + \eta^2 m_i, \quad r_i(0) = \frac{1}{2}\mathbf{V}_0^T(\mathbf{K}\mathbf{u}_i \otimes \mathbf{y}_i)\mathbf{V}_0. \tag{246}$$

Solving for the stationary point, we see that for all $1 \leq i \leq d$,

$$r_i = \frac{\eta^2 m_i Y_t}{\lambda_i(\overline{\mathbf{K}})}. \tag{247}$$

Thus, at equilibrium

$$R_t = \sum_{i=1}^d r_i = \eta^2 Y_t \sum_{i=1}^d \frac{m_i}{\lambda_i(\overline{\mathbf{K}})} = \frac{\eta^2 Y_t}{\pi d}\mathrm{Tr}(\mathbf{D}\mathbf{K}_\sigma). \tag{248}$$

However, $R_t$ can be expressed in terms of $Y$ by

$$R_t = \frac{1}{2}\left(\left(\frac{4\eta Y_t}{\pi}\right)^2 - v^2\right). \tag{249}$$

Plugging into (248) we see that

$$\frac{1}{2}\left(\left(\frac{4\eta Y_t}{\pi}\right)^2 - v^2\right) = \frac{\eta^2 Y_t}{\pi d}\mathrm{Tr}(\mathbf{D}\mathbf{K}_\sigma). \tag{250}$$

Solving for $Y_t$ yields the following positive root

$$Y_\infty = \frac{\pi}{16\eta} \left( \frac{\eta \operatorname{Tr}(\mathbf{DK}_\sigma)}{d} + \sqrt{\frac{\eta^2 \operatorname{Tr}(\mathbf{DK}_\sigma)^2}{d^2} + 16\mathfrak{v}^2} \right) \tag{251}$$

Therefore, by (247) and (249), $\frac{\mathrm{d}r_i}{\mathrm{d}t} = 0$ if and only if

$$r_i = s_i := \frac{\eta^2 m_i Y_\infty}{\lambda_i(\overline{\mathbf{K}})}, \quad \forall 1 \le i \le d. \tag{252}$$

This concludes uniqueness. The limiting risk is then given by

$$R_\infty = \frac{\eta^2 Y_\infty}{\pi d} \operatorname{Tr}(\mathbf{DK}_\sigma). \tag{253}$$

$\square$

Similarly, fixing $\eta$, we may derive unique stationary points to VANILLAODE (240):

**Theorem 8.** *If $\epsilon$ is subgaussian with variance $\mathfrak{v}^2$, $\eta \in (0, \infty)$ is a fixed learning rate and $\{v_i\}_{i=1}^d$ as given by (240), then there exists unique stationary points*

$$s_i^{SGD} = \frac{\eta \mathfrak{v}^2}{2(2d - \eta \operatorname{Tr}\mathbf{K})}, \tag{254}$$

*with limiting risk*

$$R_\infty^{SGD} = \frac{\eta \mathfrak{v}^2 \operatorname{Tr}(\mathbf{K})}{2(2d - \operatorname{Tr}(\mathbf{K})\eta)}. \tag{255}$$

**Theorem 9.** *Assume $\epsilon \sim N(0, \mathfrak{v}^2)$ and let $s_i$ be the stationary points to (13a). Then there is an absolute constant $c > 0$ so that if*

$$\eta \left( \frac{\operatorname{Tr}(\mathbf{DK}_\sigma)}{\pi d} \right) \le \min \left\{ c, \frac{4\mathfrak{v}}{\pi} \right\}, \quad \text{and} \quad R_0 \le c\mathfrak{v},$$

*then we have, setting $R_\infty = \sum_{i=1}^d s_i$ to be the limit risk,*

$$|R_t - R_\infty| \le 2(R_0 + R_\infty)e^{-t\eta\lambda_{\min}(\overline{\mathbf{K}})/(\pi\mathfrak{v})}.$$

*We note again that in these formulas, $\operatorname{Tr}(\mathbf{DK}_\sigma) = \frac{\pi}{2}\operatorname{Tr}(\mathbf{D}) = \frac{\pi}{2}\operatorname{Tr}(\overline{\mathbf{K}})$ on account of $\mathbf{K}_\sigma$ having a constant diagonal.*

*Proof.* We recall (252), in terms of which we have

$$\frac{\mathrm{d}r_i}{\mathrm{d}t} = -\frac{\lambda_i(\overline{\mathbf{K}})}{Y_t}r_i + \frac{\lambda_i(\overline{\mathbf{K}})}{Y_\infty}s_i,$$

and where we recall

$$Y_t = \frac{\pi\sqrt{2R_t + \mathfrak{v}^2}}{4\eta}.$$

Then we rewrite the evolution of $r_i$ as

$$\frac{\mathrm{d}}{\mathrm{d}t}(r_i - s_i) = -\frac{\lambda_i(\overline{\mathbf{K}})}{Y_\infty}(r_i - s_i) + \left( \frac{\lambda_i(\overline{\mathbf{K}})}{Y_\infty} - \frac{\lambda_i(\overline{\mathbf{K}})}{Y_t} \right) r_i,$$

and we set $R_\infty$ as $\sum s_i$. Now we observe that

$$\frac{Y_t^2 - Y_\infty^2}{Y_\infty^2} = \frac{\pi^2}{8\eta^2 Y_\infty^2}(R_t - R_\infty) =: \alpha(R_t - R_\infty), \tag{256}$$

from which it follows

$$\frac{1}{Y_\infty} - \frac{1}{Y_t} = \frac{Y_t^2 - Y_\infty^2}{Y_t Y_\infty (Y_t + Y_\infty)} = \frac{Y_t^2 - Y_\infty^2}{2Y_\infty^3} + \mathrm{Err}_t,$$

where $\mathrm{Err}_t$ is bounded by

$$\mathrm{Err}_t \leq C \frac{1}{\mathcal{Y}} \left( \frac{Y_t^2 - Y_\infty^2}{Y_\infty^2} \right)^2 \leq C \frac{\alpha^2}{\mathcal{Y}} (R_t - R_\infty)^2, \tag{257}$$

where $\mathcal{Y}$ is the minimum value of $Y_t$ over all time and $C$ is an absolute constant. Hence we can further develop

$$\frac{\mathrm{d}}{\mathrm{d}t}(r_i - s_i) = -\left( \frac{1}{Y_\infty} - \frac{Y_t^2 - Y_\infty^2}{2Y_\infty^3} - \mathrm{Err}_t \right) \lambda_i(\overline{\mathbf{K}})(r_i - s_i) + \left( \frac{Y_t^2 - Y_\infty^2}{2Y_\infty^3} + \mathrm{Err}_t \right) \lambda_i(\overline{\mathbf{K}}) s_i.$$

Define

$$\varrho(s) = \int_0^s \left( \frac{1}{Y_\infty} - \frac{Y_t^2 - Y_\infty^2}{2Y_\infty^3} - \mathrm{Err}_t \right) \mathrm{d}t.$$

Then by variation of parameters, we have

$$(r_i - s_i)(t) = (r_i - s_i)(0) e^{-\lambda_i \varrho(t)} + \int_0^t e^{-\lambda_i(\varrho(t) - \varrho(s))} \left( \frac{Y_s^2 - Y_\infty^2}{2Y_\infty^3} + \mathrm{Err}_s \right) \lambda_i(\overline{\mathbf{K}}) s_i \mathrm{d}s.$$

Now if we sum over all $i$, we have

$$R_t - R_\infty = \mathcal{F}(t) + \int_0^t \mathcal{K}(t, s) \left( \frac{Y_s^2 - Y_\infty^2}{2Y_\infty^3} + \mathrm{Err}_s \right) \mathrm{d}s,$$

where

$$\mathcal{F}(t) = \sum_i (r_i - s_i)(0) e^{-\lambda_i \varrho(t)} \quad \text{where} \quad \mathcal{K}(t, s) = \sum_i e^{-\lambda_i(\varrho(t) - \varrho(s))} \lambda_i(\overline{\mathbf{K}}) s_i.$$

Now suppose that on some interval of time $[0, T]$

$$\mathrm{Err}_t \leq \frac{Y_t^2 - Y_\infty^2}{2Y_\infty^3} \quad \text{and} \quad 2\frac{Y_t^2 - Y_\infty^2}{2Y_\infty^3} \leq \frac{1}{2Y_\infty}. \tag{258}$$

Then for $s < t < T$, we have

$$\varrho(t) - \varrho(s) \geq \frac{1}{2Y_\infty}(t - s),$$

and so we have the convolution Volterra upper bound for $t \leq T$

$$|R_t - R_\infty| \leq |\mathcal{F}(t)| + \frac{\alpha}{Y_\infty} \int_0^t \left( \sum_i e^{-\lambda_i(t-s)/(2Y_\infty)} \lambda_i(\overline{\mathbf{K}}) s_i \right) |R_s - R_\infty| \mathrm{d}s.$$

Now we note that we have the upper bound (for $t \leq T$)

$$|\mathcal{F}(t)| \leq (R_0 + R_\infty) e^{-(\lambda_{\min}/(2Y_\infty))t}.$$

Now suppose that $0 < T' \leq T$ is such that for $s \leq T'$

$$|R_s - R_\infty| \leq M e^{-(\lambda_{\min}/(4Y_\infty))t},$$

we have for $t \leq T'$,

$$\int_0^t \left( \sum_i e^{-\lambda_i(t-s)/(2Y_\infty)} \lambda_i(\overline{\mathbf{K}}) s_i \right) M e^{-(\lambda_{\min}/(4Y_\infty))s} \, \mathrm{d}s$$

$$= \left( \sum_i e^{-\lambda_i t/(2Y_\infty)} \left( e^{\lambda_i t/(2Y_\infty) - \lambda_{\min} t/(4Y_\infty)} - 1 \right) \frac{\lambda_i(\overline{\mathbf{K}}) s_i}{\lambda_i/(2Y_\infty) - \lambda_{\min}/(4Y_\infty)} \right) M$$

$$\leq e^{-\lambda_{\min} t/(4Y_\infty)} \left( \sum_i \frac{\lambda_i(\overline{\mathbf{K}}) s_i}{\lambda_i/(2Y_\infty) - \lambda_{\min}/(4Y_\infty)} \right) M$$

$$\leq e^{-\lambda_{\min} t/(4Y_\infty)} \left( \sum_i \frac{\lambda_i(\overline{\mathbf{K}}) s_i}{\lambda_i/(2Y_\infty) - \lambda_i/(4Y_\infty)} \right) M$$

$$\leq e^{-\lambda_{\min} t/(4Y_\infty)} (4Y_\infty R_\infty) M.$$

Hence $T' = T$, provided

$$4\alpha R_\infty < 1 \quad \text{and} \quad M = \frac{(R_0 + R_\infty)}{1 - 4\alpha R_\infty}.$$

Now we return to showing $T$ does not occur. Recall (258), which up to $T$ are satisfied. Then it suffices to have (compare (257)),

$$\alpha M \leq \frac{1}{2} \quad \text{and} \quad 2C\frac{Y_\infty}{\mathcal{Y}}\alpha M \leq 1, \quad \text{where} \quad \alpha = \frac{\pi^2}{8\eta^2 Y_\infty^2}.$$

in which case (258) is satisfied for all time. Note that we may always bound $\mathcal{Y}$ below by

$$\mathcal{Y} \geq (\pi \mathfrak{v})/(4\eta).$$

Define $H_{\mathbf{K}} = \frac{\mathrm{Tr}(\mathbf{D}\mathbf{K}_\sigma)}{\pi d}$. We now recall (251) and (249), from which

$$Y_\infty = \pi^2 \left( \frac{\eta H_{\mathbf{K}} + \sqrt{\eta^2 H_{\mathbf{K}}^2 + 16\mathfrak{v}^2/\pi^2}}{16\eta} \right) \quad \text{and} \quad R_\infty = Y_\infty \eta^2 H_{\mathbf{K}}.$$

Hence for $\eta H_{\mathbf{K}} \leq 4\mathfrak{v}/\pi$, we have

$$\frac{\pi \mathfrak{v}}{4\eta} \leq Y_\infty \leq \frac{\pi \mathfrak{v}}{\eta} \leq 4\mathcal{Y},$$

and hence we have

$$\alpha \leq \frac{2}{\mathfrak{v}} \quad \text{and} \quad R_\infty \leq \eta \pi \mathfrak{v} H_{\mathbf{K}}.$$

Thus we conclude there is an absolute constant $c > 0$ so that if

$$\eta H_{\mathbf{K}} \leq \min\left\{ c, \frac{4\mathfrak{v}}{\pi} \right\}, \quad \text{and} \quad R_0 \leq c\mathfrak{v},$$

then we have

$$|R_t - R_\infty| \leq 2(R_0 + R_\infty)e^{-t\eta\lambda_{\min}(\mathbf{K})/(\pi\mathfrak{v})}.$$

$\square$

## D. Minibatching SDEs comparison

In this section, we will present the SDE comparison between SIGNHSGD and HSGD when $b > 1$ for convenience of future work. Modifying (15), it is easy to check that HSGD with minibatching is

$$d\boldsymbol{\Theta}_t^{\text{SGD}} = -\eta_t \mathbf{K}(\boldsymbol{\Theta}_t^{\text{SGD}} - \boldsymbol{\theta}_*)dt + \eta_t \sqrt{\frac{2\mathcal{P}(\boldsymbol{\Theta}_t^{\text{SGD}})}{db}} \mathbf{K}d\mathbf{B}_t. \tag{259}$$

Once again running SGD with adaptive learning rate

$$\eta_t^{\text{SGD}} = \frac{2\eta_t}{\pi} \sqrt{\frac{b}{2\mathcal{P}(\boldsymbol{\Theta}_t^{\text{SGD}})}}, \tag{260}$$

we obtain the similar SDEs for HSGD and SIGNHSGD as in (17a) and (17b)

$$d\boldsymbol{\Theta}_t^{\text{SGD}} = -\eta_t^{\text{SGD}} \mathbf{K}(\boldsymbol{\Theta}_t^{\text{SGD}} - \boldsymbol{\theta}_*)dt + \eta_t \sqrt{\frac{4\mathbf{K}}{\pi^2 d}} d\mathbf{B}_t, \tag{261}$$

$$d\boldsymbol{\Theta}_t = -\eta_t^{\text{SGD}} \underbrace{\psi^{(b)}(\mathcal{R}(\boldsymbol{\Theta}_t))}_{\epsilon\text{-compress.}} \underbrace{\mathbf{D}^{-1}}_{\text{D.Precond.}} \mathbf{K}(\boldsymbol{\Theta}_t - \boldsymbol{\theta}_*)dt + \eta_t \underbrace{\sqrt{\frac{2\mathbf{K}_\sigma}{\pi d}}}_{\text{Reshape}} d\mathbf{B}_t, \tag{262}$$

where

$$\psi^{(b)}(x) = \frac{\pi}{2} \sqrt{\frac{2x + v^2}{2x}} \frac{\varphi^{(b)}(x)N_b}{\sqrt{b}}. \tag{263}$$

This shows that our analysis in Section 4 also applies to the mini-batch setting. Notably, the effective learning rate scales by a factor $\sqrt{b}$, while the $\epsilon$-compression term $\psi^{(b)}$ gains some additional regularity near 0 when $b > 1$. See Section 5.

# E. Additional experiments

We begin by illustrating (Figure 6) the concentration effect: as $d$ increases, the loss curves more closely match the ODEs. We also note the spread of SIGNHSGD and SIGNSGD are close across dimension.

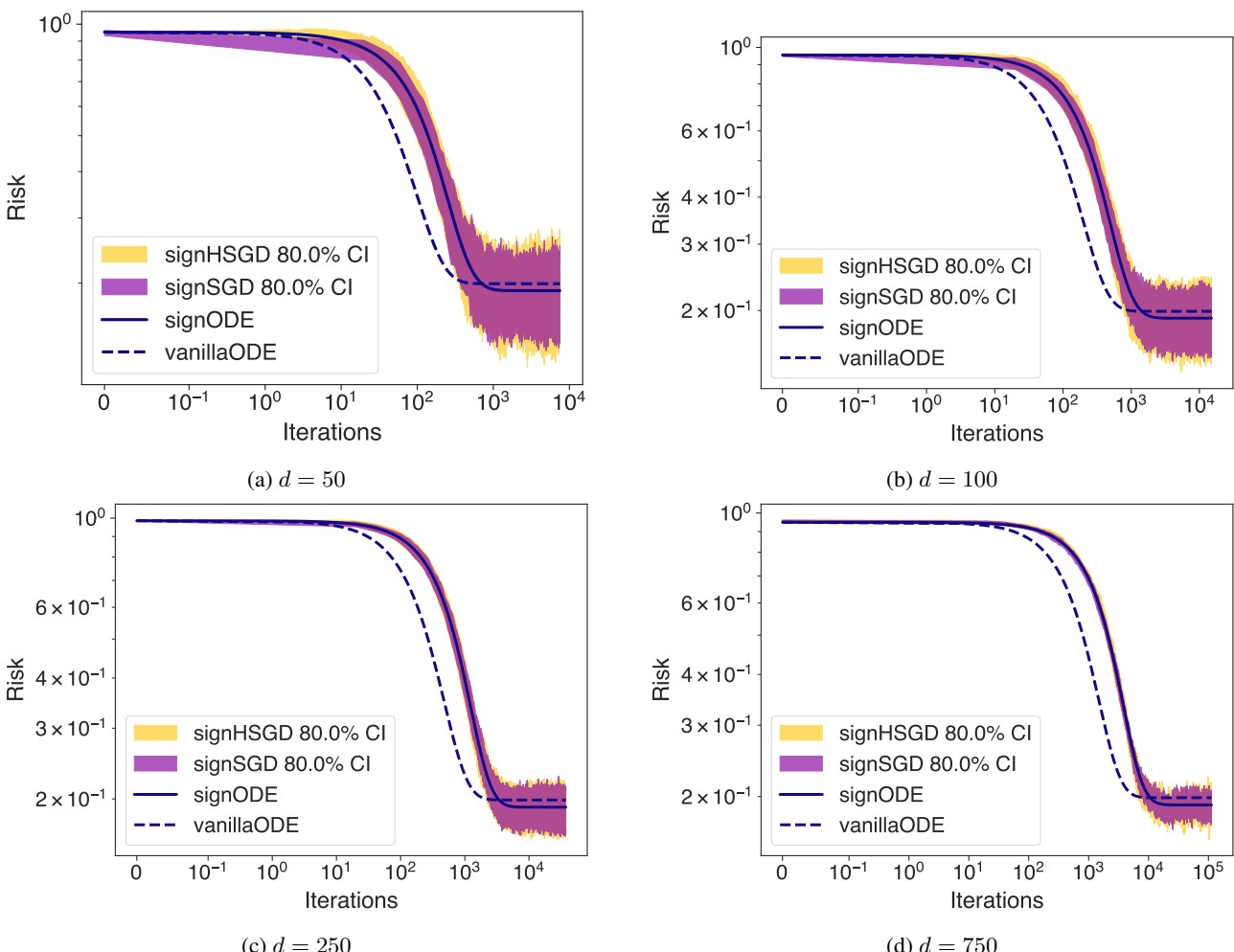

(a) $d = 50$

(b) $d = 100$

(c) $d = 250$

(d) $d = 750$

Figure 6: A demonstration that SIGNSGD, SIGNHSGD, and their deterministic equivalent concentrate in high-dimensions over long time scales. In the limit as $d \to \infty$ our main theorem shows that all these objects become the same.

In the next figure we compare the limit risk prediction in Theorem 3.

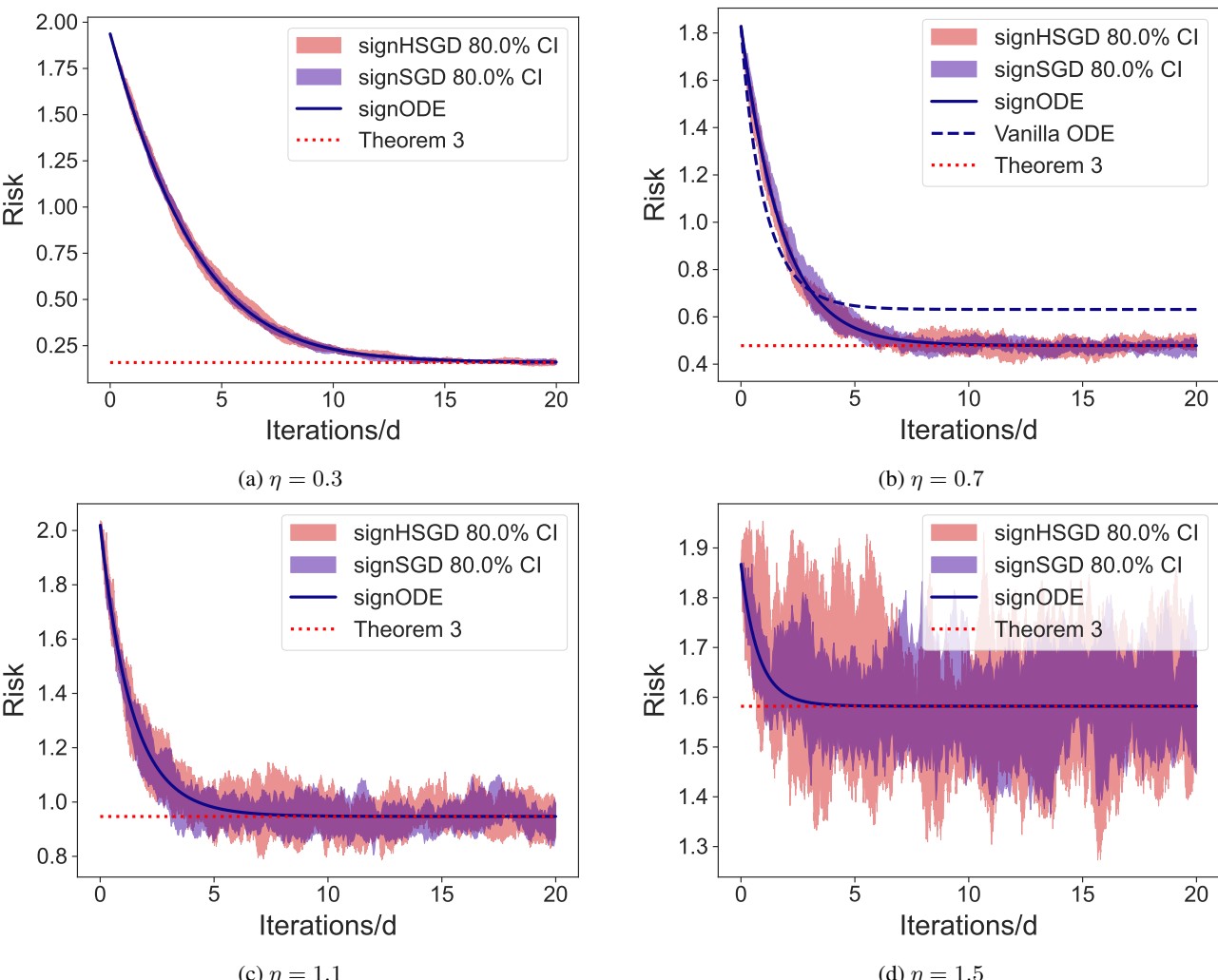

(a) $\eta = 0.3$

(b) $\eta = 0.7$

(c) $\eta = 1.1$

(d) $\eta = 1.5$

Figure 7: A demonstration of Theorem 3, over varying learning rates. Here $d = 500$, and we take Gaussian noise with $v = 0.7$

To demonstrate the convergence rates, we consider a set of diagonal covariance matrices. The eigenvalues on the diagonal are given by a uniform grid of $[0.5, 1.0]$. To these eigenvalues, we then raise them to a power $\alpha$ over the range $(1.0, \ldots, 5.0)$. This causes the smallest eigenvalue to approach 0. We then compare SIGNSGD vs SGD after $n = Td$ steps with $T = 30$. The noise is set to $v = 0.01$.

The learning rates are taken 'optimal' which is to say that they are $\eta d / \operatorname{tr}(\mathbf{K})$ and $\eta d / \operatorname{tr}(\overline{\mathbf{K}})$ for SGD and SIGNSGD respectively for a fixed multiple $\eta$. The constant is given by $\eta = 0.01$. The resulting risk curves look like:

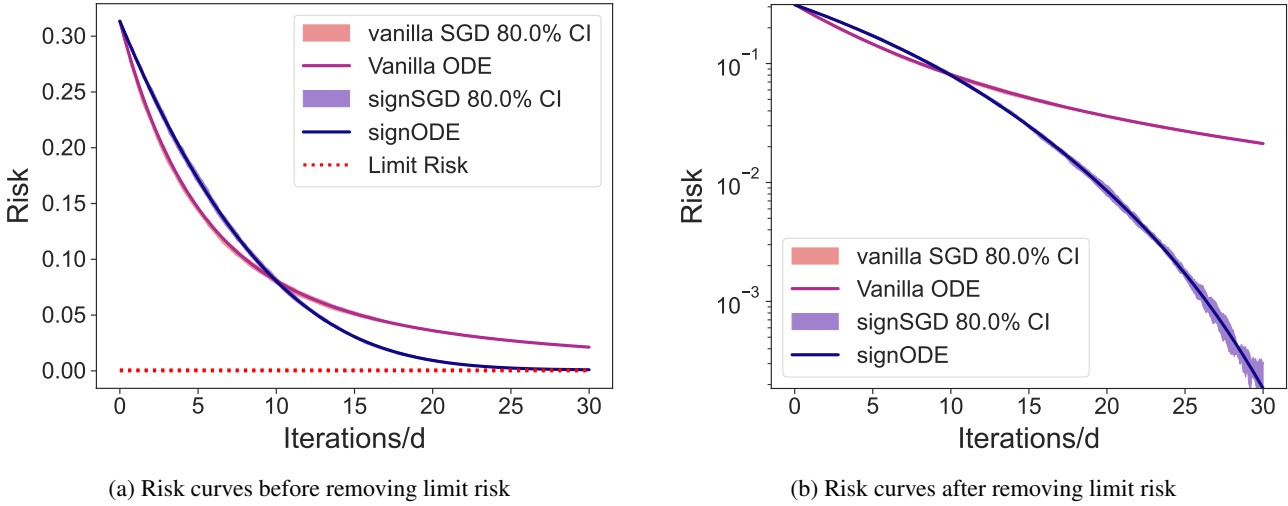

(a) Risk curves before removing limit risk

(b) Risk curves after removing limit risk

Figure 8: These are the risk curves from the setup described in the text (with $\alpha = 5.0$). After recentering using the values from Theorem 3 and (255), (b) shows the linear convergence.

Now varying over these problems over $\alpha$ between $1.0$ and $5.0$. We have:

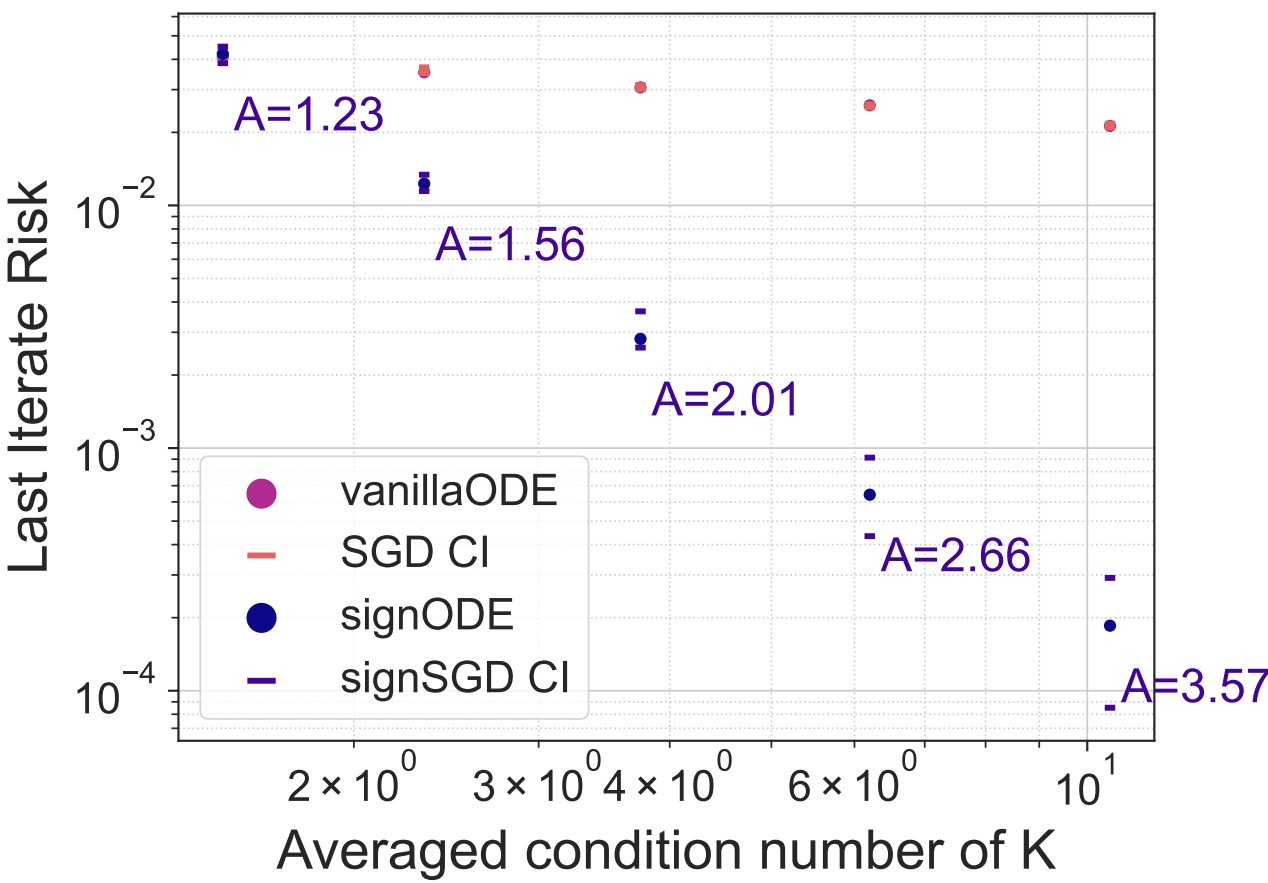

Figure 9: We plot the limiting suboptimality against the averaged condition number of the problem $\mathrm{Tr}(\mathbf{K})/(d\lambda_{\min}(\mathbf{K}))$. SGD attains this convergence rate. The label $A$ for each point is the ratio of the averaged condition number of $\mathbf{K}$ to the averaged condition number of $\overline{\mathbf{K}}$. This measures the speedup of SIGNSGD over SGD, and is the rate effect captured by Theorem 4.

## F. Heuristic for ADAM

In this section we derive a heuristic for ADAM (Kingma, 2014). This is given, in our context, by:

> Given:
>> $\eta$ : learning rate
>>
>> $\beta_1, \beta_2 \in [0, 1)$ : exponential decay rates for moment estimates
>>
>> $\epsilon_0$ : small constant for numerical stability
>
> Initialize:
>> $\boldsymbol{\theta}_0$ : initial parameter vector
>>
>> $\mathbf{m}_0 \leftarrow \mathbf{0}$ : 1st moment vector
>>
>> $\mathbf{v}_0 \leftarrow \mathbf{0}$ : 2nd moment vector
>>
>> $k \leftarrow 0$ : timestep
>
> Repeat until convergence:
>> $k \leftarrow k + 1$
>>
>> $\mathbf{g}_k \leftarrow \nabla_{\boldsymbol{\theta}} \mathcal{L}(\boldsymbol{\theta}_{k-1}, \mathbf{x}_k, y_k)$   (Get gradients w.r.t. stochastic objective at timestep k)
>>
>> $\mathbf{m}_k \leftarrow \beta_1 \cdot \mathbf{m}_{k-1} + (1 - \beta_1) \cdot \mathbf{g}_k$   (Update biased first moment estimate)
>>
>> $\mathbf{v}_k \leftarrow \beta_2 \cdot \mathbf{v}_{k-1} + (1 - \beta_2) \cdot \mathbf{g}_k^2$   (Update biased second raw moment estimate)
>>
>> $\hat{\mathbf{m}}_k \leftarrow \mathbf{m}_k / (1 - \beta_1^k)$   (Compute bias-corrected first moment estimate)
>>
>> $\hat{\mathbf{v}}_k \leftarrow \mathbf{v}_k / (1 - \beta_2^k)$   (Compute bias-corrected second raw moment estimate)
>>
>> $\boldsymbol{\theta}_k \leftarrow \boldsymbol{\theta}_{k-1} - \eta \hat{\mathbf{m}}_k / (\sqrt{\hat{\mathbf{v}}_k} + \epsilon_0)$   (Update parameters)

In a high-dimensional context, the first moment momentum $\beta_1$ has been observed to be equivalent to an effective change of learning rate, without inducing other benefits on the dynamics (see (Paquette & Paquette, 2021)), and so we ignore it.

The role of the second moment, in contrast, should induce a preconditioner. If we assume that exponential decay rate of $\beta_2$ is chosen sufficiently close to 1, we would have

$$\hat{\mathbf{v}}_k \approx_{\beta_2} \mathbb{E}(\nabla_{\boldsymbol{\theta}} \mathcal{L}(\boldsymbol{\theta}_{k-1}, \mathbf{x}_k, y_k))^2 | \mathscr{F}_{k-1}),$$

with the square applied entrywise. Using the definition of the stochastic gradient, we have

$$\hat{\mathbf{v}}_k = \mathbb{E}\left( (\mathbf{x}_k)^2 \left( \langle \mathbf{x}_k, \boldsymbol{\theta}_{k-1} - \boldsymbol{\theta}_* \rangle + \epsilon_k \right)^2 | \mathscr{F}_{k-1} \right).$$

This can be computed explicitly by Gaussian conditioning. Note that conditionally on the Gaussian $\mathbf{w} = \langle \mathbf{x_k}, \boldsymbol{\theta_{k-1}} - \boldsymbol{\theta_*} \rangle$, $\mathbf{x}_k$ develops a mean $\mathbf{K}(\boldsymbol{\theta}_{k-1} - \boldsymbol{\theta}_*)$, which has norm $O(1)$. Hence provided it also has small $\mathrm{L}^\infty$ norm, so too will all the variances of the entries of $\mathbf{x}_k$ be nearly unaffected. Hence we essentially have independence, in that

$$\hat{\mathbf{v}}_k \approx \mathbb{E}\left( (\mathbf{x}_k)^2 \right) \mathbb{E}\left( \left( \langle \mathbf{x}_k, \boldsymbol{\theta}_{k-1} - \boldsymbol{\theta}_* \rangle + \epsilon_k \right)^2 | \mathscr{F}_{k-1} \right) = \mathrm{diag}(\mathbf{K})(2\mathcal{P}).$$

Hence, we arrive at the approximate update rule for ADAM

$$\boldsymbol{\theta}_{k+1} = \boldsymbol{\theta}_k - \frac{\eta_k}{\sqrt{2\mathcal{P}(\boldsymbol{\theta}_k)}} \mathbf{D}^{-1} \nabla_{\boldsymbol{\theta}} \mathcal{L}(\boldsymbol{\theta}_k, \mathbf{x}_{k+1}, y_{k+1}). \tag{264}$$

The corresponding homogenized ADAM equation is given by

$$d\boldsymbol{\Theta}_t = -\frac{\eta_t}{\sqrt{2\mathcal{P}(\boldsymbol{\Theta}_t)}} \overline{\mathbf{K}}(\boldsymbol{\Theta}_t - \boldsymbol{\theta}_*) + \eta_t \mathbf{D}^{-1/2} \sqrt{\overline{\mathbf{K}}} d\mathbf{B}_t. \tag{265}$$

## G. Weak-approximation and high-dimension

The idea of approximating discrete stochastic processes using differential equations is not new and can be traced back to (Robbins & Monro, 1951). These approaches typically involved deriving an ODE by sending the step size of a stochastic

difference equation to zero and demonstrating that the noise vanishes in the asymptotic limit. In recent years, SDEs have also been integrated into the study of SGD dynamics. In particular, the work of (Li et al., 2019) establishes a weak-approximation framework for tracking SGD across a wide range of suitable test functions. (Malladi et al., 2022) further builds on these ideas to derive SDEs for ADAM and RMSprop. Simultaneously with our work, (Compagnoni et al., 2025) applies the weak-approximation framework to derive SDEs for SIGNSGD. Their approach leverages the noisy gradient model, which allows them to extend their theory to a broader class of loss functions. In contrast, our approach follows the high-dimensional framework developed by (Paquette et al., 2022a) and (Collins-Woodfin & Paquette, 2023).

In the high-dimensional setting, we are interested in studying a sequence of problems of growing complexity (as dimension goes to infinity). This influences our notion of convergence. As shown in Theorem 1, we demonstrate that the risk curves of SIGNSGD and its SDE, SIGNHSGD, closely track each other in a pointwise sense. This enables us to directly study the behavior of SIGNSGD by analyzing SIGNHSGD, as seen in Theorems 3 and 4. Additionally, the precision of this tracking improves as the dimension increases. In contrast, within the weak-approximation framework, for a fixed dimension, the notion of convergence is more akin to convergence in distribution between the SDE and the optimizer as the learning rate approaches zero. Below, we highlight some of the key differences between these two approaches.

In the following, we will show that SIGNHSGD as defined in (8) is *almost* recoverable from the weak-approximation framework. The weak-approximation SDE of (Compagnoni et al., 2025) for SIGNSGD with learning rate $\eta$ and constant scheduler $\kappa$ is described by

$$d\boldsymbol{\Theta}_t^{WA} = -\kappa(1 - 2\mathbb{P}\left(\nabla f_\gamma(\boldsymbol{\Theta}_t^{WA}) < 0\right))dt + \kappa\sqrt{\eta\overline{\boldsymbol{\Sigma}}(\boldsymbol{\Theta}_t^{WA})}d\mathbf{B}_t, \tag{266}$$

where $\nabla f_\gamma(\boldsymbol{\Theta}_t^{WA}) = \mathbb{E}[\nabla f_\gamma(\boldsymbol{\Theta}_t^{WA})] + Z$, for $Z$ the gradient noise, and $\overline{\boldsymbol{\Sigma}}$ the covariance matrix of $\sigma(\nabla f_\gamma(\boldsymbol{\Theta}_t^{WA})) - \mathbb{E}[\sigma(\nabla f_\gamma(\boldsymbol{\Theta}_t^{WA}))]$. Moreover, the probability and expectation are understood to be taken conditional on $\boldsymbol{\Theta}_t^{WA}$. See (Malladi et al., 2022) for more on the noisy gradient model. We consider the quadratic-loss function and to match our label-noise gradient to the noisy gradient, we set $Z(\boldsymbol{\Theta}_t^{WA}) = \mathbf{x}_t(\langle \mathbf{x}_t, \boldsymbol{\Theta}_t^{WA} - \boldsymbol{\theta}_* \rangle - \epsilon_t) - \mathbf{K}(\boldsymbol{\Theta}_t^{WA} - \boldsymbol{\theta}_*)$, for $\mathbf{x}_t \sim N(0, \mathbf{K})$ and $\epsilon_t$ independent to $\boldsymbol{\Theta}_t^{WA}$. Since the effective learning rate in high-dimension is $\eta/d$, choosing $\kappa = 1/d$ and plugging this into (266), we obtain

$$d\boldsymbol{\Theta}_t^{WA} = -\frac{1}{d}(1 - 2\mathbb{P}(\mathbf{x}_t\left(\langle \mathbf{x}_t, \boldsymbol{\Theta}_t^{WA} - \boldsymbol{\theta}_* \rangle - \epsilon_t\right) < 0))dt + \frac{1}{d}\sqrt{\eta\left(\frac{2\mathbf{K}_\sigma}{\pi} - \mathbf{M}(\boldsymbol{\Theta}_t^{WA})\right)}d\mathbf{B}_t \tag{267}$$

$$= -\frac{1}{d}\mathbb{E}\left[\sigma(\mathbf{x}_t(\langle \mathbf{x}_t, \boldsymbol{\Theta}_t^{WA} - \boldsymbol{\theta}_* \rangle - \epsilon_t))\right]dt + \frac{1}{d}\sqrt{\eta\left(\frac{2\mathbf{K}_\sigma}{\pi} - \mathbf{M}(\boldsymbol{\Theta}_t^{WA})\right)}d\mathbf{B}_t, \tag{268}$$

where $\mathbf{M}(\boldsymbol{\Theta}_t) = \mathbb{E}\left[\sigma(\mathbf{x}_t(\langle \mathbf{x}_t, \boldsymbol{\Theta}_t^{WA} - \boldsymbol{\theta}_* \rangle - \epsilon_t))\right]^{\otimes 2}$. Computing the expectation of the signed gradient is not a straightforward task and to reduce it to a simplified form requires high-dimensional concentration. See Lemmas 1, 3 and 4. Thus, taking the liberty of fixing the dimension $d$ to be large, for $\rho < 1/4$ as specified in Lemma 4,

$$d\boldsymbol{\Theta}_t^{WA} = -\frac{1}{d}\left(\frac{\varphi(\mathcal{R}(\boldsymbol{\Theta}_t^{WA}))}{\sqrt{2\mathcal{R}(\boldsymbol{\Theta}_t^{WA})}}\overline{\mathbf{K}}(\boldsymbol{\Theta}_t^{WA} - \boldsymbol{\theta}_*) + O\left(d^{3\rho-2}\right)\right)dt + \frac{1}{d}\sqrt{\eta\left(\frac{2\mathbf{K}_\sigma}{\pi} - \mathbf{M}(\boldsymbol{\Theta}_t^{WA})\right)}d\mathbf{B}_t. \tag{269}$$

We present SIGNHSGD for ease of reference,

$$d\boldsymbol{\Theta}_t = -\eta\frac{\varphi(\mathcal{R}(\boldsymbol{\Theta}_t))}{\sqrt{2\mathcal{R}(\boldsymbol{\Theta}_t)}}\overline{\mathbf{K}}(\boldsymbol{\Theta}_t - \boldsymbol{\theta}_*)dt + \eta\sqrt{\frac{2\mathbf{K}_\sigma}{\pi d}}d\mathbf{B}_t. \tag{270}$$

Most notable between SIGNHSGD and (269) is the error $O\left(d^{3\rho-2}\right)$ and $\mathbf{M}(\boldsymbol{\Theta}_t^{WA})$ in the drift and diffusion term of (269) respectively. It is clear that the error in the drift term diminishes as the dimension increases. However, the error in the diffusion term is less obvious. We once again turn towards high-dimensional concentration. Indeed, by Lemmas 3, 4 and 12, we see that for all $1 \leq i, j \leq d$, $[\mathbf{M}(\boldsymbol{\Theta}_t^{WA})]_{ij} = O\left(d^{2\rho-1}\right)$ with high-probability.

Another important distinction between the two SDEs is the presence of $\eta$ in both the drift and diffusion term of SIGNHSGD. Whereas with $\boldsymbol{\Theta}_t^{WA}$, $\eta$ is only present in the diffusion term. This relates to the different notions of convergence within the respective frameworks. In the weak-approximation setting, $\boldsymbol{\Theta}_t^{WA}$ is an order 1 approximate of $\boldsymbol{\theta}_k$ in the sense of

$$\sup_{0 \leq k \leq \lfloor T/\eta \rfloor} \left|\mathbb{E}[g(\boldsymbol{\theta}_k)] - \mathbb{E}[g(\boldsymbol{\Theta}_{k\eta}^{WA})]\right| \leq C(d, T)\eta, \tag{271}$$

where $g$ is any function with polynomial growth. We see that within a fixed dimension setting, $\boldsymbol{\Theta}_t^{WA}$ has improved performance in approximating $\boldsymbol{\theta}_k$ when $\eta \to 0$ and the drift term dominates. On the contrary, SIGNHSGD vanishes if we were to send $\eta \to 0$. Instead, we take the limit in dimension and show that SIGNHSGD improves as dimension $d \to \infty$. See Theorem 1. Comparing these two regimes, it is not immediately clear whether one can send both $\eta \to 0$ and $d \to \infty$ in (271) and still retain good approximation. For instance, if we fix $\eta$ and send $d \to \infty$ in (269), we see that the diffusion term persists as $\mathbf{K}_\sigma$ grows with dimension. On the other hand, the drift term vanishes. The consideration of dimensionality as the problem scales is an important aspect of our work.

# H. $\overline{\mathbf{K}}$ does not always reduce the condition number

As a counter example consider the covariance matrix,

$$\mathbf{K} = \begin{bmatrix} 0.17 & -0.49 & -0.19 & -0.36 \\ -0.49 & 2.34 & 0.71 & 1.79 \\ -0.19 & 0.71 & 0.32 & 0.53 \\ -0.36 & 1.79 & 0.53 & 1.44 \end{bmatrix}. \tag{272}$$

Up to two decimals the condition number 2is $\kappa(\mathbf{K}) = 115.88$. However, the condition number of $\overline{\mathbf{K}}$ is $\kappa(\overline{\mathbf{K}}) = 129.78$.

# I. Experimental details

The code to reproduce these results is available at `https://anonymous.4open.science/r/signSGD-6216/`. We summarize the experimental setup of Figure 1 in Table 1.

| Dataset | Learning Rate ($\eta$) | Dimension | Noise Distribution | Noise Details | # Iterations |
|---|---|---|---|---|---|
| Synthetic (Gaussian noise) | 0.7 | 500 | Gaussian | $\mathbb{E}[\epsilon] = 0, \mathbb{E}[\epsilon^2] = 1$ | 5,000 |
| Synthetic (Cauchy noise) | 0.5 | 500 | Cauchy | Location = 2, Scale = 1 | 5,000 |
| CIFAR10 | 0.9 | 400 | Gaussian (assumed) | $\mathbb{E}[\epsilon] = 1, \mathbb{E}[\epsilon^2] = 0.76$ | 40,000 |
| IMDB | 0.2 | 50 | 2-GMM (assumed) | $\epsilon = \pi_1 g_1 + \pi_2 g_2$ 
 $\pi_1 = \pi_2 = 0.5$ 
 $\mathbb{E}[g_1] = -0.76, \mathbb{E}[g_1^2] = 0.18$ 
 $\mathbb{E}[g_2] = 0.75, \mathbb{E}[g_2^2] = 0.17$ | 25,000 |

Table 1: A summary of experimental details of Figure 1. The full details of the experiments are available below.

The experiments creating Figure 1 were carried out on an M1 Macbook Air. Homogenized SIGNHSGD is solved via a standard Euler-Maruyama algorithm. The procedure for solving for the risk is described in Appendix B.

**Synthetic data:** The synthetic data was generated in dimension $d = 500$. The covariance matrix $\mathbf{K}$ was generated by multiplying a random unitary matrix by a diagonal matrix of $d$ log-spaced eigenvalues between 0.01 and 0.5.

$\varphi$ was explicitly computed in the Gaussian data case and was solved via numerical integration in the case of Cauchy (Student's-t family) noise. Note that vanilla SGD does not converge under Cauchy noise and thus we cannot provide a comparison. We plot the 80% confidence interval across 20 runs.

**CIFAR10:** The CIFAR10 (Krizhevsky, 2009) data was used to perform binary classification by regressing to $\pm 1$ labels being animals or vehicles. The "frog" class was removed to retain balanced classes. The data matrix $D$ is first passed through a random features model so that

$$D_{rf} = \tanh DA \tag{273}$$

where $A$ is a random features matrix of independent standard Gaussians. This choice was found to better condition the data so that SIGNODE could be effectively solved via numerical integration.

In order to estimate $\boldsymbol{\theta}_*$ the regression problem was first solved using Sci-kit learn (Pedregosa et al., 2011) and the resulting solution was taken to be $\boldsymbol{\theta}_*$. The differences $\{y_i - \langle \boldsymbol{\theta}_*, \mathbf{x}_i \rangle\}$ for all $\mathbf{x}_i \in D_{rf}$ was then assumed to be the noise. A histogram of this noise is available in Figure 10a. The noise was then fitted to a Gaussian. Finally, $\eta = 0.5$ and the SIGNSGD plot represents the 80% confidence interval over 50 runs.

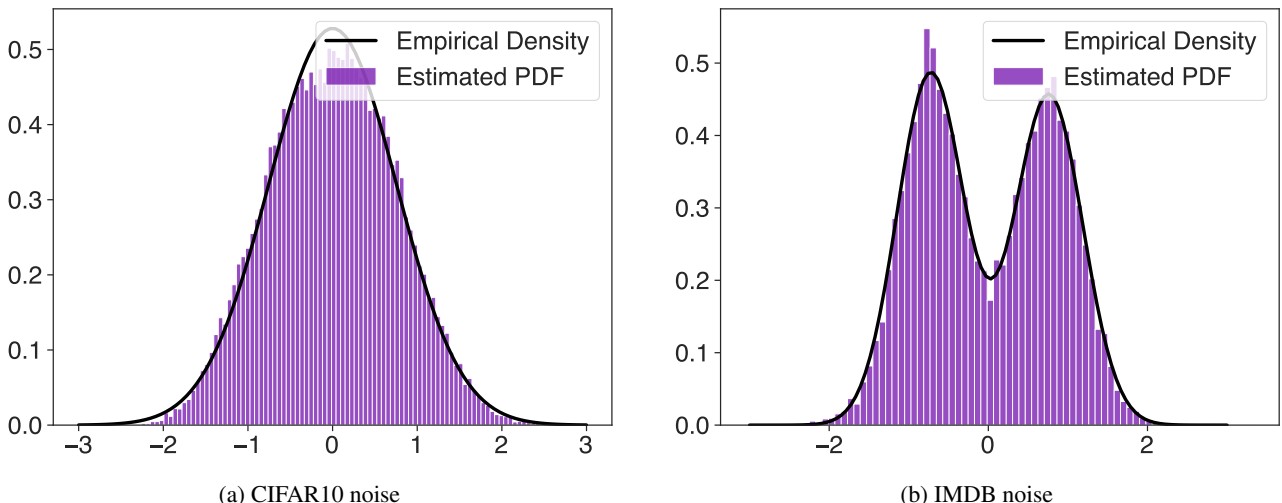

(a) CIFAR10 noise            (b) IMDB noise

Figure 10: Histograms of the estimated noise distributions for the CIFAR10 and IMDB datasets. Also shown is the estimated PDFs used to compute $\varphi$ for each case.

**IMDB:** The IMDB dataset (Maas et al., 2011) was first embedded using GLOVE (Pennington et al., 2014) into dimension 50. Then, a 2-layer random features model was applied as well as some noise added to regularize the problem. We add $sG$ where $G$ is a matrix of independent standard Gaussians. We take $s = 0.03$. This additional regularization was required in the case of text data as the covariance of the original GLOVE embedded data has extremely high condition number making numerically solving our ODEs impossible. The choice of $s = 0.03$ was found to regularize the data while maintaining the accuracy ($\approx = 73\%$) of trained models. Ultimately the data used is,

$$D_{rf} = \tanh(A' \tanh D(A + sG)). \tag{274}$$

Note that this regularization did not destroy the information contained in the original problem. Sci-kit learn achieves an accuracy of $\approx 75\%$ on the unregularized problem and the finally accuracy on the regularized problem was $\approx 73\%$.

We perform the same method as in the CIFAR10 case to first estimate $\boldsymbol{\theta}_*$ and then to estimate the distribution of the noise. We fit a mixture of two Gaussians model (GMM) to this noise. $\varphi$ is trivial to compute exactly when noise is assumed to come from a GMM. The estimated noise is again available in Figure 10.

