# OpenReview forum: "Exact risk curves of signSGD in High-Dimensions: quantifying preconditioning and noise-compression effects"
_ICML.cc/2025/Conference — ICML 2025 poster_

### Official Review · Reviewer_XYtH · 2025-03-14

**Overall Recommendation:** 4

**Summary:**

The paper studies the precise risk curves of signSGD in high dimensional limit for quadratic loss with Gaussian data under certain assumptions on the label noise. It contrasts the risk curves with SGD and quantifies the differences in terms of four effects - effective learning rate, noise compression, preconditioning and gradient noise reshaping. The exact risk curves are numerically verified for various examples.

**Claims And Evidence:**

The claims in the work are theoretical and proofs are provided in the Appendix.

**Essential References Not Discussed:**

N/A

**Experimental Designs Or Analyses:**

N/A

**Methods And Evaluation Criteria:**

N/A

**Other Comments Or Suggestions:**

N/A

**Other Strengths And Weaknesses:**

The main strength of the paper is the rigorous extension of HSGD to signSGD and then using the result for the clear discussion of the differences between the risk curves of the two.

**Questions For Authors:**

1. Can the authors explain the reasoning behind why they expect $K_\sigma$ to have a power-law spectra when $K$ has power-law spectra?

**Relation To Broader Scientific Literature:**

The work is well placed among the recent works related to analyzing preconditioned gradient algorithms.

**Theoretical Claims:**

I had a brief look at Appendix A.1, B and C, which seem mostly correct to me.

---

> ### Author Rebuttal · Authors · 2025-04-01
>
> Thank you for the review, we’re glad that you appreciate our contributions. Our expectation that $K_\sigma$ inherits a power-law spectrum from $K$ is mostly speculative. We should qualify that this claim is basis dependent. If $K$ is diagonal then $K_\sigma$ is the identity, which collapses any power-law structure.  So this is more about `very non-diagonal' matrices.
>
> This is probably best defined as having eigenvectors which are very spread out over the space, such as in CIFAR-10 or in an i.i.d. random matrix.  In the case of CIFAR-10, the map appears to preserve the power-law structure but change the power. We've included an additional plot to illustrate this effect:
> https://anonymous.4open.science/r/revision-2025-stuff-F275/cifar_eigenvalues.pdf
>
> We don't know of an existing theorem demonstrating this effect.
>
> We’ll revise this section to make ourselves clearer.

---

### Official Review · Reviewer_Rgqs · 2025-03-14

**Overall Recommendation:** 4

**Summary:**

This paper studies SignSGD, which can be viewed as Adam without the moment accumulator, in the high-dimensional limit. The main goals of the paper are to quantitatively understand the observed preconditioning and "noise compression" effects of SignSGD in practice. Toward this end, a limiting SDE (SignHSGD) and ODE (SignODE) are derived for SignSGD on square-loss linear regression. Notably, this requires non-trivial analysis due to the discontinuous sign operation. The authors then compare SignSGD to SGD in various ways, demonstrating four key differences: 1. effective learning rate, 2. noise compression, 3. diagonal preconditioning, 4. gradient noise rescaling, each of induces regimes where SignSGD may be favorable over SGD or vice versa. Most notably, SignSGD is expected to provide benefits for many non-Gaussian or heavy-tailed noise classes, supporting folk knowledge.

**Claims And Evidence:**

This claims in this paper are well-supported by rigorous theorem statements and proofs, and supporting results in the appendix. Numerical experiments are presented to support the theoretical trends.

**Essential References Not Discussed:**

None as far as I can tell.

**Experimental Designs Or Analyses:**

A sufficient description of experiment set-ups is contained in the main paper, with additional details contained in Appendix D and H.

**Methods And Evaluation Criteria:**

The numerical experiments are helpful for contextualizing the theory, and are documented in the appendix.

**Other Comments Or Suggestions:**

Minor comments:
- The authors should remember to put in the Impact Statement.
- Line 304 first column: "very SignSGD" -> "SignSGD is very".

(Possibly errant) suggestions:

- As previewed earlier, some other works have considered departing from diagonal preconditioning. The most relevant algorithm in this class is Shampoo (or more recently still, Muon). The base curvature matrix between SignSGD and Shampoo I believe are the same (AdaGrad), but Shampoo does a layer-wise "Kronecker-Factorization" rather than the diagonal. I think it would be extremely interesting to see if this type of preconditioning has similar beneficial interaction with noise, and whether it broadens the class of covariances where preconditioning is helpful.

**Other Strengths And Weaknesses:**

I think this paper is quite well-written and understandable. The main insights and arguments are likely of interest to the machine learning optimization community. I have described some of the strengths that stood out to me earlier. I have a few minor things to point out / ask:

- The analysis in this paper is restricted to linear regression and MSE loss (which is acknowledged in the Discussion), and already seems quite involved. It would be helpful to highlight to what degree the broader class of noise considered here can expand expressivity of the problem set-up. Additionally, can the "linearized" analysis here potentially extend to the NTK regime (despite the expressivity gap the NTK regime itself suffers)?

- In the main theorems (Thm 1 and 2), the gap between the discrete descent method and flow) suffers an Exp(T) factor from an application of Gronwall's inequality. This implies for fixed dimension $d$, the drift blows up exponentially in time, which doesn't seem to practically be the case. What would be required to make this bound less conservative, if possible?

**Questions For Authors:**

None beyond the above.

**Relation To Broader Scientific Literature:**

This work belongs to the general category of literature concerned with capturing the practical behavior of optimization algorithms in deep learning settings. In particular, this paper takes the approach of deriving the corresponding continuous-time SDE/ODE of its algorithm of interest, SignSGD, and deriving some key properties therein. On the other hand, most of prior works along this line study SGD. However, it is well-understood in practice that vanilla SGD experiences a host of problems in deep learning, for which various practical adjustments have been introduced. I think this work references the important prior work leading to this point, and provides some interesting tools and insights toward closing the "insight gap" between SGD and more practical methods like SignSGD and Adam. In particular, the results formalizing the unique interaction of SignSGD and noise are potentially a fruitful way of studying this type of adaptive gradient methods. Furthermore, I think the "case-by-case" results, such as when diagonal preconditioning hurts or helps (Section 4.3) are also potentially fruitful, as they may serve as a basis for understanding the relative benefit of other families of non-diagonal preconditioning methods in deep learning, such as KFAC and Shampoo.

**Theoretical Claims:**

I did not check all the proofs entirely. However, I checked the proof strategies of the main Theorems and they make sense to me. I also checked the comparisons between SGD and SignSGD; these are correct by my verification.

---

> ### Author Rebuttal · Authors · 2025-04-01
>
> Thanks for your comments, we address some key questions here.
>
> **NTK Regime**
>
> If we write consider the mean square loss for a general neural network $f(\theta, x)$ in the NTK regime such that $f(\theta, x) = f(\theta_0, x) + \nabla_{\theta} f(\theta_0, x) ^T (\theta_k - \theta_*)$ then the risk, in a student-teacher setup, is given by
>
> $$
>  \mathcal{R}(\theta_k) = E [ (f(\theta_k, x) - f(\theta_*, x)^2 ] = E[( \langle \nabla_{\theta} f(\theta_0, x), \theta_k - \theta_*\rangle^2] = (\theta_k - \theta_*)^T E[\Theta(x,x)] (\theta_k - \theta_*)
> $$
> where $\Theta$ is the NTK. Therefore in order to describe the NTK regime, we need to understand the expected signSGD updates where the “data” is given by $\nabla_{\theta} f(\theta_0, x)$. Right now, we must assume this is Gaussian. Then, if the model is such that $\nabla_{\theta} f(\theta_0, x)$ is Gaussian then our results will hold as written. In the more likely case that $\nabla_{\theta} f(\theta_0, x)$ isn’t Gaussian but is sufficiently regular (such as subgaussian) then if we can formulate our results for this larger class of data distribution then we can capture the behaviour of this NTK regime.
>
> Quite possibly our equations, if not our method of proof, already hold for subgaussian data given the good agreement with the CIFAR-10 and IMDB datasets which are not themselves Gaussian.
>
>
> **Exponential blowup in T**
>
> This $e^T$ term appears because we allow for very flexible learning rate scheduling, including schedules where the risks diverge.  With an additional restriction on the smallness of the learning rate, one could improve the exponential explosion in $T$.  It will not go away entirely, however -- optimistically one could put a $polylog(T)$.  On very long time-frames there will always be a chance for the stochastic optimizer to explode.
>
> We don't have a full picture of how a proof like this would go, but roughly you would want to show that the projection of the errors in each eigendirection of $\bar{K}$ are $O(1/\sqrt(d))$ and crucially remain bounded that way in time due to the contractive nature of the dynamics in that eigendirection.  This is also the reason the small stepsize is needed -- to ensure the dynamics tends to contract in each eigendirection, and hence the approximation errors are shrunk as well.  (Our current proof uses a resolvent argument, which is an average over all eigendirections; this is technically simpler, but would not be good as the contractive-ness of the optimizer dynamics is lost).

---

### Official Review · Reviewer_ugE8 · 2025-03-18

**Overall Recommendation:** 4

**Summary:**

The paper studies the dynamics of signSGD in a linear regression setting with Gaussian covariates $x \sim N(0,K)$ and noisy labels $y = \langle x,\theta_\ast \rangle + \epsilon$. The authors derive a limiting SDE (signHSGD) that describes the dynamics of $\theta$ as the dimension $d \to \infty$ which depends on both the covariance $K$ and the distribution of $\epsilon$. After transforming this process onto a process on the residual $r$, they derive a deterministic equivalent (signODE) which approximates the loss of of signSGD as $d \to \infty$. Finally, they compare the SDE for signHSGD to the corresponding SDE for SGD to compare the behavior of the two algorithms and to predict when signSGD should outperform vanilla SGD.

**Update after Rebuttal:** I am satisfied with the author's responses and have updated my score.

**Claims And Evidence:**

This paper is focused on theoretical understanding and the technical claims are supported by the proofs in Appendices A, B.

There is a minor claim/conjecture which is unsupported (lines 416-424). It would be good to include a few words about why the authors expect this to hold.

**Essential References Not Discussed:**

n/a

**Experimental Designs Or Analyses:**

n/a

**Methods And Evaluation Criteria:**

This paper is focused on developing theoretical understanding in a linear regression setting, and the experiments on more realistic datasets support the idea that Gaussian universality may allow the analysis to extend beyond the Gaussian setting. However, this is not a central claim of the paper.

**Other Comments Or Suggestions:**

- The use of $\eta_t'$ for the signSGD learning rate and $\eta_t := \eta(t)$ for the rescaled learning rate is a bit confusing
- The discussion for the effects of $\epsilon$-compression (4.2) and preconditioning (4.3) are clear but the noise reshaping (section 4.4) is a bit handwavy. In particular, the conjectures in the last paragraph are difficult to follow and could merit additional explanation in the Appendix, even if there is no rigorous theory to back them up. It could also be useful to run some ablations with and without the reshaped noise covariance to check if it's actually beneficial or just a side-effect of the preconditioning.
- In Assumption 5, $z$ is not defined. Is the assumption that this holds for all $z$? Could this assumption simply be replaced with the assumption that $v^T (\theta_0 - \theta^\ast)$ is sub-Gaussian for any $v$? Also, why does this assumption hold for a deterministic $\theta_0,\theta_\ast$? If $v = R(z;\overline{K})_i$ is aligned with $\theta_0 - \theta^\ast$ then Assumption 5 forces $\|\theta_0 - \theta^\ast\| \lesssim d^{-1/2}$?

**Other Strengths And Weaknesses:**

Strengths:
- The paper is generally well written and includes an extended discussion of the different terms in eq. 8
- The theory seem to match experiments incredibly well (in the linear regression setting)
- The derivation of the SDE for signHSGD is highly nontrivial

Weaknesses:
- As acknowledged by the authors, this paper focuses on the simple setting of linear regression with Gaussian covariates. However, even in this setting, we have a very poor understanding of coordinate-wise adaptive optimizers like signSGD or Adam.
- The paper focuses on online SGD with batch size 1. It would be interesting to extend these results to the more general case.
- There is limited discussion about the effect of the noise reshaping $K \to K_\sigma$.

**Questions For Authors:**

- This paper focuses on the batch-size $1$ setting. How are both signHSGD and the strength of the SDE approximation affected by the batch size? Would your analysis recover the "square root" scaling rule in Malladi et al. for Adam? Additionally, would the analysis extend to batch sizes that grow with $d$?

- The analysis restricts to learning rates $\eta_t' = \Theta(1/d)$ (eq. 6). Is the issue that if $\eta_t' \gg 1/d$ the noise will dominate and the SDE becomes degenerate in the limit $d \to \infty$?

**Relation To Broader Scientific Literature:**

This paper is attempting to understand the effectiveness of coordinate-wise adaptivity (as in Adam) in a concrete theoretical setting. The approach in this paper is most closely related to (Collins-Woodfin & Paquette 2023) which performed a similar analysis for SGD.

**Theoretical Claims:**

I skimmed the proofs in appendices A, B but did not verify their correctness.

---

> ### Author Rebuttal · Authors · 2025-04-01
>
> We thank the author for their detailed review and address weaknesses and questions below.
>
> ## Weaknesses
>
>
> **Noise reshaping**
>
> For the $K \to K_\sigma$ mapping, there is strong dependence on the eigenbasis of $K$. For example if $K$ is diagonal then $K_\sigma = Id$. Otherwise, if the eigenbasis is relatively random (such as what is seen in CIFAR and in Figure 4), the operation can preserve some spectral properties of $K$.  For example, in CIFAR-10, $K_\sigma$ appears to preserve the power-law structure of $K$, albeit with a different power (plots [here](https://anonymous.4open.science/r/revision-2025-stuff-F275/cifar_eigenvalues.pdf)).
>
>
> One major point (discussed below in response to the learning-rate) is that the the trace of $K_\sigma$ can be much larger than the trace of $K$, which will cause sign-SGD gradient noise to be a major slow-down
> Rigorously saying more is probably difficult. But, we'll rewrite this section in order to be clearer.
>
> **Assumption 5**
>
> Thank you for catching this. We like your suggestion to instead assume $v^T(\theta_0 - \theta_*)$ is subgaussian with constant $O(d^{-1/2})$. You're right that general deterministic initialization and targets do not fit the assumption.  For the record, the assumption is for all $z \in B(||K||)$.
>
>
> ## Questions
>
> **Batch sizes**
>
> This is a very good question -- thanks for bringing it up.  We have worked out some of the mathematics for batch sizes small compared to $d$, and plan to add the following results to an appendix.
>
> So first, just to be clear, we are following usual optimization (and standard package) conventions, and we are applying the minibatch average prior to applying the sign-function.
>
> We will take a fixed batch size b independent of the dimension of the problem.  One could also look at other scaling regimes, and we expect (based on the fixed batch size case) that the behavior continues to hold for batch sizes that satisfy (b/d) to 0, but we’re only 100% confident for fixed $b,$ and $d \to \infty$.  The good news is that even in this limited setting, the story is already interesting.
>
> The bias-term (or descent-term) is the only term affected by minibatching.  The gradient noise term (the Brownian term in homogenized sgd) is unaffected.
>
> The homogenized-SGD equation is changed to
>
> $$ \mathrm{d}\Theta_t= -\eta_t N_b \frac{\varphi^{(b)}(\left(\mathcal{R}({\Theta}_t)\right)}{\sqrt{2 \mathcal{R}(\Theta_t)}}
> \overline{\mathbf{K}} \left(\Theta_t-\theta^{*}\right) \mathrm{d}t
> $$
>
> $$ + \eta_t \sqrt{\frac{2 \mathbf{K}_\sigma}{\pi d}} \mathrm{~d} \mathbf{B}_t$$
>
> The $N_b$ is a numerical constant (the mean of a chi-variable with $b$ degrees of freedom, to the mean of a chi variable with $1$ degree of freedom).  It is asymptotic to $\sqrt{b}$, consistent with what is shown in Malladi et al.
>
> The $\varphi^{(b)}$ is the $\varphi$ of a new convolved noise.  It is the $\varphi$ that results from looking at $\sum_{i=1}^b \epsilon^{(i)} \omega_i$ where $\epsilon^{(i)}$ are iid copies of the label noise and where $\omega$ is an independent random vector, drawn uniformly from the sphere of dimension $b$.
>
> We have run some simulations implementing this (code is available in the anonymous code repository for the paper). Simulations at fixed learning rate, varying batch size are available [here](https://anonymous.4open.science/r/revision-2025-stuff-F275/rademacher_batch_comparison_dimension_500_evals_logspace_num_runs_10.pdf) and with $\sqrt{b}$ scaling [here](https://anonymous.4open.science/r/revision-2025-stuff-F275/rademacher_batch_comparison_dimension_500_evals_logspace_num_runs_10_rescaled_lr.pdf).
>
>
> **Learning rate scaling**
>
> Indeed, the choice of $\eta’ = O(1/d)$ is needed to prevent (gradient) noise domination. See the Hessian term of equation 94 where the Hilbert-Schmidt inner product is $O(Tr(K_\sigma))$. In this setting, if $\eta’$ decays slower than $1/d$ this noise term would dominate. Had $\eta’$ decayed faster, the noise term would vanish and we would be performing gradient flow in the high-dimensional limit.
>
> For SGD it is possible to instead rescale by the “intrinsic dimension” of the data, defined as the trace of covariance normalized by its operator norm ( $Tr(K) / ||K|| $). We think a similar change could be made for signSGD but some care has to be taken since signSGD reshapes the gradient covariance to $K_\sigma$. This connects to our discuss about the reshaping of gradient noise. Given a diagonal $K$ with intrinsic dimension $O(\sqrt{d})$, signSGD still has intrinsic dimension $O(d)$ since $K_\sigma = Id$. This leads to signSGD having learning rates $O(d^{1/2})$ smaller than vanilla SGD to guarantee convergence, leading to slow dynamics.
>
> Let us know if we can clarify anything else; if we have addressed your concerns satisfactorily, we hope that you will consider revisiting your review score.

---

### Official Review · Reviewer_ZURo · 2025-03-20

**Overall Recommendation:** 4

**Summary:**

The authors analyze Sign-SGD for linear regression in high dimensions by deriving limiting differential equations that describe its behavior. Their analysis quantifies four main effects: learning rate adjustment, noise compression, diagonal preconditioning, and gradient noise reshaping. The paper includes theoretical proofs and experimental validation on both synthetic and real datasets.

**Claims And Evidence:**

Main Claims and Evidence:

Claims:
1. Sign-HSGD and its deterministic equivalent ODE are good models for the risk dynamics induced by Sign-SGD
2. The risk curves of Sign-SGD are well approximated by the risk curves of Sign-HSGD and this approximation improves as dimension grows.
3. Convergence to a neighbourhood of the solution (?) under a fixed learning rate
4. The condition number affects whether Sign-SGD or SGD is should be favoured

Evidence:
1. Figure 1
2. Theorem 1 / Theorem 2
3. Theorem 3
4. Theorem 4

The evidence to me was convincing and followed from my intuition of the algorithm, however I did not review the appendices in detail. I especially appreciated the mixture of theoretical analysis with a grounded set of experiments which verified the results derived seemed to make sense.

**Essential References Not Discussed:**

Possibly might want to look at "Heavy-Tailed Class Imbalance and Why Adam Outperforms Gradient Descent on Language Models"

**Experimental Designs Or Analyses:**

The experimental designs at there face seem sound, and tailored to fit the larger narrative of the paper.

**Methods And Evaluation Criteria:**

Benchmarks were a mixture of toy problems like linear regression under varying noise and preconditioned assumptions which more closely resembled the theoretical setting analyzed, as well as a set of more practical experiments on common ML benchmarks like CIFAR10 and IMDB.

**Other Comments Or Suggestions:**

- The "In a nonconvex setting, identifying ...." paragraph is broken and needs to be revised.

**Other Strengths And Weaknesses:**

Strengths:

1. Comprehensive theoretical framework:
- Proves convergence to limiting equations
- Derives exact formulas for limiting risk
- Quantifies specific effects (learning rate, noise compression, preconditioning, noise reshaping)

2. Strong empirical validation:
- Theory matches experiments even at moderate dimension (d=500)
- Works on both synthetic and real datasets
- Demonstrates convergence under different noise conditions

3. Clear practical implications:
- Shows when Sign-SGD might outperform SGD (based on condition numbers)
- Explains behavior under different noise distributions

Weaknesses:

1. Limited scope:
- Theory only covers linear regression
- Strongest results require specific noise conditions (C² density near zero)
- Extensions to non-smooth noise only work above risk threshold

2. Some findings lack theoretical foundation:
- Success on real (non-Gaussian) data isn't fully explained
- Authors note this is left for future work

3. Some practical aspects not addressed:
- Doesn't analyze mini-batch settings
- Doesn't fully connect to practical deep learning scenarios / analyze non-convex settings.

**Questions For Authors:**

- Can you explain: "But we expect the conclusion of (27) remains mostly true in well-conditioned settings."
- From the above:  I would like to double check that the result from theorem 3 roughly align with standard results from stochastic convex optimization regarding SGD with a fixed step-size. If they do not would the authors explain in more detail the differences.

**Relation To Broader Scientific Literature:**

I think that this work seems to address a unique and important problem posed by modern optimization research, and I believe a better understanding of the optimization dynamics of algorithms like sign decent would benefit the larger machine learning community.

**Theoretical Claims:**

I did not verify the claims in detail but based on my experience they seem reasonable. I would like to double check that the result from theorem 3 roughly align with standard results from stochastic convex optimization regarding SGD with a fixed step-size. If they do not would the authors explain in more detail the differences.

---

> ### Author Rebuttal · Authors · 2025-04-01
>
> We thank the reviewer for their comments, and address some key points below.
>
> ## Minbatching:
>
> We have added more details regarding batch sizes $b$ in our response to Reviewer ugE8. One particular consequence of minibatching is that our condition on the behaviour of the noise near $0$ relaxes significantly. Averaging multiple sources of noise effectively conditions the noise even for $b = 2$. Numerical results can be found [here](https://anonymous.4open.science/r/revision-2025-stuff-F275/rademacher_batch_comparison_dimension_500_evals_logspace_num_runs_10.pdf) for fixed learning rate, varying batch size.
>
> ## Questions:
>
> **First question**
>
> (27) is comparing the risk obtained by the optimally scheduled signSGD to optimally scheduled SGD when data is isotropic. We don’t know the optimal learning rate schedule in non-isotropic settings but what we expect is that whether or not signSGD outperforms SGD depends generally on the magnitude of $\psi$.
>
> The argument why can be sketched like: for any signSGD learning rate $\eta^{S}(t)$ run SGD with a learning rate of $\eta(t)  =\eta^{S}(t)\phi(R_t) / \sqrt{R_t}$. Now the descent terms are the same up to the $K$ vs $\overline{K}$ distinction. With this learning rate we see the function $\psi$ in the variance term on the SGD risk equation. If the various matrices involved ($K, \overline{K}, K_\sigma$) are similar enough (as they are when $K=Id$) then whether or not signSGD is beaten by SGD with this learning rate is again determined by the magnitude of $\psi$. This “similar enough” statement is what we mean by well-conditioned settings.
>
> **Second question**
>
> For SGD in the same setting, the limiting risk can be found in Equation (254) and is
>
> $$
> \frac{\eta v^2 Tr(K) }{2(2d-Tr(K)\eta)}.
> $$
>
> For sufficiently small $\eta$, the limiting SGD risk is $O(\eta v^2 Tr(K) / d)$.  This is consistent with 'neighborhood' convergence for fixed step-size SGD when $v > 0$ (e.g. Theorem 5.5 of Garrigos-Gower '23, Handbook of Convergence Theorems) and convergence with fixed step-size in the case $v=0$.
>
> In contrast we can use Theorem 3 to approximate the limiting signSGD risk as up to constants:
>
> $$
> \frac{\eta Tr(\overline{K})}{d} \max \left (\frac{\eta Tr(\overline{K})}{d}, v \right)
> $$
>
> Notably, even if $v = 0$ this limiting risk is not $0$. The simple explanation is that with SGD, the average magnitude of the gradient naturally decreases with the risk and so we effectively take smaller steps; however for signSGD the magnitude of the gradient is always $\sqrt{d}$ and does not change with the risk. To achieve similar results with signSGD the stepsize needs to be decreased. This aligns with what is seen in practice for algorithms like Adam.
>
>
> ## Other:
>
> "In a nonconvex setting, identifying ...." paragraph.  We have attempted to improve the wording for clarity, but we’d be happy for additional guidance here.  Our intention is to say that dividing by the norm of the gradient could be reasonable in vanishing gradient situations, such as near saddles.

---

### Decision · Program_Chairs · 2025-05-01

**Decision:**

Accept (poster)

**Comment:**

The paper studies the dynamics of signSGD in a linear regression setting with Gaussian covariates and label noise and derives a limiting SDE (SignHSGD). This paper makes solid theoretical contribution towards understanding coordinate-wise adaptivity in optimization algorithms, which is at the core of the optimization theory in the era of LLMs. Though the theory is limited to linear regression, the result and analysis is already quite complicated, yet they predict the experiments incredibly well, even for the real dataset like CIFAR10 (though it is linear regression). Reviewers unanimously recommend to accept.